# Improved Regret Bounds for Gaussian Process Upper Confidence Bound in Bayesian Optimization

**Shogo Iwazaki**
LY Corporation
Tokyo, Japan
siwazaki@lycorp.co.jp

## Abstract

This paper addresses the Bayesian optimization problem (also referred to as the Bayesian setting of the Gaussian process bandit), where the learner seeks to minimize the regret under a function drawn from a known Gaussian process (GP). Under a Matérn kernel with a certain degree of smoothness, we show that the Gaussian process upper confidence bound (GP-UCB) algorithm achieves $\widetilde{O}(\sqrt{T})$ cumulative regret with high probability. Furthermore, our analysis yields $O(\sqrt{T \ln^2 T})$ regret under a squared exponential kernel. These results fill the gap between the existing regret upper bound for GP-UCB and the best-known bound provided by Scarlett [46]. The key idea in our proof is to capture the concentration behavior of the input sequence realized by GP-UCB, enabling a more refined analysis of the GP's information gain.

## 1 Introduction

We study the Bayesian optimization (BO) problem, where the learner seeks to minimize the regret under a random function drawn from a known Gaussian process (GP) [18, 19]. Throughout this paper, we focus on the GP-UCB algorithm [51], which combines the posterior distribution of GP with the optimism principle. Due to its simple algorithm construction and general theoretical framework provided by Srinivas et al. [51], GP-UCB has played an important role in the advancement of the BO field. On the other hand, our theoretical understanding of the performance of GP-UCB has not been improved from [51] in the Bayesian setting, while its frequentist counterpart is studied in several existing works [11, 61]. Specifically, the current regret upper bound for GP-UCB, as provided by Srinivas et al. [51], is known to be worse than that of the algorithm in [46], which achieves state-of-the-art $O(\sqrt{T \ln T})$ cumulative regret. Then, the natural question is whether there is further room for improvement in the existing regret upper bound of GP-UCB. This paper provides an affirmative answer to this question by showing that GP-UCB achieves $\widetilde{O}(\sqrt{T})$ regret with high probability.

**Contribution.** We summarize our contributions as follows.

- We show that the GP-UCB proposed by Srinivas et al. [51] achieves $\widetilde{O}(\sqrt{T})$ regret with high probability under a Matérn kernel with a certain degree of smoothness (precise condition is provided in Theorem 3). Here, $\widetilde{O}(\cdot)$ is the order notation that hides polylogarithmic dependence. This result is comparable to state-of-the-art $O(\sqrt{T \ln T})$ regret provided by Scarlett [46] up to a polylogarithmic factor and strictly improves upon the existing $\widetilde{O}(T^{\frac{\nu+d}{2\nu+d}})$ upper bound of GP-UCB [51, 58]. Here, $d$ and $\nu$ denote the dimension of the input domain and smoothness parameter, respectively.

39th Conference on Neural Information Processing Systems (NeurIPS 2025).

- Furthermore, for a squared exponential kernel, we establish $O\left(\sqrt{T \ln^2 T}\right)$ cumulative regret of GP-UCB. This improves the existing $O\left(\sqrt{T \ln^{d+2} T}\right)$ upper bound provided by Srinivas et al. [51] for any $d \geq 1$.

- The key idea behind our analysis is to refine the existing information gain bounds by leveraging algorithm-dependent behavior and sample path properties of the GP. We also discuss the applicability of this technique to other algorithms and settings in Section 4.

## 1.1 Related Works

BO has been extensively studied in the past few decades. Some of them are constructed so as to maximize the utility-based acquisition function defined through the GP posterior, including expected improvement [37], knowledge gradient [17], and the entropy-based algorithms [24]. The theoretical aspect of BO has also been actively studied through the lens of the bandit algorithms, such as GP-UCB [51], Thompson sampling [43], and information directed sampling [44]. In contrast to the noisy observation setting, which these algorithms focus on, algorithms for the noise-free setting form a separate line of research [14, 23, 32]. Extensions of these algorithms to more advanced settings have also been well-studied, e.g., contextual [34], parallel observation [15], high-dimensional [29], time-varying [6], and multi-fidelity setting [30]. Unlike the Bayesian assumption on the objective function adopted in this paper, existing works also extensively study the frequentist assumption of the function, which is also referred to as the frequentist setting of BO or GP bandits [7, 9, 11, 26, 35, 45, 47, 56, 59].

Among the existing studies, [46] is closely related to this paper, which propose a successive elimination-based algorithm and shows an $O(\sqrt{T \ln T})$ upper bound and an $\Omega(\sqrt{T})$ lower bound of the cumulative regret for a one-dimensional BO problem. The fundamental theoretical assumptions and the high-level idea of our analysis are built on the proof provided by Scarlett [46]. Following [46], Wang et al. [60] also proves similar regret guarantees under the one-dimensional Brownian motion.

In addition to [46], some parts of our analysis are inspired by the technique leveraged in [8, 28]. Firstly, Cai et al. [8] studies the GP-UCB algorithm through a relaxed version of regret, which is called *lenient regret*. In our analysis, the cumulative regret is decomposed into the lenient regret-based term, and we leverage their technique to analyze it. Secondly, Janz et al. [28] proposed the input partitioning-based algorithm for obtaining a superior regret in the frequentist setting. Roughly speaking, the high-level idea of their analysis is based on the fact that tighter information gain bounds can be obtained within a properly shrinking partition of the input. The key idea provided in Section 3.1 is motivated by this fact, while our analysis itself is substantially different from that in [28].

## 2 Preliminaries

Let $f : \mathcal{X} \to \mathbb{R}$ be a black-box objective function whose input domain $\mathcal{X}$ is $\mathcal{X} := [0, r]^d$ with some $r > 0$. At each step $t \in \mathbb{N}_+$, the learner chooses a query point $x_t \in \mathcal{X}$, and then receives a noisy observation $y_t = f(x_t) + \epsilon_t$. Here, $\epsilon_t$ is a mean-zero noise random variable. We consider a Bayesian setting, where the objective function $f$ and the noise sequence $(\epsilon_t)$ are drawn from a known zero-mean Gaussian process (GP) and a Gaussian distribution, respectively. We formally describe it using the following assumptions.

**Assumption 1.** *Let $k : \mathcal{X} \times \mathcal{X} \to \mathbb{R}$ be the known positive definite kernel with $\forall x \in \mathcal{X}, k(x, x) \leq 1$. Then, assume $f \sim \mathcal{GP}(0, k)$, where $\mathcal{GP}(0, k)$ denotes the mean-zero GP characterized by the covariance function $k$.*

**Assumption 2.** *The noise sequence $(\epsilon_t)_{t \in \mathbb{N}_+}$ is mutually independent. Furthermore, assume $\epsilon_t \sim \mathcal{N}(0, \sigma^2)$, where $\sigma > 0$ is the known constant. Here, $\mathcal{N}(\mu, \sigma^2)$ denotes the Gaussian distribution with mean $\mu$ and variance $\sigma^2$.*

These are standard sets of assumptions in the existing theory of BO [43, 51]. Specifically, in Assumption 1, we focus on the following squared exponential (SE) kernel $k_{\mathrm{SE}}$ and Matérn kernel

**Algorithm 1** Gaussian process upper confidence bound

---

**Require:** Kernel $k$, confidence width parameters $(\beta_t^{1/2})_{t \in \mathbb{N}_+}$.
 1: **for** $t = 1, 2, \dots$ **do**
 2:  $\boldsymbol{x}_t \leftarrow \arg\max_{\boldsymbol{x} \in \mathcal{X}} \mu(\boldsymbol{x}; \mathbf{X}_{t-1}, \boldsymbol{y}_{t-1}) + \beta_t^{1/2} \sigma(\boldsymbol{x}; \mathbf{X}_{t-1})$.
 3:  Observe $y_t$ and update the posterior mean and variance.
 4: **end for**

---

$k_{\text{Matérn}}$:

$$k_{\text{SE}}(\boldsymbol{x}, \widetilde{\boldsymbol{x}}) = \exp\left(-\frac{\|\boldsymbol{x} - \widetilde{\boldsymbol{x}}\|_2^2}{2\ell^2}\right), \quad k_{\text{Matérn}}(\boldsymbol{x}, \widetilde{\boldsymbol{x}}) = \frac{2^{1-\nu}}{\Gamma(\nu)}\left(\frac{\sqrt{2\nu}\|\boldsymbol{x} - \widetilde{\boldsymbol{x}}\|_2}{\ell}\right)^\nu J_\nu\left(\frac{\sqrt{2\nu}\|\boldsymbol{x} - \widetilde{\boldsymbol{x}}\|_2}{\ell}\right), \tag{1}$$

where $\ell > 0$ and $\nu > 0$ are the known lengthscale and smoothness parameters, respectively. In addition, $J_\nu(\cdot)$ and $\Gamma(\cdot)$ respectively denote modified Bessel and Gamma functions. Under Assumptions 1 and 2, the learner can infer the function $f$ through the GP posterior distribution. Let $\mathcal{H}_t := (\boldsymbol{x}_i, y_i)_{i \leq t}$ be the history that the learner obtained up to the end of step $t$. Given $\mathcal{H}_t$, the posterior distribution of $f$ is again GP, whose posterior mean and variance are respectively defined as

$$\mu(\boldsymbol{x}; \mathbf{X}_t, \boldsymbol{y}_t) = \boldsymbol{k}(\mathbf{X}_t, \boldsymbol{x})^\top (\mathbf{K}(\mathbf{X}_t, \mathbf{X}_t) + \sigma^2 \boldsymbol{I}_t)^{-1} \boldsymbol{y}_t, \tag{2}$$

$$\sigma^2(\boldsymbol{x}; \mathbf{X}_t) = k(\boldsymbol{x}, \boldsymbol{x}) - \boldsymbol{k}(\mathbf{X}_t, \boldsymbol{x})^\top (\mathbf{K}(\mathbf{X}_t, \mathbf{X}_t) + \sigma^2 \boldsymbol{I}_t)^{-1} \boldsymbol{k}_t(\mathbf{X}_t, \boldsymbol{x}), \tag{3}$$

where $\boldsymbol{k}(\mathbf{X}_t, \boldsymbol{x}) := [k(\boldsymbol{x}, \widetilde{\boldsymbol{x}})]_{\boldsymbol{x} \in \mathbf{X}_t}$ and $\boldsymbol{y}_t := (y_1, \dots, y_t)^\top$ are the $t$-dimensional kernel and output vectors, respectively. Here, we set $\mathbf{X}_t = (\boldsymbol{x}_1, \dots, \boldsymbol{x}_t)$. Furthermore, $\mathbf{K}(\mathbf{X}_t, \mathbf{X}_t) := [k(\boldsymbol{x}, \widetilde{\boldsymbol{x}})]_{\boldsymbol{x}, \widetilde{\boldsymbol{x}} \in \mathbf{X}_t}$ and $\boldsymbol{I}_t$ denote $t \times t$-gram matrix and $t \times t$-identity matrix, respectively.

**Learner's goal.** Under the total step size $T \in \mathbb{N}_+$, the learner's goal is to minimize the cumulative regret $R_T := \sum_{t \in [T]} f(\boldsymbol{x}^*) - f(\boldsymbol{x}_t)$, where $\boldsymbol{x}^* \in \arg\max_{\boldsymbol{x} \in \mathcal{X}} f(\boldsymbol{x})$ and $[T] = \{1, \dots, T\}$.

**Maximum information gain.** To quantify the regret, the existing theory utilizes the following information-theoretic quantity $\gamma_T(\mathcal{X})$ arising from GP:

$$\gamma_T(\mathcal{X}) = \sup_{\boldsymbol{x}_1, \dots, \boldsymbol{x}_T \in \mathcal{X}} I(\mathbf{X}_T), \quad \text{where } I(\mathbf{X}_T) = \frac{1}{2} \ln \det(\boldsymbol{I}_T + \sigma^{-2} \mathbf{K}(\mathbf{X}_T, \mathbf{X}_T)). \tag{4}$$

The quantity $\gamma_T(\mathcal{X})$ is referred to as the *maximum information gain* (MIG) over $\mathcal{X}$ [51], since $I(\mathbf{X}_T)$ equals the mutual information between the function values $(f(\boldsymbol{x}_t))_{t \in [T]}$ and the outputs $(y_t)_{t \in [T]}$ under Assumptions 1 and 2, and the input sequence $\mathbf{X}_t = (\boldsymbol{x}_1, \dots, \boldsymbol{x}_t)$. MIG plays a vital role in the theoretical analysis of BO, and its increasing speed is analyzed in several commonly used kernels. For example, $\gamma_T(\mathcal{X}) = O(\ln^{d+1} T)$ as $T \to \infty$ under $k = k_{\text{SE}}$ [51]. For the notational convenience, we also define $\gamma_i(\mathcal{X}) = \gamma_{\lceil i \rceil}(\mathcal{X})$ for any non-integer $i > 0$.

**Probabilistic property of GP sample path.** The existing theory of GP-UCB under the Bayesian setting utilizes the regularity conditions of the realized sample path of GP. We summarize the existing known properties of the GP sample path in the following lemmas.

**Lemma 1** (Lipchitz condition of sample path, e.g., [51]). *Suppose $k = k_{\text{SE}}$ or $k = k_{\text{Matérn}}$ with $\nu > 2$. Assume Assumption 1. Then, there exist the constants $a, b > 0$ such that*

$$\forall L > 0, \; \mathbb{P}\left(\forall \boldsymbol{x}, \widetilde{\boldsymbol{x}} \in \mathcal{X}, \; |f(\boldsymbol{x}) - f(\widetilde{\boldsymbol{x}})| \leq L \|\boldsymbol{x} - \widetilde{\boldsymbol{x}}\|_1\right) \geq 1 - da \exp\left(-\frac{L^2}{b^2}\right). \tag{5}$$

**Lemma 2** (Sample path condition for the global maximizer, e.g., [13, 14, 46]). *Suppose $k = k_{\text{SE}}$ or $k = k_{\text{Matérn}}$ with $\nu > 2$. Assume Assumption 1. Then, for any $\delta_{\text{GP}} \in (0, 1)$, there exist the strictly positive constants $c_{\text{gap}}, c_{\text{sup}}, c_{\text{quad}}, \rho_{\text{quad}} > 0$ such that the following statements simultaneously hold with probability at least $1 - \delta_{\text{GP}}$:*

  *1. The function $f$ has a unique maximizer $\boldsymbol{x}^* \in \mathcal{X}$ such that $f(\boldsymbol{x}^*) > f(\widetilde{\boldsymbol{x}}^*) + c_{\text{gap}}$ holds for any local maximizer $\widetilde{\boldsymbol{x}}^* \in \mathcal{X}$ of $f$.*

2. *The sup-norm of the sample path is bounded as $\|f\|_\infty \le c_{\text{sup}}$.*

3. *The function $f$ satisfies $\forall \boldsymbol{x} \in \mathcal{B}_2(\rho_{\text{quad}}; \boldsymbol{x}^*)$, $f(\boldsymbol{x}^*) - c_{\text{quad}}\|\boldsymbol{x}^* - \boldsymbol{x}\|_2^2 \ge f(\boldsymbol{x})$, where $\mathcal{B}_2(\rho_{\text{quad}}; \boldsymbol{x}^*) \coloneqq \{\boldsymbol{x} \in \mathcal{X} \mid \|\boldsymbol{x}^* - \boldsymbol{x}\|_2 \le \rho_{\text{quad}}\}$ is the L2-ball on $\mathcal{X}$, whose radius and center are $\rho_{\text{quad}}$ and $\boldsymbol{x}^*$, respectively.*

Lemma 1 states that the sample path $f$ of GP is a Lipschitz function with high probability. This property is leveraged in the theory of GP-UCB to control the discretization error arising from the confidence bound construction in the continuous input domain. As described in [51], Lemma 1 is a direct consequence of Theorem 5 in [21] under the existence of fourth-order mixed partial derivatives of the kernel, which are satisfied under $k = k_{\text{SE}}$ and $k = k_{\text{Matérn}}$ with $\nu > 2^1$. Lemma 2 specifies the regularity condition of $f$ related to the maximizer $\boldsymbol{x}^*$. Here, property 1 is implied from the fact that the GP-sample path has a unique maximizer almost surely under $k_{\text{SE}}$ and $k_{\text{Matérn}}$ [e.g., Lemma 2.6 in 33]. Property 2 is implied from, e.g., the compactness of $\mathcal{X}$ and the almost-sure continuity of the sample path under $k_{\text{SE}}$ and $k_{\text{Matérn}}$. Property 3 also holds automatically under $k = k_{\text{SE}}$ and $k = k_{\text{Matérn}}$ with $\nu > 2$ and is used in existing works. See Theorem 5 in [13], Assumption 3 in [46], and the discussions provided by them for further details. Note that the properties in Lemma 2 are not used in the existing proof of GP-UCB in [51]. As described in the next section, we analyze the realized input sequence $\mathbf{X}_T$ of GP-UCB by relating it to conditions in Lemma 2.

**Summary of existing analysis of GP-UCB.** We briefly summarize the existing analysis of GP-UCB (Algorithm 1) provided by Srinivas et al. [51]. Based on Assumptions 1 and 2, we can construct the high-probability confidence bound of the underlying function value $f(\boldsymbol{x})$ for each $\boldsymbol{x}$ and $t \in \mathbb{N}_+$ through the posterior distribution of $f(\boldsymbol{x})$. Specifically, by choosing a properly designed finite representative input set $\mathcal{X}_t \subset \mathcal{X}$ and taking into account the discretization error with Lemma 1, Srinivas et al. [51] showed the following events hold simultaneously with probability at least $1 - \delta$:

1. **Confidence bound.** For any $t \in \mathbb{N}_+$, the function value at the queried point $\boldsymbol{x}_t$ satisfies $\mu(\boldsymbol{x}_t; \mathbf{X}_{t-1}, \boldsymbol{y}_{t-1}) - \beta_t^{1/2}\sigma(\boldsymbol{x}_t; \mathbf{X}_{t-1}) \le f(\boldsymbol{x}_t)$. Furthermore, for any $t \in \mathbb{N}_+$, any function value $f(\boldsymbol{x})$ on $\mathcal{X}_t$ satisfies $f(\boldsymbol{x}) \le \mu(\boldsymbol{x}; \mathbf{X}_{t-1}, \boldsymbol{y}_{t-1}) + \beta_t^{1/2}\sigma(\boldsymbol{x}; \mathbf{X}_{t-1})$.

2. **Discretization error.** The discretization error arising from $\mathcal{X}_t$ is at most $1/t^2$. Namely, $|f(\boldsymbol{x}) - f([\boldsymbol{x}]_t)| \le 1/t^2$ holds for any $\boldsymbol{x} \in \mathcal{X}$ and $t \in \mathbb{N}_+$, where $[\boldsymbol{x}]_t$ denotes one of the closest points of $\boldsymbol{x}$ on $\mathcal{X}_t$.

In the above statements, $\beta_t^{1/2}$ is chosen based on the constants $a$, $b$ in Lemma 1 and the length $r$ of $\mathcal{X}$, and is defined as

$$\beta_t = 2\ln\frac{2t^2\pi^2}{3\delta} + 2d\ln\left(t^2 dbr\sqrt{\ln\frac{4da}{\delta}}\right). \tag{6}$$

The above two events and the UCB-selection rule for $\boldsymbol{x}_t$ imply

$$R_T = \sum_{t=1}^T f(\boldsymbol{x}^*) - f([\boldsymbol{x}^*]_t) + \sum_{t=1}^T f([\boldsymbol{x}^*]_t) - f(\boldsymbol{x}_t) \le \frac{\pi^2}{6} + 2\beta_T^{1/2}\sum_{t=1}^T \sigma(\boldsymbol{x}_t; \mathbf{X}_{t-1}). \tag{7}$$

In the above expression, the upper bound $\sum_{t=1}^T f(\boldsymbol{x}^*) - f([\boldsymbol{x}^*]_t) \le \sum_{t=1}^T 1/t^2 \le \pi^2/6$ follows from the second event (discretization error). The inequality $\sum_{t=1}^T f([\boldsymbol{x}^*]_t) - f(\boldsymbol{x}_t) \le 2\beta_T^{1/2}\sum_{t=1}^T \sigma(\boldsymbol{x}_t; \mathbf{X}_{t-1})$ also follows from the first event (confidence bound) and the definition of $\boldsymbol{x}_t$. See the proof of Theorem 2 in [51] for details. The above inequality suggests that the regret upper bound of GP-UCB depends on the sum of the posterior standard deviations $\sum_{t=1}^T \sigma(\boldsymbol{x}_t; \mathbf{X}_{t-1})$. Srinivas et al. [51] provides the upper bound of this term by leveraging the information gain $I(\mathbf{X}_T)$ as follows:

$$\sum_{t=1}^T \sigma(\boldsymbol{x}_t; \mathbf{X}_{t-1}) \le \sqrt{CTI(\mathbf{X}_T)} \le \sqrt{CT\gamma_T(\mathcal{X})}, \tag{8}$$

where $C = \frac{2}{\ln(1+\sigma^{-2})}$. From Eqs. (7) and (8), we conclude that the regret upper bound of GP-UCB is $O\left(\sqrt{\beta_T T \gamma_T(\mathcal{X})}\right)$ with probability at least $1 - \delta$. By combining the explicit upper bound

---
[1]Differentiability of $k_{\text{Matérn}}$ is derived in the existing works, e.g., Chapter 2.7 in [52].

of $\gamma_T(\mathcal{X})$ [51, 58], we also obtain $O\left(\sqrt{T \ln^{d+2} T}\right)$ and $\widetilde{O}\left(T^{\frac{\nu+d}{2\nu+d}}\right)$ regret upper bounds for SE and Matérn kernels, respectively.

## 3 Improved Regret Bound for GP-UCB

The following theorem presents our main result: a new regret upper bound for GP-UCB.

**Theorem 3** (Improved regret upper bound for GP-UCB). *Suppose Assumptions 1 and 2 hold. Set $k = k_{\mathrm{SE}}$ or $k = k_{\mathrm{Matérn}}$ with $\nu > 2$. Furthermore, assume that $d, \nu, \ell, r$, and $\sigma^2$ are fixed constants. Fix any $\delta_{\mathrm{GP}} \in (0, 1)$, and set the confidence width parameter $\beta_t$ of GP-UCB as defined in Eq. (6) with any fixed $\delta \in (0, 1 - \delta_{\mathrm{GP}})$. Then, with probability at least $1 - \delta_{\mathrm{GP}} - \delta$, the cumulative regret of GP-UCB (Algorithm 1) satisfies*

$$R_T = \begin{cases} \widetilde{O}\left(\sqrt{T}\right) & \text{if } k = k_{\mathrm{Matérn}} \text{ with } 2\nu + d \le \nu^2, \\ O\left(\sqrt{T \ln^2 T}\right) & \text{if } k = k_{\mathrm{SE}}. \end{cases} \tag{9}$$

*The hidden constants in the above expressions may depend on $\ln(1/\delta), d, \nu, \ell, r, \sigma^2$, and the constants $c_{\mathrm{sup}}, c_{\mathrm{gap}}, \rho_{\mathrm{quad}}, c_{\mathrm{quad}}$ corresponding with $\delta_{\mathrm{GP}}$, which are guaranteed to exist by Lemma 2.*

We would like to note the following three aspects of our results. First, the constants associated with the sample path properties defined in Lemma 2 are used solely for analyzing the regret. On the other hand, the existing algorithm provided by Scarlett [46], which shows the same $\widetilde{O}(\sqrt{T})$ regret as ours, requires prior information about these constants for the algorithm run. This is often unrealistic in practice. Secondly, our result does not imply the upper bound of Bayesian expected regret $\mathbb{E}[R_T]$. The main issue is that the dependence of the constants in Lemma 2 on $\delta_{\mathrm{GP}}$ is not explicitly known. We leave future work to break this limitation; however, note that the same limitation exists in the algorithm provided by Scarlett [46]. Thirdly, our results in Theorem 3 only focus on the dependence of the total step size $T$ in the regret. Therefore, we cannot claim any improvements of the regret on the dependence of the other parameters. For example, compared to the existing $R_T = O(\sqrt{T \ln^{d+2} T})$ regret under $k = k_{\mathrm{SE}}$, our regret upper bound $R_T = O(\sqrt{T \ln^2 T})$ indeed avoids the dependence of $d$ in the logarithmic factor; however, under the joint limit of $d$ and $T$ ($d, T \to \infty$), it easily behaves super-linearly even under the slowly increasing $d$ (e.g., $d = \Theta(\ln \ln T)$) due to the hidden constants in the regret.

### 3.1 Intuitive Explanation of our Analysis

Before we describe the proof, we provide an intuitive explanation of why GP-UCB achieves a tighter regret than the existing $O(\sqrt{\beta_T T \gamma_T(\mathcal{X})})$ upper bound. The motivation for our new analysis comes from the observation that the upper bound of the information gain: $I(\mathbf{X}_T) \le \gamma_T(\mathcal{X})$ in Eq. (8) is not always tight depending on the specific realization of the input sequence $\mathbf{X}_T$. To see this, let us observe the following two simple extreme cases of $\mathbf{X}_T$ where the inequality $I(\mathbf{X}_T) \le \gamma_T(\mathcal{X})$ is loose and tight:

- **Case I:** $I(\mathbf{X}_T) \le \gamma_T(\mathcal{X})$ **is loose**: Let us assume all the input is equal to the unique maximizer $\boldsymbol{x}^*$ (namely, $\forall t \in [T], \boldsymbol{x}_t = \boldsymbol{x}^*$). Then, when the kernel function satisfies $\forall \boldsymbol{x} \in \mathcal{X}, k(\boldsymbol{x}, \boldsymbol{x}) = 1$ as with $k_{\mathrm{SE}}$ and $k_{\mathrm{Matérn}}$, we have:

$$I(\mathbf{X}_T) = \frac{1}{2} \ln \det(\boldsymbol{I}_T + \sigma^{-2} \mathbf{K}(\mathbf{X}_T, \mathbf{X}_T)) = \frac{1}{2} \sum_{i=1}^T \ln(1 + \sigma^{-2}\lambda_i) = \frac{1}{2} \ln(1 + \sigma^{-2}T), \quad (10)$$

  where $\lambda_i$ is the $i$-th eigenvalue of $\mathbf{K}(\mathbf{X}_T, \mathbf{X}_T) = \mathbf{1}\mathbf{1}^\top$ with $\mathbf{1} = (1, \dots, 1)^\top \in \mathbb{R}^T$. The third equation uses the fact that $\mathbf{1}\mathbf{1}^\top$ is rank 1, and its unique non-zero eigenvalue is $T$.

- **Case II:** $I(\mathbf{X}_T) \le \gamma_T(\mathcal{X})$ **is tight**: Let us assume that $(\boldsymbol{x}_t)$ is the same as the input sequence generated by the maximum variance reduction (MVR) algorithm (namely, $\forall t \in [T], \boldsymbol{x}_t \in \operatorname{argmax}_{\boldsymbol{x} \in \mathcal{X}} \sigma(\boldsymbol{x}; \mathbf{X}_{t-1})$) [51, 56]. Then, from the discussion in Sections 2 and 5 in [51], we already know that $\gamma_T(\mathcal{X}) \le (1 - 1/e)^{-1} I(\mathbf{X}_T)$. This suggests that $I(\mathbf{X}_T) \le \gamma_T(\mathcal{X})$ is tight up to a constant factor when $\mathbf{X}_T$ is realized by MVR.

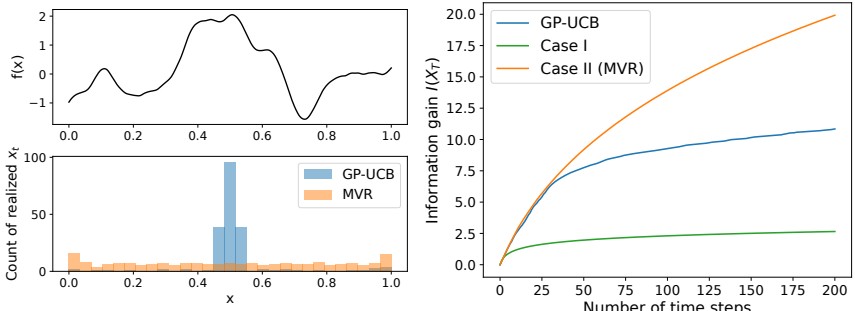

Figure 1: The behavior of the realized input sequence $\mathbf{X}_T$ (left) and the corresponding information gain $I(\mathbf{X}_T)$ (right) in the 1-dimensional BO problem with $\sigma^2 = 3$. The top left figure shows the objective function $f$ realized by GP under $k = k_{\text{Matérn}}$ with $\nu = 5/2$. The bottom left figure shows the histogram of the realized inputs: $(x_t)_{t \in [200]}$ with GP-UCB (blue) and MVR (orange) under $f$ in the top left figure. Furthermore, the right plot shows the corresponding information gain $I(\mathbf{X}_t)$ under GP-UCB or MVR. We also plot $I(\mathbf{X}_t) := 0.5 \ln(1 + \sigma^{-2}t)$, corresponding to Case I described in Section 3.1. We can observe that the inputs selected by GP-UCB are concentrated around the maximizer from the left figure. Then, from the right figure, we also observe that the corresponding information gain increases more slowly than that of MVR, and behaves similarly to Case I on $t \geq 30$. More comprehensive empirical results are also provided in Appendix D.

From Case I, we observe that $I(\mathbf{X}_T)$ satisfies $\Theta(\ln T) \leq I(\mathbf{X}_T) \leq \gamma_T(\mathcal{X})$ depending on $\mathbf{X}_T$. Furthermore, by comparing the input sequences in cases I and II, we expect that $I(\mathbf{X}_T)$ becomes small if $\mathbf{X}_T$ concentrates around the neighborhood of $x^*$, while $I(\mathbf{X}_T)$ becomes large if $\mathbf{X}_T$ spreads over the entire input domain $\mathcal{X}$. Then, from the fact that the worst-case regret of GP-UCB increases sub-linearly with the speed of $O(\sqrt{\beta_T T \gamma_T(\mathcal{X})})$, we can deduce that the input sequence $\mathbf{X}_T$ of GP-UCB will eventually concentrate around the maximizer $x^*$ if $x^*$ is unique and $\|f\|_\infty$ is not extremely small[2]. We provide an illustrative image in Figure 1. Our proof is designed so as to capture the above intuition that $I(\mathbf{X}_T)$ could be improved from $\gamma_T(\mathcal{X})$ to $\Theta(\ln T)$ under "favorable" sample path $f$.

### 3.2 Proof of Theorem 3

Let $\mathcal{A}$ be an event such that the two high-probability events of the original GP-UCB proof (described in the last paragraph in Section 2) and Lemma 2 with the confidence level $\delta_{\text{GP}}$ simultaneously hold. Note that event $\mathcal{A}$ occurs with probability at least $1 - \delta_{\text{GP}} - \delta$ from the union bound. Therefore, it is enough to prove our upper bound under $\mathcal{A}$. To encode the high-level idea in the previous section, we need to capture the concentration behavior of the input sequence $\mathbf{X}_T$ around the maximizer $x^*$. From this motivation, given some constant $\varepsilon > 0$, we decompose the regret as $R_T = R_T^{(1)}(\varepsilon) + R_T^{(2)}(\varepsilon)$, where:

$$R_T^{(1)}(\varepsilon) = \sum_{t \in \mathcal{T}(\varepsilon)} f(x^*) - f(x_t), \ R_T^{(2)}(\varepsilon) = \sum_{t \in \mathcal{T}^c(\varepsilon)} f(x^*) - f(x_t). \tag{11}$$

We set $\mathcal{T}(\varepsilon) = \{t \in [T] \mid f(x^*) - f(x_t) > \varepsilon\}$ and $\mathcal{T}^c(\varepsilon) = [T] \setminus \mathcal{T}(\varepsilon)$ in the above definition. A key observation is that, if we set sufficiently small $\varepsilon$ depending on the constants in Lemma 2, the inputs $(x_t)$ in $R_T^{(2)}(\varepsilon)$ (namely, inputs $(x_t)$ such that $f(x^*) - f(x_t) \leq \varepsilon$ holds) are on the locally quadratic region around the maximizer $x^*$ due to conditions 1 and 3 in Lemma 2. The formal descriptions are provided in Lemma 20 in Appendix C. This fact is originally leveraged in [46] to analyze the successive elimination-based algorithm. In the analysis of GP-UCB, it enables us to

---

[2]Specifically, if $T\|f\|_\infty \leq O(\sqrt{\beta_T T \gamma_T(\mathcal{X})})$, we cannot make any claims about $\mathbf{X}_T$ based on the worst-case bound since any sequence $\mathbf{X}_T$ satisfies the worst-case bound without concentrating around maximizer. This is why our analysis technique does not improve the worst-case regret in the frequentist setting. Indeed, in the proof of the worst-case lower bound for the frequentist setting [47], the existence of the function $f$ with $T\|f\|_\infty = O(\sqrt{\beta_T T \gamma_T(\mathcal{X})})$ is guaranteed.

analyze the behavior of the sub-input sequence $\{x_t \mid f(x^*) - f(x_t) \leq \varepsilon\}$ through the regularity constant $c_{\text{quad}}$. Below, we formally give the upper bound for $R_T^{(2)}(\varepsilon)$.

**Lemma 4** (General upper bound of $R_T^{(2)}$). *Suppose $(x_t)_{t \in [T]}$ is the input query sequence realized by the GP-UCB algorithm. Furthermore, let $\overline{\gamma}_t$ is the upper bound of MIG $\gamma_t(\mathcal{X})$ such that $\overline{\gamma}_t/t$ is non-increasing on $[\overline{T}, \infty)$ with some $\overline{T} \in \mathbb{N}_+$ [3]. Then, under event $\mathcal{A}$, we have*

$$R_T^{(2)}(\varepsilon) \leq 2c_{\text{sup}}\overline{T} + \frac{\pi^2}{3}\left(\log_2 T + 1\right) + \frac{2\sqrt{2C\beta_T T}}{\sqrt{2}-1} \max_{i \in [\bar{i}]} \sqrt{\gamma_{(T/2^{i-1})}\left(\mathcal{B}_2\left(\sqrt{c_{\text{quad}}^{-1}\eta_i}; x^*\right)\right)},$$

*where $C = 2/\ln(1 + \sigma^{-2})$, $\bar{i} = \lfloor \log_2 \frac{T}{T} \rfloor + 1$, $\eta_i = \frac{2\left(2\sqrt{C\beta_T(T/2^{i-1})\overline{\gamma}_{T/2^{i-1}}} + \frac{\pi^2}{6}\right)}{(T/2^{i-1})}$, and $\varepsilon = \min\{c_{\text{gap}}, c_{\text{quad}}\rho_{\text{quad}}^2\}$.*

We give the full proof in Appendix A.1. Here, the dominant term in the above lemma is given as:

$$R_T^{(2)}(\varepsilon) = \widetilde{O}\left(\max_i \sqrt{T\gamma_{(T/2^{i-1})}\left(\mathcal{B}_2\left(\sqrt{c_{\text{quad}}^{-1}\eta_i}; x^*\right)\right)}\right). \tag{12}$$

Note that $\eta_i$ is decreasing as the time index $T/2^{i-1}$ of MIG increases. In other words, the input domain $\mathcal{B}_2\left(\sqrt{c_{\text{quad}}^{-1}\eta_i}; x^*\right)$ of MIG shrinks as the time index $T/2^{i-1}$ increases. This property is beneficial for obtaining a tighter upper bound than that from the existing technique. For example, under $k = k_{\text{Matérn}}$ with $2\nu + d \leq \nu^2$, we can confirm that the dominant polynomial term in MIG is canceled out by the shrinking of the input domain in MIG. Namely, we can obtain the following result under $k = k_{\text{Matérn}}$:

$$\max_i \gamma_{(T/2^{i-1})}\left(\mathcal{B}_2\left(\sqrt{c_{\text{quad}}^{-1}\eta_i}; x^*\right)\right) = \widetilde{O}(1) \;\; (\text{as } T \to \infty), \tag{13}$$

which leads to $R_T^{(2)}(\varepsilon) = \widetilde{O}(\sqrt{T})$. This strictly improves the trivial upper bound $R_T^{(2)}(\varepsilon) = \widetilde{O}(\sqrt{T\gamma_T(\mathcal{X})})$ under $k = k_{\text{Matérn}}$. The formal descriptions are given in the next lemma.

**Lemma 5** (Upper bound of $R_T^{(2)}$ under $k_{\text{SE}}$ and $k_{\text{Matérn}}$). *Suppose $(x_t)_{t \in [T]}$ is the input sequence realized by the GP-UCB algorithm. Furthermore, $\varepsilon$ is set as that in Lemma 4. Then, under event $\mathcal{A}$,*

$$R_T^{(2)}(\varepsilon) = \begin{cases} \widetilde{O}(\sqrt{T}) & \text{if } k = k_{\text{Matérn}} \text{ with } 2\nu + d \leq \nu^2, \\ O\left(\sqrt{T \ln^2 T}\right) & \text{if } k = k_{\text{SE}}. \end{cases} \tag{14}$$

The full proof is given in Appendix A.2. The remaining interest is the upper bound of $R_T^{(1)}(\varepsilon)$. The definition of $R_T^{(1)}(\varepsilon)$ is the same as the *lenient regret* [8], which is known to be smaller than the original regret $R_T$ in GP-UCB. Although Cai et al. [8] studies the frequentist setting, their proof strategy is also applicable to the Bayesian setting as described in Section 3.4 in [8]. The following lemma provides the formal statement about the upper bound of $R_T^{(1)}(\varepsilon)$.

**Lemma 6** (Upper bound of $R_T^{(1)}$, adaptation of the proof of Theorem 1 in [8]). *Fix any $\varepsilon > 0$. Suppose $k = k_{\text{SE}}$ or $k = k_{\text{Matérn}}$. Then, when running GP-UCB, $R_T^{(1)}(\varepsilon) = \widetilde{O}(1)$ holds under event $\mathcal{A}$.*

We provide the proof in Appendix A.3 for completeness. For both kernels, $R_T^{(1)}(\varepsilon)$ is dominated by the upper bound of $R_T^{(2)}(\varepsilon)$. Finally, we obtain the desired results by aggregating the inequalities in Lemmas 5 and 6.

## 4 Discussions

Below, we discuss the limitations of our results and outline possible directions for future research.

---

[3] Namely, $\forall t \geq \overline{T}, \forall \epsilon \geq 0, \overline{\gamma}_t/t \geq \overline{\gamma}_{t+\epsilon}/(t+\epsilon)$ and $\forall t \geq \overline{T}, \gamma_t(\mathcal{X}) \leq \overline{\gamma}_t$ hold for some $\overline{T} \in \mathbb{N}_+$.

- **Optimality.** Based on the $\Omega(\sqrt{T})$ lower bound on the expected regret provided by Scarlett [46], we conjecture that our $\widetilde{O}(\sqrt{T})$ high-probability regret bound for GP-UCB is near-optimal. However, it is not straightforward to extend the lower bound for the expected regret in [46] to a high probability result. Specifically, the lower bound in [46] is quantified by a mutual information term (Lemma 4 in [46]); however, to our knowledge, the technique used to handle this term appears to be specific to the expected regret setting. We believe that the rigorous optimality argument for the Bayesian high probability regret is an important direction for future research.

- **Smoothness condition.** In our result for the Matérn kernel, we require an additional smoothness constraint to obtain a $\widetilde{O}(\sqrt{T})$ regret bound[4] To overcome this issue in our proof, we believe that we need stronger regularity conditions on the sample path around the maximizer than those assumed in Lemma 2.

- **Extension to the expected regret.** Our regret bounds involve regularity constants that depend on the sample path. However, to our knowledge, there is no existing research that rigorously analyzes how these constants depend on the confidence level $\delta_{GP}$. This makes it difficult to obtain the expected regret guarantees as with the original GP-UCB, whose expected regret bounds are established by properly decreasing the confidence level as a function of $T$ (e.g., [40, 53]). To overcome this issue, further analysis for Lemma 2, or another idea to quantify the sample path regularities, is required.

- **Extension to other algorithms.** One limitation of our technique is its restricted applicability to other algorithms. To apply our proof, at least the algorithm should satisfy the following two conditions: (i) on any index subset, the sub-linear cumulative regret is obtained with high probability (Lemma 21), and (ii) the high probability lenient regret bound is provided (Lemma 6). The existing analysis of the other major algorithms in the Bayesian setting (e.g., Thompson sampling [43], information directed sampling [44]) does not provide these properties. Nevertheless, we believe that the high-level ideas in our proof (see Section 3.1) could be beneficial for future refined analyses of other algorithms.

- **Instance dependent analysis in the frequentist setting.** As described in the footnote in Section 3.1, we believe that our analysis does not improve the worst-case regret upper bound in the frequentist setting. On the other hand, our technique can be applied to the instance-dependent analysis [49] for GP-UCB. We expect that our proof strategy could yield a $\widetilde{O}(\sqrt{T})$ instance-dependent regret for GP-UCB by replacing the sample path condition 3 in Lemma 2 with the *growth condition* (Definition 4 in [49]) of the function. It is an interesting direction for future research.

## 5 Conclusion

We provide a refined analysis of GP-UCB in the BO problem. For both SE and Matérn kernels, our results improve upon existing regret guarantees and fill the gap between the existing regret of GP-UCB and the current best upper bound in [46]. The core idea of our analysis is to capture the shrinking behavior of the input sequence by relating it to the worst-case upper bound and the sample path regularity conditions. Although our current analysis is limited to GP-UCB in the Bayesian setting, we believe it lays the foundation for several promising future research directions.

## Acknowledgments

We thank Jonathan Scarlett and Shion Takeno for their valuable comments on revising the manuscript.

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

# A  Proofs in Section 3

## A.1  Proof of Lemma 4

*Proof.* From Lemma 21, we have the following upper bound for any index set $\mathcal{T} \subset [T]$ under $\mathcal{A}$:

$$\sum_{t \in \mathcal{T}} f(x^*) - f(x_t) \le 2\sqrt{C\beta_T |\mathcal{T}| \overline{\gamma}_{|\mathcal{T}|}} + \frac{\pi^2}{6}. \tag{15}$$

Here, for any $i$ such that $T/2^{i-1} \ge \overline{T}$, we set $(\eta_i)$ as

$$\eta_i = \frac{2\left(2\sqrt{C\beta_T (T/2^{i-1}) \overline{\gamma}_{T/2^{i-1}}} + \frac{\pi^2}{6}\right)}{(T/2^{i-1})}. \tag{16}$$

As described in the proof below, these $(\eta_i)$ are designed so that we can obtain the upper bound of $|\mathcal{T}(\eta_i)|$ in a dyadic manner. Here, we consider the upper bound of $|\mathcal{T}(\eta_i)|$ based on the worst-case upper bound in Eq. (15). From the definition of $\mathcal{T}(\eta)$ and Eq. (15) with $\mathcal{T} = [T]$, the condition $|\mathcal{T}(\eta_1)|\eta_1 \le 2\sqrt{C\beta_T T \overline{\gamma}_T} + \pi^2/6$ must be satisfied; otherwise, we have $\sum_{t \in [T]} f(x^*) - f(x_t) \ge \sum_{t \in \mathcal{T}(\eta_1)} f(x^*) - f(x_t) \ge |\mathcal{T}(\eta_1)|\eta_1 > 2\sqrt{C\beta_T T \overline{\gamma}_T} + \pi^2/6$, which contradicts worst-case upper bound in Eq. (15). Therefore, we can obtain the following upper bound:

$$|\mathcal{T}(\eta_1)| \le \max\left\{t \le T \mid t\eta_1 \le 2\sqrt{C\beta_T T \overline{\gamma}_T} + \frac{\pi^2}{6}\right\} = \frac{T}{2}. \tag{17}$$

Furthermore, since $\eta_i$ is monotonic due to the condition about $\overline{\gamma}_t$, we have $\eta_1 \le \eta_2$, which implies $\mathcal{T}(\eta_2) \subset \mathcal{T}(\eta_1)$. From Eq. (15) with $\mathcal{T} = \mathcal{T}(\eta_1)$, Eq. (17), and $\mathcal{T}(\eta_2) \subset \mathcal{T}(\eta_1)$, we further obtain

$$|\mathcal{T}(\eta_2)| \le \max\left\{t \le T/2 \mid t\eta_2 \le 2\sqrt{C\beta_T (T/2) \overline{\gamma}_{(T/2)}} + \frac{\pi^2}{6}\right\} = \frac{T}{4}. \tag{18}$$

Similarly to $|\mathcal{T}(\eta_2)|$, we have $\mathcal{T}(\eta_3) \subset \mathcal{T}(\eta_2)$ and

$$|\mathcal{T}(\eta_3)| \le \max\left\{t \le T/4 \mid t\eta_3 \le 2\sqrt{C\beta_T (T/4) \overline{\gamma}_{(T/4)}} + \frac{\pi^2}{6}\right\} = \frac{T}{8}. \tag{19}$$

By repeating this argument $i$ times while $T/2^{i-1} \ge \overline{T}$ holds, we have the following inequality for any $i \le \lfloor \log_2 \frac{T}{\overline{T}} \rfloor + 1$:

$$|\mathcal{T}(\eta_i)| \le \max\left\{t \le T/2^{i-1} \mid t\eta_i \le \sqrt{C\beta_T (T/2^{i-1}) \overline{\gamma}_{(T/2^{i-1})}} + \frac{\pi^2}{6}\right\} = \frac{T}{2^i}. \tag{20}$$

Then, we have

$$R_T^{(2)}(\varepsilon) = \sum_{t \in \mathcal{T}^c(\varepsilon)} f(x^*) - f(x_t) \tag{21}$$

$$= \sum_{t \in \mathcal{T}^c(\varepsilon) \cap \mathcal{T}(\eta_1)} f(x^*) - f(x_t) + \sum_{t \in \mathcal{T}^c(\varepsilon) \cap \mathcal{T}^c(\eta_1)} f(x^*) - f(x_t) \tag{22}$$

$$= \sum_{t \in \mathcal{T}^c(\varepsilon) \cap \mathcal{T}(\eta_1) \cap \mathcal{T}(\eta_2)} f(x^*) - f(x_t) + \sum_{t \in \mathcal{T}^c(\varepsilon) \cap \mathcal{T}(\eta_1) \cap \mathcal{T}^c(\eta_2)} f(x^*) - f(x_t)$$
$$+ \sum_{t \in \mathcal{T}^c(\varepsilon) \cap \mathcal{T}^c(\eta_1)} f(x^*) - f(x_t) \tag{23}$$

$$= \sum_{t \in \mathcal{T}^c(\varepsilon) \cap \mathcal{T}(\eta_2)} f(x^*) - f(x_t) + \sum_{i=1}^{2} \sum_{t \in \mathcal{T}^c(\varepsilon) \cap \mathcal{T}(\eta_{i-1}) \cap \mathcal{T}^c(\eta_i)} f(x^*) - f(x_t), \tag{24}$$

where the last line follows from $\mathcal{T}(\eta_2) \subset \mathcal{T}(\eta_1)$. In the above inequality, we define $\mathcal{T}(\eta_0)$ as $\mathcal{T}(\eta_0) = [T]$ for notational convenience. By repeatedly applying the above decomposition, we

obtain

$$\sum_{t\in\mathcal{T}^c(\varepsilon)\cap\mathcal{T}(\eta_2)} f(\boldsymbol{x}^*) - f(\boldsymbol{x}_t) + \sum_{i=1}^{2}\sum_{t\in\mathcal{T}^c(\varepsilon)\cap\mathcal{T}(\eta_{i-1})\cap\mathcal{T}^c(\eta_i)} f(\boldsymbol{x}^*) - f(\boldsymbol{x}_t) \tag{25}$$

$$= \sum_{t\in\mathcal{T}^c(\varepsilon)\cap\mathcal{T}(\eta_3)} f(\boldsymbol{x}^*) - f(\boldsymbol{x}_t) + \sum_{i=1}^{3}\sum_{t\in\mathcal{T}^c(\varepsilon)\cap\mathcal{T}(\eta_{i-1})\cap\mathcal{T}^c(\eta_i)} f(\boldsymbol{x}^*) - f(\boldsymbol{x}_t) \tag{26}$$

$$\vdots$$

$$= \sum_{t\in\mathcal{T}^c(\varepsilon)\cap\mathcal{T}(\eta_{\bar{i}})} f(\boldsymbol{x}^*) - f(\boldsymbol{x}_t) + \sum_{i=1}^{\bar{i}}\sum_{t\in\mathcal{T}^c(\varepsilon)\cap\mathcal{T}(\eta_{i-1})\cap\mathcal{T}^c(\eta_i)} f(\boldsymbol{x}^*) - f(\boldsymbol{x}_t), \tag{27}$$

where $\bar{i} = \lfloor \log_2 \frac{T}{\overline{T}} \rfloor + 1$. Regarding the first term in Eq. (27), we have

$$\sum_{t\in\mathcal{T}^c(\varepsilon)\cap\mathcal{T}(\eta_{\bar{i}})} f(\boldsymbol{x}^*) - f(\boldsymbol{x}_t) \le 2c_{\sup}|\mathcal{T}(\eta_{\bar{i}})| \le 2c_{\sup}\overline{T}, \tag{28}$$

where the last inequality follows from $|\mathcal{T}(\eta_{\bar{i}})| \le \overline{T}$, which is implied by $|\mathcal{T}(\eta_{\bar{i}})| \le T/2^{\bar{i}}$ from Eq. (20) and the definition of $\bar{i}$. Next, regarding the second term in Eq. (27), we first define $\mathcal{T}_i$ and $\mathcal{X}^{(i)}$ as $\mathcal{T}_i = \mathcal{T}^c(\varepsilon) \cap \mathcal{T}(\eta_{i-1}) \cap \mathcal{T}^c(\eta_i)$ and $\mathcal{X}^{(i)} = \{\boldsymbol{x}_t \mid t \in \mathcal{T}_i\}$, respectively. Then, by applying Lemma 21 with $\mathcal{T} = \mathcal{T}_i$, we have

$$\sum_{t\in\mathcal{T}^c(\varepsilon)\cap\mathcal{T}(\eta_{i-1})\cap\mathcal{T}^c(\eta_i)} f(\boldsymbol{x}^*) - f(\boldsymbol{x}_t) = \sum_{t\in\mathcal{T}_i} f(\boldsymbol{x}^*) - f(\boldsymbol{x}_t) \tag{29}$$

$$\le 2\sqrt{C\beta_T|\mathcal{T}_i|I(\mathcal{X}^{(i)})} + \frac{\pi^2}{6} \tag{30}$$

$$\le 2\sqrt{C\beta_T|\mathcal{T}_i|\gamma_{|\mathcal{T}_i|}(\mathcal{X}^{(i)})} + \frac{\pi^2}{6} \tag{31}$$

$$\le 2\sqrt{C\beta_T|\mathcal{T}(\eta_{i-1})|\gamma_{|\mathcal{T}(\eta_{i-1})|}(\mathcal{X}^{(i)})} + \frac{\pi^2}{6} \tag{32}$$

$$\le 2\sqrt{C\beta_T(T/2^{i-1})\gamma_{(T/2^{i-1})}(\mathcal{X}^{(i)})} + \frac{\pi^2}{6}, \tag{33}$$

where the third inequality follows from $|\mathcal{T}_i| \le |\mathcal{T}(\eta_{i-1})|$, and the last inequality follows from Eq. (20). By aggregating Eqs. (27), (28), and (33), we obtain the following inequality under $\mathcal{A}$:

$$R_T^{(2)}(\varepsilon) \le 2c_{\sup}\overline{T} + 2\sum_{i=1}^{\bar{i}}\left[\sqrt{C\beta_T(T/2^{i-1})\gamma_{(T/2^{i-1})}(\mathcal{X}^{(i)})} + \frac{\pi^2}{6}\right] \tag{34}$$

$$\le 2c_{\sup}\overline{T} + \frac{\pi^2}{3}\left(\log_2 T + 1\right) + 2\sqrt{C\beta_T T}\sum_{i=1}^{\bar{i}}\frac{1}{2^{(i-1)/2}}\sqrt{\gamma_{(T/2^{i-1})}(\mathcal{X}^{(i)})} \tag{35}$$

$$\le 2c_{\sup}\overline{T} + \frac{\pi^2}{3}\left(\log_2 T + 1\right) + \frac{2\sqrt{2C\beta_T T}}{\sqrt{2}-1}\max_{i\in[\bar{i}]}\sqrt{\gamma_{(T/2^{i-1})}(\mathcal{X}^{(i)})}. \tag{36}$$

The last line follows from $\sum_{i=1}^{\bar{i}}\frac{1}{2^{(i-1)/2}} \le \sum_{i=1}^{\infty}\frac{1}{2^{(i-1)/2}} = \frac{1}{1-1/\sqrt{2}} = \frac{\sqrt{2}}{\sqrt{2}-1}$. The last part of the proof is to specify the radius of the ball $\mathcal{B}_2(\cdot;\boldsymbol{x}^*)$ such that $\mathcal{X}^{(i)}$ is included in it.

**Conversion of the sub-optimality gap into the upper bound input radius.** From condition 3 in Lemma 2, the definition of $\mathcal{T}^c(\varepsilon)$, $\varepsilon$, and Lemma 20, we have $\boldsymbol{x} \in \mathcal{B}_2(\rho_{\text{quad}};\boldsymbol{x}^*)$ for any $\boldsymbol{x} \in \mathcal{X}^{(i)}$. This implies $\forall \boldsymbol{x} \in \mathcal{X}^{(i)}, f(\boldsymbol{x}^*) - f(\boldsymbol{x}) \ge c_{\text{quad}}\|\boldsymbol{x}-\boldsymbol{x}^*\|_2^2$ from condition 3 in Lemma 2. Since $\forall \boldsymbol{x} \in \mathcal{X}^{(i)}, f(\boldsymbol{x}^*) - f(\boldsymbol{x}) \le \eta_i$ from $\mathcal{T}_i \subset \mathcal{T}^c(\eta_i)$, we have $\eta_i \ge c_{\text{quad}}\|\boldsymbol{x}-\boldsymbol{x}^*\|_2^2 \Leftrightarrow \sqrt{c_{\text{quad}}^{-1}\eta_i} \ge \|\boldsymbol{x}-\boldsymbol{x}^*\|_2$, which implies $\mathcal{X}^{(i)} \subset \mathcal{B}_2(\sqrt{c_{\text{quad}}^{-1}\eta_i};\boldsymbol{x}^*)$. Therefore, we have

$$\gamma_{(T/2^{i-1})}(\mathcal{X}^{(i)}) \le \gamma_{(T/2^{i-1})}\left(\mathcal{B}_2\left(\sqrt{c_{\text{quad}}^{-1}\eta_i};\boldsymbol{x}^*\right)\right). \tag{37}$$

Finally, combining Eq. (35) with Eq. (37), we have

$$R_T^{(2)}(\varepsilon) \le 2c_{\sup}\overline{T} + \frac{\pi^2}{3}\left(\log_2 T + 1\right) + \frac{2\sqrt{2C\beta_T T}}{\sqrt{2}-1} \max_{i\in[\bar{i}]} \sqrt{\gamma_{(T/2^{i-1})}\left(\mathcal{B}_2\left(\sqrt{c_{\text{quad}}^{-1}\eta_i}; x^*\right)\right)}. \quad (38)$$

□

## A.2 Proof of Lemma 5

To prove Lemma 5, we require the upper bound of MIG with the explicit dependence on the radius of the input domain. In Corollary 8 in Appendix B, we provide it with a full proof. Below, we establish the proof of Lemma 5 based on Corollary 8.

**When $k = k_{\text{Matérn}}$.** Set $C_{\text{Mat}} > 0$ as the constant such that the following inequalities hold:

$$\forall t \ge 2, \gamma_t(\mathcal{X}) \le C_{\text{Mat}} t^{\frac{d}{2\nu+d}} \ln^{\frac{4\nu+d}{2\nu+d}} t, \quad (39)$$

$$\forall t \ge 2, \forall \eta > 0, \gamma_t\left(\{x \in \mathbb{R}^d \mid \|x\|_2 \le \eta\}\right) \le C_{\text{Mat}}\left(\eta^{\frac{2\nu d}{2\nu+d}} t^{\frac{d}{2\nu+d}} \ln^{\frac{4\nu+d}{2\nu+d}} t + \ln^2 t\right). \quad (40)$$

The existence of $C_{\text{Mat}}$ is guaranteed by the upper bound of MIG established in Corollary 8[5]. Note that $C_{\text{Mat}}$ is the constant that may depend on $d, \ell, \nu, r$, and $\sigma^2$. Furthermore, we set $\overline{\gamma}_t = C_{\text{Mat}} t^{\frac{d}{2\nu+d}} \ln^{\frac{4\nu+d}{2\nu+d}} t$. For function $g(t) := \overline{\gamma}_t/t$, we have

$$g'(t) = -\frac{2\nu C_{\text{Mat}}}{2\nu+d} t^{-\frac{2\nu}{2\nu+d}-1} \ln^{\frac{4\nu+d}{2\nu+d}} t + C_{\text{Mat}} \frac{4\nu+d}{2\nu+d} t^{-\frac{2\nu}{2\nu+d}-1} \ln^{\frac{2\nu}{2\nu+d}} t \quad (41)$$

$$= \frac{C_{\text{Mat}}}{2\nu+d} t^{-\frac{2\nu}{2\nu+d}-1} (\ln^{\frac{2\nu}{2\nu+d}} t)(-2\nu \ln t + 4\nu + d). \quad (42)$$

From the above expression, if $2\nu \ln t \ge 4\nu + d \Leftrightarrow t \ge \exp(2 + d/(2\nu))$, $\overline{\gamma}_t/t$ is non-increasing. Therefore, we set $\overline{T} = \lceil \exp(2 + d/(2\nu)) \rceil$, which is independent of $T$. Here, for any $\eta > 0$ and $t \ge 2$, we have

$$\gamma_t\left(\mathcal{B}_2\left(\eta; x^*\right)\right) \le \gamma_t\left(\{x \in \mathbb{R}^d \mid \|x - x^*\|_2 \le \eta\}\right) \quad (43)$$

$$= \gamma_t\left(\{x \in \mathbb{R}^d \mid \|x\|_2 \le \eta\}\right) \quad (44)$$

$$\le C_{\text{Mat}}\left(\eta^{\frac{2\nu d}{2\nu+d}} t^{\frac{d}{2\nu+d}} \ln^{\frac{4\nu+d}{2\nu+d}} t + \ln^2 t\right), \quad (45)$$

where the second line follows from the fact that $k_{\text{Matérn}}$ is the stationary kernel (namely, $k_{\text{Matérn}}$ is transition invariant against any shift of inputs). Regarding $\eta_i$ in Lemma 4, by setting $T_i$ as $T_i = T/2^{i-1}$, we have

$$\eta_i = \frac{2\left(2\sqrt{C\beta_T T_i \overline{\gamma}_{T_i}} + \frac{\pi^2}{6}\right)}{T_i} \quad (46)$$

$$= \frac{4\sqrt{C\beta_T T_i \overline{\gamma}_{T_i}}}{T_i} + \frac{\pi^2}{3T_i} \quad (47)$$

$$= \frac{4\sqrt{C\beta_T T_i C_{\text{Mat}} T_i^{\frac{d}{2\nu+d}} \ln^{\frac{4\nu+d}{2\nu+d}} T_i}}{T_i} + \frac{\pi^2}{3T_i} \quad (48)$$

$$= 4\sqrt{CC_{\text{Mat}}\beta_T}\left(T_i^{-\frac{\nu}{2\nu+d}} \ln^{\frac{4\nu+d}{4\nu+2d}} T_i\right) + \frac{\pi^2}{3T_i} \quad (49)$$

$$\le \widetilde{C}_{\text{Mat}}\sqrt{\beta_T}\left(T_i^{-\frac{\nu}{2\nu+d}} \ln^{\frac{4\nu+d}{4\nu+2d}} T_i\right), \quad (50)$$

---

[5]If we rely on the result in [58], we can tighten the logarithmic term from $\ln^{\frac{4\nu+d}{2\nu+d}} t$ to $\ln^{\frac{2\nu}{2\nu+d}} t$; however, due to the technical issue of [58] described in Appendix B, we proceed our proof based on Corollary 8.

where $\widetilde{C}_{\mathrm{Mat}} > 0$ is a sufficiently large constant such that $\widetilde{C}_{\mathrm{Mat}}\sqrt{\beta_T}\left(T_i^{-\frac{\nu}{2\nu+d}}\ln^{\frac{4\nu+d}{4\nu+2d}}T_i\right) \geq$ $4\sqrt{CC_{\mathrm{Mat}}\beta_T}\left(T_i^{-\frac{\nu}{2\nu+d}}\ln^{\frac{4\nu+d}{4\nu+2d}}T_i\right) + \frac{\pi^2}{3T_i}$ for any $T_i \geq 2$. Note that we can choose $\widetilde{C}_{\mathrm{Mat}} > 0$ without depending on $T$. From Eqs. (45) and (50), for any $i$, we have

$$\gamma_{T/2^{i-1}}\left(\mathcal{B}_2\left(\sqrt{c_{\mathrm{quad}}^{-1}\eta_i};x^*\right)\right) \tag{51}$$

$$\leq C_{\mathrm{Mat}}\left(c_{\mathrm{quad}}^{-\frac{\nu d}{2\nu+d}}\eta_i^{\frac{\nu d}{2\nu+d}}T_i^{\frac{d}{2\nu+d}}\ln^{\frac{4\nu+d}{2\nu+d}}T_i + \ln^2 T_i\right) \tag{52}$$

$$\leq C_{\mathrm{Mat}}\left[c_{\mathrm{quad}}^{-\frac{\nu d}{2\nu+d}}\widetilde{C}_{\mathrm{Mat}}^{\frac{\nu d}{2\nu+d}}\beta_T^{\frac{\nu d}{2(2\nu+d)}}\left(T_i^{-\frac{\nu}{2\nu+d}}\ln^{\frac{4\nu+d}{4\nu+2d}}T_i\right)^{\frac{\nu d}{2\nu+d}}T_i^{\frac{d}{2\nu+d}}\ln^{\frac{4\nu+d}{2\nu+d}}T_i + \ln^2 T\right]. \tag{53}$$

Furthermore, by noting condition $2\nu + d \leq \nu^2$, we have

$$\left(T_i^{-\frac{\nu}{2\nu+d}}\ln^{\frac{4\nu+d}{4\nu+2d}}T_i\right)^{\frac{\nu d}{2\nu+d}}T_i^{\frac{d}{2\nu+d}}\ln^{\frac{4\nu+d}{2\nu+d}}T_i = \widetilde{O}\left(T_i^{-\frac{\nu^2 d}{(2\nu+d)^2}+\frac{d}{2\nu+d}}\right) \tag{54}$$

$$= \widetilde{O}\left(T_i^{\frac{d(2\nu+d)-\nu^2 d}{(2\nu+d)^2}}\right) \tag{55}$$

$$= \widetilde{O}\left(T_i^{\frac{d(2\nu+d-\nu^2)}{(2\nu+d)^2}}\right) \tag{56}$$

$$= \widetilde{O}(1). \tag{57}$$

From the above inequalities, we have $\gamma_{T/2^{i-1}}\left(\mathcal{B}_2\left(\sqrt{c_{\mathrm{quad}}^{-1}\eta_i};x^*\right)\right) = \widetilde{O}(1)$. Therefore, Lemma 4 implies

$$R_T^{(2)}(\varepsilon) \leq 2c_{\mathrm{sup}}\overline{T} + \frac{\pi^2}{3}\left(\log_2 T + 1\right) + \frac{2\sqrt{2C\beta_T T}}{\sqrt{2}-1}\times\widetilde{O}(1) \tag{58}$$

$$= \widetilde{O}(\sqrt{T}). \tag{59}$$

**When $k = k_{\mathrm{SE}}$.** The proof for $k = k_{\mathrm{SE}}$ is not as straightforward as the proof for $k = k_{\mathrm{Matérn}}$. Specifically, we have to choose a proper $\overline{T}$ so as to obtain an $O(\ln T)$ upper bound of MIG. Let $C_{\mathrm{SE}} > 0$ be the constant such that the following inequalities hold:

$$\forall t \geq 2, \gamma_t(\mathcal{X}) \leq C_{\mathrm{SE}}\ln^{d+1}t, \tag{60}$$

$$\forall t \geq 2, \forall \eta \in \left(0, \sqrt{\frac{2\ell^2}{e^2 c_d}}\right), \gamma_t(\{x \in \mathbb{R}^d \mid \|x\|_2 \leq \eta\}) \leq C_{\mathrm{SE}}\left(\frac{\ln^{d+1}t}{\ln^d\left(\frac{2\ell^2}{\eta^2 e c_d}\right)} + \ln T\right). \tag{61}$$

The existence of such $C_{\mathrm{SE}}$ is guaranteed by Corollary 8. In the above inequalities, $c_d$ is the constant defined in Corollary 8. We also set $\overline{\gamma}_t$ as $\overline{\gamma}_t = C_{\mathrm{SE}}\ln^{d+1}t$. We choose $\overline{T}$ later such that we can leverage the second statement in the above inequalities. Under $k = k_{\mathrm{SE}}$, we have

$$\eta_i = \frac{2\left(2\sqrt{C\beta_T T_i\overline{\gamma}_{T_i}} + \frac{\pi^2}{6}\right)}{T_i} \tag{62}$$

$$= \frac{4\sqrt{C\beta_T T_i\overline{\gamma}_{T_i}}}{T_i} + \frac{\pi^2}{3T_i} \tag{63}$$

$$= \frac{4\sqrt{C\beta_T T_i C_{\mathrm{SE}}\ln^{d+1}T_i}}{T_i} + \frac{\pi^2}{3T_i} \tag{64}$$

$$= 4\sqrt{CC_{\mathrm{SE}}\beta_T}\left(T_i^{-\frac{1}{2}}\ln^{\frac{d+1}{2}}T_i\right) + \frac{\pi^2}{3T_i} \tag{65}$$

$$\leq \widetilde{C}_{\mathrm{SE}}\sqrt{\beta_T}\left(T_i^{-\frac{1}{2}}\ln^{\frac{d+1}{2}}T_i\right), \tag{66}$$

where $\widetilde{C}_{\mathrm{SE}} > 0$ is a sufficiently large constant such that $\widetilde{C}_{\mathrm{SE}}\sqrt{\beta_T}\left(T_i^{-\frac{1}{2}}\ln^{\frac{d+1}{2}}T_i\right) \geq 4\sqrt{CC_{\mathrm{SE}}\beta_T}\left(T_i^{-\frac{1}{2}}\ln^{\frac{d+1}{2}}T_i\right) + \frac{\pi^2}{3T_i}$ for any $T_i \geq 2$. Hereafter, we define $\overline{\eta}_i := \widetilde{C}_{\mathrm{SE}}\sqrt{\beta_T}\left(T_i^{-\frac{1}{2}}\ln^{\frac{d+1}{2}}T_i\right)$.
Then, to apply Eq. (61), we consider the lower bound of $T_i$ such that $\sqrt{c_{\mathrm{quad}}^{-1}\overline{\eta}_i} < \sqrt{2\ell^2/(e^2c_d)}$ hold.
From the condition $\sqrt{c_{\mathrm{quad}}^{-1}\overline{\eta}_i} < \sqrt{2\ell^2/(e^2c_d)}$, we have

$$\sqrt{c_{\mathrm{quad}}^{-1}\overline{\eta}_i} < \sqrt{\frac{2\ell^2}{e^2c_d}} \Leftrightarrow c_{\mathrm{quad}}^{-1}\frac{e^2c_d}{2\ell^2}\widetilde{C}_{\mathrm{SE}}\sqrt{\beta_T}\ln^{\frac{d+1}{2}}T_i < T_i^{\frac{1}{2}} \tag{67}$$

$$\Leftarrow c_{\mathrm{quad}}^{-1}\frac{e^2c_d}{2\ell^2}\widetilde{C}_{\mathrm{SE}}\sqrt{\beta_T}\ln^{\frac{d+1}{2}}T < T_i^{\frac{1}{2}} \tag{68}$$

$$\Leftrightarrow \left(\frac{e^2c_d\widetilde{C}_{\mathrm{SE}}}{2\ell^2c_{\mathrm{quad}}}\right)^2\beta_T\ln^{d+1}T < T_i. \tag{69}$$

From the above inequality, we set $\overline{T}$ such that

$$\left(\frac{e^2c_d\widetilde{C}_{\mathrm{SE}}}{2\ell^2c_{\mathrm{quad}}}\right)^2\beta_T\ln^{d+1}T < \overline{T}. \tag{70}$$

Then, from $T_i \geq \overline{T}$ and Eqs. (67), and (70),

$$\gamma_{T/2^{i-1}}\left(\mathcal{B}_2\left(\sqrt{c_{\mathrm{quad}}^{-1}\eta_i}; x^*\right)\right) \leq \gamma_{T_i}\left(\left\{x \in \mathbb{R}^d \mid \|x\|_2 \leq \sqrt{c_{\mathrm{quad}}^{-1}\eta_i}\right\}\right) \tag{71}$$

$$\leq C_{\mathrm{SE}}\left(\frac{\ln^{d+1}T_i}{\ln^d\left(\frac{2c_{\mathrm{quad}}\ell^2}{\overline{\eta}_iec_d}\right)} + \ln T\right). \tag{72}$$

Based on Eq. (72), we further consider the lower bound of $T_i$ such that

$$\frac{\ln^{d+1}T_i}{\ln^d\left(\frac{2c_{\mathrm{quad}}\ell^2}{\overline{\eta}_iec_d}\right)} = O(\ln T). \tag{73}$$

For the condition in Eq. (73), we have

$$\frac{2c_{\mathrm{quad}}\ell^2}{\overline{\eta}_iec_d} \geq T_i^{1/4} \Leftrightarrow \frac{2c_{\mathrm{quad}}\ell^2}{ec_d\widetilde{C}_{\mathrm{SE}}\sqrt{\beta_T}T_i^{-1/2}\ln^{\frac{d+1}{2}}T_i} \geq T_i^{1/4} \tag{74}$$

$$\Leftrightarrow T_i^{1/4} \geq \frac{ec_d\widetilde{C}_{\mathrm{SE}}\sqrt{\beta_T}\ln^{\frac{d+1}{2}}T_i}{2c_{\mathrm{quad}}\ell^2} \tag{75}$$

$$\Leftarrow T_i^{1/4} \geq \frac{ec_d\widetilde{C}_{\mathrm{SE}}\sqrt{\beta_T}\ln^{\frac{d+1}{2}}T}{2c_{\mathrm{quad}}\ell^2} \tag{76}$$

$$\Leftrightarrow T_i \geq \left(\frac{ec_d\widetilde{C}_{\mathrm{SE}}\sqrt{\beta_T}\ln^{\frac{d+1}{2}}T}{2c_{\mathrm{quad}}\ell^2}\right)^4. \tag{77}$$

Hence, if $\overline{T} \geq \left(\frac{ec_d\widetilde{C}_{\mathrm{SE}}\sqrt{\beta_T}\ln^{\frac{d+1}{2}}T}{2c_{\mathrm{quad}}\ell^2}\right)^4$, we have

$$C_{\mathrm{SE}}\left(\frac{\ln^{d+1}T_i}{\ln^d\left(\frac{2c_{\mathrm{quad}}\ell^2}{\overline{\eta}_iec_d}\right)} + \ln T\right) \leq C_{\mathrm{SE}}\left(\frac{\ln^{d+1}T_i}{4^{-d}\ln^dT_i} + \ln T\right) \tag{78}$$

$$\leq C_{\mathrm{SE}}\left(4^d\ln T + \ln T\right). \tag{79}$$

By aggregating the conditions (70) and (77), we set $\overline{T}$ as the smallest natural number such that the following inequalities hold:

$$\overline{T} \geq \left(\frac{e^2 c_d \widetilde{C}_{\mathrm{SE}}}{2\ell^2 c_{\mathrm{quad}}}\right)^2 \beta_T \ln^{d+1} T, \quad \text{and} \quad \overline{T} \geq \left(\frac{e c_d}{2 c_{\mathrm{quad}} \ell^2}\right)^4 \widetilde{C}_{\mathrm{SE}}^4 \beta_T^2 \ln^{2(d+1)} T. \tag{80}$$

Then, from Eqs. (72) and (79), we have

$$\sqrt{\gamma_{(T/2^{i-1})}\left(\mathcal{B}_2\left(\sqrt{c_{\mathrm{quad}}^{-1}\eta_i}; \boldsymbol{x}^*\right)\right)} = O(\sqrt{\ln T}). \tag{81}$$

Finally, by noting $\overline{T} = O(\ln^{2d+4} T)$, we obtain the following result from Lemma 4:

$$R_T^{(2)}(\varepsilon) = O\left(\ln^{2d+4} T + \sqrt{T \ln^2 T}\right). \tag{82}$$

Since $d$ is a fixed constant, the above equation implies $R_T^{(2)}(\varepsilon) = O(\sqrt{T \ln^2 T})$. $\qquad\square$

### A.3 Proof of Lemma 6

*Proof.* From the upper bound of the discretization error in event $\mathcal{A}$, we have $\forall t \geq \sqrt{2/\varepsilon}, \forall \boldsymbol{x} \in \mathcal{X}, |f(\boldsymbol{x}) - f([\boldsymbol{x}]_t)| \leq \varepsilon/2$. Here, we set $\underline{\mathcal{T}}(\varepsilon) = \{t \in \mathbb{N}_+ \mid t \geq \sqrt{2/\varepsilon}\}$. By relying on the standard argument of MIG [51], we observe the following inequality for any realizations and $\varepsilon > 0$:

$$\min_{t \in \mathcal{T}(\varepsilon) \cap \underline{\mathcal{T}}(\varepsilon)} \sigma(\boldsymbol{x}_t; \mathbf{X}_{t-1}) \leq \sqrt{\frac{C\gamma_{|\mathcal{T}(\varepsilon) \cap \underline{\mathcal{T}}(\varepsilon)|}(\mathcal{X})}{|\mathcal{T}(\varepsilon) \cap \underline{\mathcal{T}}(\varepsilon)|}}, \tag{83}$$

where $\mathcal{T}(\varepsilon) = \{t \in [T] \mid f(\boldsymbol{x}^*) - f(\boldsymbol{x}_t) > \varepsilon\}$ and $C = 2/\ln(1 + \sigma^{-2})$. Under $\mathcal{A}$, we further have the following inequalities for any $\widetilde{t} \in \mathrm{argmin}_{t \in \mathcal{T}(\epsilon) \cap \underline{\mathcal{T}}(\varepsilon)} \sigma(\boldsymbol{x}_t; \mathbf{X}_{t-1})$[6]:

$$\mu(\boldsymbol{x}_{\widetilde{t}}; \mathbf{X}_{\widetilde{t}-1}; \boldsymbol{y}_{\widetilde{t}-1}) + \beta_{\widetilde{t}}^{1/2}\sigma(\boldsymbol{x}_{\widetilde{t}}; \mathbf{X}_{\widetilde{t}-1}) \tag{84}$$

$$= \mu(\boldsymbol{x}_{\widetilde{t}}; \mathbf{X}_{\widetilde{t}-1}; \boldsymbol{y}_{\widetilde{t}-1}) - \beta_{\widetilde{t}}^{1/2}\sigma(\boldsymbol{x}_{\widetilde{t}}; \mathbf{X}_{\widetilde{t}-1}) + 2\beta_{\widetilde{t}}^{1/2}\sigma(\boldsymbol{x}_{\widetilde{t}}; \mathbf{X}_{\widetilde{t}-1}) \tag{85}$$

$$\leq f(\boldsymbol{x}_{\widetilde{t}}) + 2\beta_{\widetilde{t}}^{1/2}\sigma(\boldsymbol{x}_{\widetilde{t}}; \mathbf{X}_{\widetilde{t}-1}) \tag{86}$$

$$< f(\boldsymbol{x}^*) - \varepsilon + 2\sqrt{\frac{C\beta_{\widetilde{t}}\gamma_{|\mathcal{T}(\epsilon) \cap \underline{\mathcal{T}}(\varepsilon)|}(\mathcal{X})}{|\mathcal{T}(\epsilon) \cap \underline{\mathcal{T}}(\varepsilon)|}} \tag{87}$$

$$\leq |f(\boldsymbol{x}^*) - f([\boldsymbol{x}^*]_{\widetilde{t}})| + f([\boldsymbol{x}^*]_{\widetilde{t}}) - \varepsilon + 2\sqrt{\frac{C\beta_{\widetilde{t}}\gamma_{|\mathcal{T}(\epsilon) \cap \underline{\mathcal{T}}(\varepsilon)|}(\mathcal{X})}{|\mathcal{T}(\epsilon) \cap \underline{\mathcal{T}}(\varepsilon)|}} \tag{88}$$

$$\leq \mu([\boldsymbol{x}^*]_{\widetilde{t}}; \mathbf{X}_{\widetilde{t}-1}; \boldsymbol{y}_{\widetilde{t}-1}) + \beta_{\widetilde{t}}^{1/2}\sigma([\boldsymbol{x}^*]_{\widetilde{t}}; \mathbf{X}_{\widetilde{t}-1}) - \frac{\varepsilon}{2} + 2\sqrt{\frac{C\beta_{\widetilde{t}}\gamma_{|\mathcal{T}(\epsilon) \cap \underline{\mathcal{T}}(\varepsilon)|}(\mathcal{X})}{|\mathcal{T}(\epsilon) \cap \underline{\mathcal{T}}(\varepsilon)|}}, \tag{89}$$

where the second inequality follows from the definition of $\mathcal{T}(\varepsilon)$, and the last inequality follows from $\widetilde{t} \in \underline{\mathcal{T}}(\varepsilon)$ and event $\mathcal{A}$. Therefore, under $\mathcal{A}$, the inequality $-\frac{\varepsilon}{2} + 2\sqrt{\frac{C\beta_{\widetilde{t}}\gamma_{|\mathcal{T}(\epsilon) \cap \underline{\mathcal{T}}(\varepsilon)|}(\mathcal{X})}{|\mathcal{T}(\epsilon) \cap \underline{\mathcal{T}}(\varepsilon)|}} \geq 0$ must hold; otherwise, $\mu(\boldsymbol{x}_{\widetilde{t}}; \mathbf{X}_{\widetilde{t}-1}; \boldsymbol{y}_{\widetilde{t}-1}) + \beta_{\widetilde{t}}^{1/2}\sigma(\boldsymbol{x}_{\widetilde{t}}; \mathbf{X}_{\widetilde{t}-1}) < \mu([\boldsymbol{x}^*]_{\widetilde{t}}; \mathbf{X}_{\widetilde{t}-1}; \boldsymbol{y}_{\widetilde{t}-1}) + \beta_{\widetilde{t}}^{1/2}\sigma([\boldsymbol{x}^*]_{\widetilde{t}}; \mathbf{X}_{\widetilde{t}-1})$, which contradicts $\boldsymbol{x}_{\widetilde{t}} \in \mathrm{argmax}_{\boldsymbol{x} \in \mathcal{X}} \mu(\boldsymbol{x}; \mathbf{X}_{\widetilde{t}-1}; \boldsymbol{y}_{\widetilde{t}-1}) + \beta_{\widetilde{t}}^{1/2}\sigma(\boldsymbol{x}; \mathbf{X}_{\widetilde{t}-1})$. This further implies

$$|\mathcal{T}(\epsilon) \cap \underline{\mathcal{T}}(\varepsilon)| \leq \frac{16C\beta_{\widetilde{t}}\gamma_{|\mathcal{T}(\epsilon) \cap \underline{\mathcal{T}}(\varepsilon)|}(\mathcal{X})}{\varepsilon^2} \leq \frac{16C\beta_T\gamma_{|\mathcal{T}(\epsilon) \cap \underline{\mathcal{T}}(\varepsilon)|}(\mathcal{X})}{\varepsilon^2} \tag{90}$$

---

[6]If $\mathcal{T}(\epsilon) \cap \underline{\mathcal{T}}(\varepsilon) = \emptyset$, the theorem's statement clearly holds; therefore, we suppose $\mathcal{T}(\epsilon) \cap \underline{\mathcal{T}}(\varepsilon) \neq \emptyset$ in this proof.

for any $\varepsilon > 0$. Furthermore,

$$R_T^{(1)}(\varepsilon) = \sum_{t \in \mathcal{T}(\epsilon)} f(\boldsymbol{x}^*) - f(\boldsymbol{x}_t) \tag{91}$$

$$= 2c_{\sup}\sqrt{\frac{2}{\varepsilon}} + \sum_{t \in \mathcal{T}(\epsilon) \cap \underline{\mathcal{T}}(\varepsilon)} f(\boldsymbol{x}^*) - f(\boldsymbol{x}_t) \tag{92}$$

$$\leq 2c_{\sup}\sqrt{\frac{2}{\varepsilon}} + \frac{\pi^2}{6} + 2\sqrt{C\beta_T|\mathcal{T}(\epsilon) \cap \underline{\mathcal{T}}(\varepsilon)|\gamma_{|\mathcal{T}(\epsilon) \cap \underline{\mathcal{T}}(\varepsilon)|}(\mathcal{X})} \tag{93}$$

for any $\varepsilon > 0$. In the above expressions, the last inequality follows from Lemma 21. The remaining part of the proof is to substitute the quantity $|\mathcal{T}(\epsilon) \cap \underline{\mathcal{T}}(\varepsilon)|$ in Eq. (93) into its upper bound, which is deduced from Eq. (90) depending on the kernel.

**For $k = k_{\mathrm{SE}}$.** Under $k = k_{\mathrm{SE}}$, we crudely take the upper bound of $|\mathcal{T}(\epsilon) \cap \underline{\mathcal{T}}(\varepsilon)|$ as

$$|\mathcal{T}(\epsilon) \cap \underline{\mathcal{T}}(\varepsilon)| \leq \frac{16C\beta_T\gamma_{|\mathcal{T}(\epsilon) \cap \underline{\mathcal{T}}(\varepsilon)|}(\mathcal{X})}{\varepsilon^2} \leq \frac{16C\beta_T\gamma_T(\mathcal{X})}{\varepsilon^2}. \tag{94}$$

The above upper bound implies $|\mathcal{T}(\epsilon) \cap \underline{\mathcal{T}}(\varepsilon)| = O(\beta_T\gamma_T(\mathcal{X}))$. Since $\gamma_T(\mathcal{X}) = O(\ln^{d+1} T)$ under $k = k_{\mathrm{SE}}$, Eq. (93) implies

$$R_T^{(1)}(\varepsilon) \leq 2c_{\sup}\sqrt{\frac{2}{\varepsilon}} + \frac{\pi^2}{6} + O\left(\sqrt{\beta_T(\beta_T\gamma_T(\mathcal{X}))\ln^{d+1}(\beta_T\gamma_T(\mathcal{X}))}\right) \tag{95}$$

$$= O\left(\beta_T\sqrt{(\ln^{d+1} T)\ln^{d+1}(\ln^{d+2} T)}\right) \tag{96}$$

$$= O\left(\sqrt{(\ln T)^{d+3}(\ln \ln T)^{d+1}}\right) \tag{97}$$

$$= \widetilde{O}(1). \tag{98}$$

**For $k = k_{\mathrm{Matérn}}$.** Set $C_{\mathrm{Mat}} > 0$ as the constant such that the following inequality holds:

$$\forall t \geq 2, \gamma_t(\mathcal{X}) \leq C_{\mathrm{Mat}} t^{\frac{d}{2\nu+d}} \ln^{\frac{4\nu+d}{2\nu+d}} t. \tag{99}$$

The existence of $C_{\mathrm{Mat}}$ is guaranteed by the upper bound of MIG established in Corollary 8. Then, if $|\mathcal{T}(\epsilon) \cap \underline{\mathcal{T}}(\varepsilon)| \geq 2$ holds, Eq. (90) implies

$$|\mathcal{T}(\epsilon) \cap \underline{\mathcal{T}}(\varepsilon)| \leq \frac{16C\beta_T C_{\mathrm{Mat}}|\mathcal{T}(\epsilon) \cap \underline{\mathcal{T}}(\varepsilon)|^{\frac{d}{2\nu+d}} \ln^{\frac{4\nu+d}{2\nu+d}} |\mathcal{T}(\epsilon) \cap \underline{\mathcal{T}}(\varepsilon)|}{\varepsilon^2} \tag{100}$$

$$\Rightarrow |\mathcal{T}(\epsilon) \cap \underline{\mathcal{T}}(\varepsilon)| \leq \frac{16C\beta_T C_{\mathrm{Mat}}|\mathcal{T}(\epsilon) \cap \underline{\mathcal{T}}(\varepsilon)|^{\frac{d}{2\nu+d}} \ln^{\frac{4\nu+d}{2\nu+d}} T}{\varepsilon^2} \tag{101}$$

$$\Leftrightarrow |\mathcal{T}(\epsilon) \cap \underline{\mathcal{T}}(\varepsilon)|^{\frac{2\nu}{2\nu+d}} \leq \frac{16C\beta_T C_{\mathrm{Mat}} \ln^{\frac{4\nu+d}{2\nu+d}} T}{\varepsilon^2} \tag{102}$$

$$\Leftrightarrow |\mathcal{T}(\epsilon) \cap \underline{\mathcal{T}}(\varepsilon)| \leq \left(\frac{16C\beta_T C_{\mathrm{Mat}} \ln^{\frac{4\nu+d}{2\nu+d}} T}{\varepsilon^2}\right)^{1+\frac{d}{2\nu}}. \tag{103}$$

Therefore, we have $|\mathcal{T}(\epsilon) \cap \underline{\mathcal{T}}(\varepsilon)| = \widetilde{O}(1)$ under fixed $\varepsilon$, $d$, and $\nu$. Hence, from Eq. (93), we obtain $R_T^{(1)}(\varepsilon) = \widetilde{O}(1)$. □

# B  Information Gain Upper Bound

Our analysis requires the upper bound of MIG with explicit dependence on the radius of the input domain. Several existing works [4, 27, 28] established such a result by extending the proof in [51]. However, the proof strategy in [51] result in $\widetilde{O}(T^{\frac{d(d+1)}{2\nu+d(d+1)}})$ upper bound of MIG in Matérn kernel,

which is strictly worse than the best achievable $\widetilde{O}(T^{\frac{d}{2\nu+d}})$ upper bound. Vakili et al. [58] shows $\widetilde{O}(T^{\frac{d}{2\nu+d}})$ upper bound of MIG with $\nu > 1/2$ under the uniform boundness assumption of the eigenfunctions. Furthermore, the following work [55] shows $\gamma_T(\{x \in \mathbb{R}^d \mid \|x\|_2 \leq \eta\}) = \widetilde{O}(\eta^{\frac{2\nu d}{2\nu+d}} T^{\frac{d}{2\nu+d}})$ for any radius $\eta > 0$ if there exist eigenfunctions uniformly bounded without depending on $\eta > 0$. Some of the related results supports the uniform boundness assumption under $d = 1$ [27, 62], or under the approximated version of the original Matérn kernel [42, 50]; however, to our knowledge, we are not aware of any literature that rigorously support uniform boundness assumption under the general compact input domain with $d \geq 2$ and $\nu > 1/2$. See Chapter 4.4 in [27] for the detailed discussion. Therefore, this section's goal is twofold: (i) prove $\widetilde{O}(T^{\frac{d}{2\nu+d}})$ upper bound as claimed in [58] without relying on the uniform boundness assumption, and (ii) clarify the explicit dependence on the input radius in the upper bound proved in (i).

Below, we formally describe our MIG upper bound.

**Theorem 7.** *Fix any $d \in \mathbb{N}_+$, $\sigma^2 > 0$, and $T \in \mathbb{N}_+$. Let us assume $X = \{x \in \mathbb{R}^d \mid \|x\|_2 \leq 1\}$. Then,*

- *For $k = k_{\mathrm{SE}}$, $\gamma_T(X)$ satisfies*

$$\gamma_T(X) \leq \frac{C_d^{(1)}}{\theta^d} \ln^{d+1}\left(1 + \frac{T}{\sigma^2}\right) + \ln\left(1 + \frac{T}{\sigma^2}\right) + C_d^{(2)} \exp\left(-\frac{2}{\theta} + \frac{1}{\theta^2}\right) \tag{104}$$

*if $\theta \leq e^2 c_d$ and $T/(e-1) \geq \sigma^2$. Furthermore, for any $\theta > e^2 c_d$, we have*

$$\gamma_T(X) \leq \frac{C_d^{(3)}}{\ln^d\left(\frac{\theta}{ec_d}\right)} \ln^{d+1}\left(1 + \frac{T}{\sigma^2}\right) + C_d^{(4)} \ln\left(1 + \frac{T}{\sigma^2}\right) + C_d^{(5)}. \tag{105}$$

*Here, we set $\theta = 2\ell^2$ and $c_d = \max\left\{1, \exp\left(\frac{1}{e}\left(\frac{d}{2} - 1\right)\right)\right\}$. Furthermore, $C_d^{(1)}, C_d^{(2)}, C_d^{(3)}, C_d^{(4)}, C_d^{(5)} > 0$ are the constants only depending on $d$.*

- *For $k = k_{\mathrm{Matérn}}$ with $\nu > 1/2$, $\gamma_T(X)$ satisfies*

$$\gamma_T(X) \leq C(T, \nu, \sigma^2)\overline{\gamma}_T + C \tag{106}$$

*where $C(T, \nu, \sigma^2) = \max\left\{1, \log_2\left(1 + \frac{\Gamma(\nu)}{C_\nu} \ln \frac{T^2}{\sigma^2}\right) + \frac{1}{\nu} \log_2\left(\frac{T^2}{\nu\Gamma(\nu)\sigma^2}\right) + 1\right\}$. Here, $C_\nu > 0$ and $C > 0$ are the constant that only depends on $\nu > 0$, and an absolute constant, respectively. Furthermore, $\overline{\gamma}_T$ is defined as*

$$\overline{\gamma}_T = C_{d,\nu}^{(1)} \ln\left(1 + \frac{2T}{\sigma^2}\right) + C_{d,\nu}^{(2)} \left(\frac{T}{\sigma^2 \ell^{2\nu}}\right)^{\frac{d}{2\nu+d}} \ln^{\frac{2\nu}{2\nu+d}}\left(1 + \frac{2T}{\sigma^2}\right), \tag{107}$$

*where $C_{d,\nu}^{(1)}, C_{d,\nu}^{(2)} > 0$ are the constants only depending on $d$ and $\nu$.*

We also obtain the following corollary by adjusting the lengthscale parameter $\ell > 0$ based on the radius of the input domain.

**Corollary 8.** *Fix any $d \in \mathbb{N}_+$, $\sigma^2 > 0$, $T \in \mathbb{N}_+$, $\eta > 0$. Let us assume $X = \{x \in \mathbb{R}^d \mid \|x\|_2 \leq \eta\}$. Then,*

- *For $k = k_{\mathrm{SE}}$, $\gamma_T(X)$ satisfies*

$$\gamma_T(X) \leq \frac{C_d^{(3)}}{\ln^d\left(\frac{2\ell^2}{\eta^2 ec_d}\right)} \ln^{d+1}\left(1 + \frac{T}{\sigma^2}\right) + C_d^{(4)} \ln\left(1 + \frac{T}{\sigma^2}\right) + C_d^{(5)}. \tag{108}$$

*if $2\ell^2/\eta^2 > e^2 c_d$.*

- *For $k = k_{\mathrm{Matérn}}$ with $\nu > 1/2$, $\gamma_T(X)$ satisfies Eq. (106), with*

$$\overline{\gamma}_T = C_{d,\nu}^{(1)} \ln\left(1 + \frac{2T}{\sigma^2}\right) + C_{d,\nu}^{(2)} \eta^{\frac{2\nu d}{2\nu+d}} \left(\frac{T}{\sigma^2 \ell^{2\nu}}\right)^{\frac{d}{2\nu+d}} \ln^{\frac{2\nu}{2\nu+d}}\left(1 + \frac{2T}{\sigma^2}\right). \tag{109}$$

*The constants in the above statements are the same as those in Theorem 7.*

While the above results ignore the explicit dependence on $d$ and $\nu$, all the other parameters are explicitly stated in our upper bound of MIG. We would like to emphasize that we do not rely on the uniform boundness assumption of the eigenfunctions to prove the above results. In the above results for $k = k_{\mathrm{SE}}$, we obtain the same $O(\ln^{d+1} T)$ upper bound as that in [58] except for the constant factor. For $k = k_{\mathrm{Matérn}}$, we also obtain the same $\widetilde{O}(T^{\frac{d}{2\nu+d}})$ upper bound as that in [58], while the logarithmic dependence get worse from $O\left(\ln^{d/(2\nu+d)} T\right)$ to $O\left(\ln^{(4\nu+d)/(2\nu+d)} T\right)$. Furthermore, the above result reveals the explicit dependence of the radius $\eta$ of the input domain. Regarding the case $k = k_{\mathrm{Matérn}}$, our result suggests $\widetilde{O}(\eta^{\frac{2\nu d}{2\nu+d}} T^{\frac{d}{2\nu+d}})$ upper bound of MIG, which is consistent with that in [55] with uniform boundness assumption.

**Proof overview.** The basic proof strategy follows that in [58], which leverages the Mercer decomposition of the kernel. To bypass the uniform boundness assumption in the proof of [58], we must resort to other specific properties of the eigenfunction. However, except for some exceptional cases, the eigenfunction of the kernel on the general compact domain is difficult to specify in an analytical form and complex to analyze. To avoid this issue, instead of studying the original definition of the MIG on $\mathbb{R}^d$, we consider reducing the original MIG on $\mathcal{X} := \{\boldsymbol{x} \in \mathbb{R}^d \mid \|\boldsymbol{x}\|_2 \leq 1\}$ to that on a hypersphere $\mathbb{S}^d := \{\boldsymbol{x} \in \mathbb{R}^{d+1} \mid \|\boldsymbol{x}\|_2 = 1\}$ defined in $\mathbb{R}^{d+1}$. The eigensystems on $\mathbb{S}^d$ are one of the exceptional cases, whose eigenfunctions are specified as a special function on $\mathbb{S}^d$, called *spherical harmonics* [1, 16]. Indeed, by using the addition theorem of the spherical harmonics (Theorem 14), the existing works [25, 31, 57] already demonstrated that the upper bound of MIG on $\mathbb{S}^d$ is proved as with [58] without the uniform boundness assumption. We use their technique to show the upper bound of MIG under SE and Matérn kernels on $\mathbb{R}^d$, while the original motivation of these existing works is to study the MIG under the neural tangent kernel on a hypersphere. The remaining parts of this section are constructed as follows:

- In Section B.1, we show our core result (Lemma 9) that guarantees the MIG on $\{\boldsymbol{x} \in \mathbb{R}^d \mid \|\boldsymbol{x}\|_2 \leq 1\}$ is bounded from above by that on $\mathbb{S}^d$ up to logarithmic factor.

- In Section B.2, we summarize the basic known results about Mercer decomposition on $\mathbb{S}^d$, which is the foundation of the following subsections.

- In Section B.3, we provide the general upper bound of the information gain on $\mathbb{S}^d$ (Lemma 15), represented by the kernel function's eigenvalues. This subsection's result has no intrinsic change from those in [31, 57]; however, we provide details for completeness.

- In Section B.4, we provide the upper bound of the decaying rate of the eigenvalues in SE and Matérn kernels.

- In Section B.5, we establish the full proof of Theorem 7 based on the results in Sections B.1–B.4.

## B.1 Reduction of the MIG on $\mathbb{R}^d$ to $\mathbb{S}^d$

**Lemma 9** (Reduction to the hypersphere in $\mathbb{R}^{d+1}$). *Fix any $d \in \mathbb{N}_+$, $\sigma^2 > 0$, and $T \in \mathbb{N}_+$. Suppose $\mathcal{X} = \{(x_1, \ldots, x_d, 0)^\top \in \mathbb{R}^{d+1} \mid \sum_{i=1}^d x_i^2 \leq 1\}$, and define $\mathbb{S}^d$ as $\mathbb{S}^d = \{\boldsymbol{x} \in \mathbb{R}^{d+1} \mid \|\boldsymbol{x}\|_2 = 1\}$. Then,*

- *For $k = k_{\mathrm{SE}}$, we have*

$$\max_{\boldsymbol{x}_1,\ldots,\boldsymbol{x}_T \in \mathcal{X}} \ln\det(\boldsymbol{I}_T + \sigma^{-2}\mathbf{K}(\mathbf{X}_T, \mathbf{X}_T)) \leq \max_{\boldsymbol{x}_1,\ldots,\boldsymbol{x}_T \in \mathbb{S}^d} \ln\det(\boldsymbol{I}_T + \sigma^{-2}\mathbf{K}(\mathbf{X}_T, \mathbf{X}_T)). \quad (110)$$

- *For $k = k_{\mathrm{Matérn}}$, we have*

$$\max_{\boldsymbol{x}_1,\ldots,\boldsymbol{x}_T \in \mathcal{X}} \ln\det(\boldsymbol{I}_T + \sigma^{-2}\mathbf{K}(\mathbf{X}_T, \mathbf{X}_T)) \quad (111)$$

$$\leq C(T, \nu, \sigma^2) \max_{\boldsymbol{x}_1,\ldots,\boldsymbol{x}_T \in \mathbb{S}^d} \ln\det(\boldsymbol{I}_T + 2\sigma^{-2}\mathbf{K}(\mathbf{X}_T, \mathbf{X}_T)) + C, \quad (112)$$

*where $C(T, \nu, \sigma^2) = \max\left\{0, \log_2\left(1 + \frac{\Gamma(\nu)}{C_\nu} \ln \frac{T^2}{\sigma^2}\right) + \frac{1}{\nu}\log_2\left(\frac{T^2}{\nu\Gamma(\nu)\sigma^2}\right) + 1\right\}$. Here, $C_\nu > 0$ is the constant that only depends on $\nu > 0$. Furthermore, $C > 0$ is an absolute constant.*

*Proof.* For any $\boldsymbol{x}_1, \ldots, \boldsymbol{x}_T \in \mathcal{X}$, we construct the new input sequence $\widetilde{\boldsymbol{x}}_1, \ldots, \widetilde{\boldsymbol{x}}_T$ on $\mathbb{S}^d$, where $\widetilde{\boldsymbol{x}}_i = \left(x_{i,1}, \ldots, x_{i,d}, \sqrt{1 - \sum_{j=1}^d x_{i,j}^2}\right)^\top$.

**Under $k = k_{\mathrm{SE}}$.** It is enough to show the following inequality:

$$\det(\boldsymbol{I}_T + \sigma^{-2}\mathbf{K}(\mathbf{X}_T, \mathbf{X}_T)) \le \det(\boldsymbol{I}_T + \sigma^{-2}\mathbf{K}(\widetilde{\mathbf{X}}_T, \widetilde{\mathbf{X}}_T)), \tag{113}$$

where $\widetilde{\mathbf{X}}_T = (\widetilde{\boldsymbol{x}}_1, \ldots, \widetilde{\boldsymbol{x}}_T)$. From the definition of $\widetilde{\boldsymbol{x}}_i$, we rewrite R.H.S. in the above inequality as

$$\det(\boldsymbol{I}_T + \sigma^{-2}\mathbf{K}(\widetilde{\mathbf{X}}_T, \widetilde{\mathbf{X}}_T)) = \det(\widetilde{\mathbf{K}} \odot (\boldsymbol{I}_T + \sigma^{-2}\mathbf{K}(\mathbf{X}_T, \mathbf{X}_T)), \tag{114}$$

where $[\widetilde{\mathbf{K}}]_{i,j} = k(\widetilde{\boldsymbol{x}}_i, \widetilde{\boldsymbol{x}}_j)/k(\boldsymbol{x}_i, \boldsymbol{x}_j)$. Here, $A \odot B$ denotes the Hadamard product of the matrices $A$ and $B$. Then, Oppenheim inequality (e.g., Theorem 7.27 in [63]) implies $\det(A \odot B) \ge \det(B) \prod_i A_{ii}$ for any positive semi-definite matrices $A$ and $B$. Therefore, if $\widetilde{\mathbf{K}}$ is a positive semi-definite matrix, Eq. (114) immediately implies

$$\det(\boldsymbol{I}_T + \sigma^{-2}\mathbf{K}(\widetilde{\mathbf{X}}_T, \widetilde{\mathbf{X}}_T)) \ge \det(\boldsymbol{I}_T + \sigma^{-2}\mathbf{K}(\mathbf{X}_T, \mathbf{X}_T)) \prod_{i \in [T]} \widetilde{\mathbf{K}}_{ii} = \det(\boldsymbol{I}_T + \sigma^{-2}\mathbf{K}(\mathbf{X}_T, \mathbf{X}_T)). \tag{115}$$

From the definition of $k_{\mathrm{SE}}$ and $\widetilde{\boldsymbol{x}}_i$, we have

$$\frac{k(\widetilde{\boldsymbol{x}}_i, \widetilde{\boldsymbol{x}}_j)}{k(\boldsymbol{x}_i, \boldsymbol{x}_j)} = \exp\left(-\frac{\|\boldsymbol{x}_i - \boldsymbol{x}_j\|_2^2 + \left(\sqrt{1 - \|\boldsymbol{x}_i\|_2^2} - \sqrt{1 - \|\boldsymbol{x}_j\|_2^2}\right)^2}{2\ell^2} + \frac{\|\boldsymbol{x}_i - \boldsymbol{x}_j\|_2^2}{2\ell^2}\right) \tag{116}$$

$$= \exp\left(-\frac{\left(\sqrt{1 - \|\boldsymbol{x}_i\|_2^2} - \sqrt{1 - \|\boldsymbol{x}_j\|_2^2}\right)^2}{2\ell^2}\right). \tag{117}$$

The above equation suggests that $\widetilde{\mathbf{K}}$ is equal to the kernel matrix of the one-dimensional SE-kernel, whose inputs are transformed by $\sqrt{1 - \|\cdot\|_2^2}$. Since the SE kernel is positive definite, the matrix $\widetilde{\mathbf{K}}$ is also positive semi-definite, and we complete the proof for $k = k_{\mathrm{SE}}$.

**Under $k = k_{\mathrm{Matérn}}$.** Similarly to the proof for $k = k_{\mathrm{SE}}$, we consider the application of Oppenheim inequality; however, the positive semi-definiteness of element-wise quotient matrix $\widetilde{\mathbf{K}}$ is unknown for $k = k_{\mathrm{Matérn}}$. To avoid this problem, we leverage the following representation of $k_{\mathrm{Matérn}}$, which is given as the form of the lengthscale mixture of the SE kernel [54]:

$$k(\boldsymbol{x}, \widetilde{\boldsymbol{x}}) = \frac{1}{\Gamma(\nu)} \int_0^\infty z^{\nu-1} e^{-z} \exp\left(-\frac{\|\boldsymbol{x} - \widetilde{\boldsymbol{x}}\|_2^2}{2\ell^2 z\nu^{-1}}\right) \mathrm{d}z. \tag{118}$$

Based on the above representation, we decompose the original kernel function $k$ into the following three components:

$$k(\boldsymbol{x}, \widetilde{\boldsymbol{x}}) = k_1(\boldsymbol{x}, \widetilde{\boldsymbol{x}}) + k_2(\boldsymbol{x}, \widetilde{\boldsymbol{x}}) + k_3(\boldsymbol{x}, \widetilde{\boldsymbol{x}}), \tag{119}$$

where:

$$k_1(\boldsymbol{x}, \widetilde{\boldsymbol{x}}) = \frac{1}{\Gamma(\nu)} \int_0^{\eta_1} z^{\nu-1} e^{-z} \exp\left(-\frac{\|\boldsymbol{x} - \widetilde{\boldsymbol{x}}\|_2^2}{2\ell^2 z\nu^{-1}}\right) \mathrm{d}z, \tag{120}$$

$$k_2(\boldsymbol{x}, \widetilde{\boldsymbol{x}}) = \frac{1}{\Gamma(\nu)} \int_{\eta_1}^{\eta_2} z^{\nu-1} e^{-z} \exp\left(-\frac{\|\boldsymbol{x} - \widetilde{\boldsymbol{x}}\|_2^2}{2\ell^2 z\nu^{-1}}\right) \mathrm{d}z, \tag{121}$$

$$k_3(\boldsymbol{x}, \widetilde{\boldsymbol{x}}) = \frac{1}{\Gamma(\nu)} \int_{\eta_2}^\infty z^{\nu-1} e^{-z} \exp\left(-\frac{\|\boldsymbol{x} - \widetilde{\boldsymbol{x}}\|_2^2}{2\ell^2 z\nu^{-1}}\right) \mathrm{d}z \tag{122}$$

with some $\eta_2 > \eta_1 > 0$[7]. Then, as with the proof of Theorem 3 in [34], we have

$$
\begin{aligned}
&\ln \det(\boldsymbol{I}_T + \sigma^{-2}\boldsymbol{K}(\boldsymbol{X}_T, \boldsymbol{X}_T)) \\
&\leq \underbrace{\ln \det(\boldsymbol{I}_T + \sigma^{-2}\boldsymbol{K}_1(\boldsymbol{X}_T, \boldsymbol{X}_T))}_{(i)} + \underbrace{\ln \det(\boldsymbol{I}_T + \sigma^{-2}\boldsymbol{K}_2(\boldsymbol{X}_T, \boldsymbol{X}_T))}_{(ii)} + \underbrace{\ln \det(\boldsymbol{I}_T + \sigma^{-2}\boldsymbol{K}_3(\boldsymbol{X}_T, \boldsymbol{X}_T))}_{(iii)},
\end{aligned}
\tag{123}
$$

where $\boldsymbol{K}_1(\boldsymbol{X}_T, \boldsymbol{X}_T), \boldsymbol{K}_2(\boldsymbol{X}_T, \boldsymbol{X}_T)$, and $\boldsymbol{K}_3(\boldsymbol{X}_T, \boldsymbol{X}_T)$ are the kernel matrix of $k_1, k_2$, and $k_3$, respectively. We set sufficiently small $\eta_1$ and large $\eta_2$ so that the crude upper bound of the first term (i) and the third term (iii) are sufficiently small. For this purpose, the following settings of $\eta_1$ and $\eta_2$ are sufficient (we confirm in the next paragraphs):

$$
\eta_1 = \left(\frac{\nu \Gamma(\nu) \sigma^2}{T^2}\right)^{\frac{1}{\nu}}, \quad \eta_2 = \max\left\{1, \frac{\Gamma(\nu)}{C_\nu} \ln \frac{T^2}{\sigma^2}\right\},
\tag{124}
$$

where $C_\nu > 0$ is the constant such that $\forall z \geq 1, \ z^{\nu-1}e^{-z} \leq C_\nu e^{-z/2}$. Hereafter, we suppose that $\eta_1 < \eta_2$ holds with the above definition. The case $\eta_1 \geq \eta_2$ is considered in the last parts of the proof.

**Upper bound for the first term (i).** From the definition of $k_1$, we have

$$
|k_1(\boldsymbol{x}, \widetilde{\boldsymbol{x}})| = \frac{1}{\Gamma(\nu)} \int_0^{\eta_1} z^{\nu-1} e^{-z} \exp\left(-\frac{\|\boldsymbol{x} - \widetilde{\boldsymbol{x}}\|_2^2}{2\ell^2 z\nu^{-1}}\right) \mathrm{d}z
\tag{125}
$$

$$
\leq \frac{1}{\Gamma(\nu)} \int_0^{\eta_1} z^{\nu-1} e^{-z} \mathrm{d}z
\tag{126}
$$

$$
\leq \frac{1}{\Gamma(\nu)} \int_0^{\eta_1} z^{\nu-1} \mathrm{d}z
\tag{127}
$$

$$
= \frac{1}{\nu \Gamma(\nu)} [z^\nu]_0^{\eta_1}
\tag{128}
$$

$$
= \frac{1}{\nu \Gamma(\nu)} \eta_1^\nu.
\tag{129}
$$

Then, from the definition of $\eta_1$, we have

$$
\frac{1}{\nu \Gamma(\nu)} \eta_1^\nu = \frac{1}{\nu \Gamma(\nu)} \left(\frac{\nu \Gamma(\nu) \sigma^2}{T^2}\right) = \frac{\sigma^2}{T^2}.
\tag{130}
$$

Therefore, by denoting the eigenvalues of $\boldsymbol{K}_1(\boldsymbol{X}_T, \boldsymbol{X}_T)$ with decreasing order as $(\lambda_i)_{i \in [T]}$[8], we have

$$
\ln \det(\boldsymbol{I}_T + \sigma^{-2}\boldsymbol{K}_1(\boldsymbol{X}_T, \boldsymbol{X}_T)) = \ln \prod_{i=1}^T (1 + \sigma^{-2}\lambda_i)
\tag{131}
$$

$$
\leq \ln(1 + \sigma^{-2}\lambda_1)^T
\tag{132}
$$

$$
\leq \ln(1 + T^{-1})^T,
\tag{133}
$$

where the last inequality follows from $\lambda_1 \leq \sqrt{\sum_{i=1}^T \lambda_i^2} = \|\boldsymbol{K}_1(\boldsymbol{X}_T, \boldsymbol{X}_T)\|_F = \sqrt{\sum_{i,j} k_1(\boldsymbol{x}_i, \boldsymbol{x}_j)^2} \leq \sigma^2/T$. Since $\ln(1 + T^{-1})^T \to 1$ as $T \to \infty$, there exists constant $C > 0$ such that $\ln \det(\boldsymbol{I}_T + \sigma^{-2}\boldsymbol{K}_1(\boldsymbol{X}_T, \boldsymbol{X}_T)) \leq C$ for all $T \in \mathbb{N}_+$.

---

[7]Note that the linear combination of the positive definite kernel with non-negative coefficients and its limit are also positive definite. Therefore, $k_1, k_2$, and $k_3$ are also positive definite as far as $\eta_2 > \eta_1$.

[8]Note that $\lambda_i$ is non-negative from the positive semi-definiteness of $\boldsymbol{K}_1(\boldsymbol{X}_T, \boldsymbol{X}_T)$.

**Upper bound for the third term (iii).** From the definition of $k_3$, we have

$$|k_3(\boldsymbol{x}, \widetilde{\boldsymbol{x}})| = \frac{1}{\Gamma(\nu)} \int_{\eta_2}^{\infty} z^{\nu-1} e^{-z} \exp\left(-\frac{\|\boldsymbol{x} - \widetilde{\boldsymbol{x}}\|_2^2}{2\ell^2 z \nu^{-1}}\right) \mathrm{d}z \tag{134}$$

$$\leq \frac{1}{\Gamma(\nu)} \int_{\eta_2}^{\infty} z^{\nu-1} e^{-z} \mathrm{d}z \tag{135}$$

$$\leq \frac{C_\nu}{\Gamma(\nu)} \int_{\eta_2}^{\infty} e^{-z/2} \mathrm{d}z \tag{136}$$

$$= \frac{-2C_\nu}{\Gamma(\nu)} [e^{-z/2}]_{\eta_2}^{\infty} \tag{137}$$

$$= \frac{2C_\nu}{\Gamma(\nu)} \exp\left(-\frac{\eta_2}{2}\right). \tag{138}$$

Then, from the definition of $\eta_2$, we have

$$\frac{2C_\nu}{\Gamma(\nu)} \exp\left(-\frac{\eta_2}{2}\right) \leq \frac{2C_\nu}{\Gamma(\nu)} \exp\left(-\frac{\Gamma(\nu)}{2C_\nu} \ln\left(\frac{T^2}{\sigma^2}\right)\right) = \frac{\sigma^2}{T^2}. \tag{139}$$

By following the same arguments after Eq. (130) in the upper bound of the first term (i), we conclude that there exists constant $C > 0$ such that $\ln \det(\boldsymbol{I}_T + \sigma^{-2} \mathbf{K}_3(\mathbf{X}_T, \mathbf{X}_T)) \leq C$ for all $T \in \mathbb{N}_+$.

**Upper bound for the second term (ii).** We further divide $k_2$ with dyadic manner:

$$k_2(\boldsymbol{x}, \widetilde{\boldsymbol{x}}) = \sum_{q=1}^{Q} k_2^{(q)}(\boldsymbol{x}, \widetilde{\boldsymbol{x}}), \tag{140}$$

where:

$$k_2^{(q)}(\boldsymbol{x}, \widetilde{\boldsymbol{x}}) = \frac{1}{\Gamma(\nu)} \int_{\eta_1 2^{q-1}}^{\min\{\eta_1 2^q, \eta_2\}} z^{\nu-1} e^{-z} \exp\left(-\frac{\|\boldsymbol{x} - \widetilde{\boldsymbol{x}}\|_2^2}{2\ell^2 z \nu^{-1}}\right) \mathrm{d}z. \tag{141}$$

Here, $Q \in \mathbb{N}_+$ is the minimum number such that $\eta_1 2^Q \geq \eta_2$ holds. Then, as with Eq. (123), we have

$$\ln \det(\boldsymbol{I}_T + \sigma^{-2} \mathbf{K}_2(\mathbf{X}_T, \mathbf{X}_T)) \leq \sum_{q=1}^{Q} \ln \det\left(\boldsymbol{I}_T + \sigma^{-2} \mathbf{K}_2^{(q)}(\mathbf{X}_T, \mathbf{X}_T)\right), \tag{142}$$

where $\mathbf{K}_2^{(q)}(\mathbf{X}_T, \mathbf{X}_T)$ is the kernel matrix of $k_2^{(q)}$. Next, for any $q$, we define new kernel function $\widetilde{k}^{(q)}(\boldsymbol{x}, \widetilde{\boldsymbol{x}})$ as

$$\widetilde{k}_2^{(q)}(\boldsymbol{x}, \widetilde{\boldsymbol{x}}) = k_2^{(q)}(\boldsymbol{x}, \widetilde{\boldsymbol{x}}) \exp\left(-\frac{\left(\sqrt{1 - \|\boldsymbol{x}\|_2^2} - \sqrt{1 - \|\widetilde{\boldsymbol{x}}\|_2^2}\right)^2}{2\ell^2 \nu^{-1} \min\{\eta_1 2^q, \eta_2\}}\right). \tag{143}$$

We further denote the kernel matrix of $\widetilde{k}_2^{(q)}$ by $\widetilde{\mathbf{K}}_2^{(q)}(\mathbf{X}_T, \mathbf{X}_T)$. Then, from Oppenheim's inequality, we have

$$\ln \det\left(\boldsymbol{I}_T + \sigma^{-2} \mathbf{K}_2^{(q)}(\mathbf{X}_T, \mathbf{X}_T)\right) \leq \ln \det\left(\boldsymbol{I}_T + \sigma^{-2} \widetilde{\mathbf{K}}_2^{(q)}(\mathbf{X}_T, \mathbf{X}_T)\right). \tag{144}$$

Furthermore, for any $z \in [\eta_1 2^{q-1}, \min\{\eta_1 2^q, \eta_2\}]$, the following kernel function $\widehat{k}_2^{(q)}(\boldsymbol{x}, \widetilde{\boldsymbol{x}}; z)$ is positive definite (e.g., Lemma A.5 in [10]):

$$\widehat{k}_2^{(q)}(\boldsymbol{x}, \widetilde{\boldsymbol{x}}; z) = 2 \exp\left(-\frac{\left(\sqrt{1 - \|\boldsymbol{x}\|_2^2} - \sqrt{1 - \|\widetilde{\boldsymbol{x}}\|_2^2}\right)^2}{2\ell^2 \nu^{-1} z}\right) - \exp\left(-\frac{\left(\sqrt{1 - \|\boldsymbol{x}\|_2^2} - \sqrt{1 - \|\widetilde{\boldsymbol{x}}\|_2^2}\right)^2}{2\ell^2 \nu^{-1} \min\{\eta_1 2^q, \eta_2\}}\right).$$

$$\tag{145}$$

Note that $2k_2^{(q)}(\widetilde{\boldsymbol{x}}_i, \widetilde{\boldsymbol{x}}_j) - \widetilde{k}_2^{(q)}(\boldsymbol{x}_i, \boldsymbol{x}_j)$ is represented as

$$2k_2^{(q)}(\widetilde{\boldsymbol{x}}_i, \widetilde{\boldsymbol{x}}_j) - \widetilde{k}_2^{(q)}(\boldsymbol{x}_i, \boldsymbol{x}_j) \tag{146}$$

$$= \frac{1}{\Gamma(v)} \int_{\eta_1 2^{q-1}}^{\min\{\eta_1 2^q, \eta_2\}} z^{v-1} e^{-z} \exp\left(-\frac{\|\boldsymbol{x}_i - \boldsymbol{x}_j\|_2^2}{2\ell^2 z v^{-1}}\right) \widehat{k}_2^{(q)}(\boldsymbol{x}_i, \boldsymbol{x}_j; z) \mathrm{d}z. \tag{147}$$

By noting that the product of two positive definite kernels is also positive definite, the above expression implies that $2\mathbf{K}_2^{(q)}(\widetilde{\mathbf{X}}_T, \widetilde{\mathbf{X}}_T) - \widetilde{\mathbf{K}}_2^{(q)}(\mathbf{X}_T, \mathbf{X}_T)$ is the positive semi-definite matrix. Therefore, we have[9]

$$\sum_{q=1}^{Q} \ln\det\left(\boldsymbol{I}_T + \sigma^{-2}\widetilde{\mathbf{K}}_2^{(q)}(\mathbf{X}_T, \mathbf{X}_T)\right) \leq \sum_{q=1}^{Q} \ln\det\left(\boldsymbol{I}_T + 2\sigma^{-2}\mathbf{K}_2^{(q)}(\widetilde{\mathbf{X}}_T, \widetilde{\mathbf{X}}_T)\right) \tag{148}$$

$$\leq Q \ln\det\left(\boldsymbol{I}_T + 2\sigma^{-2}\mathbf{K}(\widetilde{\mathbf{X}}_T, \widetilde{\mathbf{X}}_T)\right), \tag{149}$$

where the second inequality follows from the fact that $\mathbf{K}(\widetilde{\mathbf{X}}_T, \widetilde{\mathbf{X}}_T) - \mathbf{K}_2^{(q)}(\widetilde{\mathbf{X}}_T, \widetilde{\mathbf{X}}_T)$ is positive semi-definite. From the definition of $Q$, we have

$$Q \leq \log_2\left(\frac{\eta_2}{\eta_1}\right) + 1 \tag{150}$$

$$= \log_2 \eta_2 - \log_2 \eta_1 + 1 \tag{151}$$

$$= \log_2 \max\left\{1, \frac{\Gamma(v)}{C_v} \ln\frac{T^2}{\sigma^2}\right\} - \log_2\left(\frac{v\Gamma(v)\sigma^2}{T^2}\right)^{\frac{1}{v}} + 1 \tag{152}$$

$$\leq \log_2\left(1 + \frac{\Gamma(v)}{C_v} \ln\frac{T^2}{\sigma^2}\right) + \frac{1}{v}\log_2\left(\frac{T^2}{v\Gamma(v)\sigma^2}\right) + 1. \tag{153}$$

By combining Eqs. (142), (144), (149), and (153), we conclude

$$\ln\det(\boldsymbol{I}_T + \sigma^{-2}\mathbf{K}_2(\mathbf{X}_T, \mathbf{X}_T)) \tag{154}$$

$$\leq \left[\log_2\left(1 + \frac{\Gamma(v)}{C_v} \ln\frac{T^2}{\sigma^2}\right) + \frac{1}{v}\log_2\left(\frac{T^2}{v\Gamma(v)\sigma^2}\right) + 1\right] \ln\det\left(\boldsymbol{I}_T + 2\sigma^{-2}\mathbf{K}(\widetilde{\mathbf{X}}_T, \widetilde{\mathbf{X}}_T)\right). \tag{155}$$

By aggregating the upper bounds of (i), (ii), and (iii), we have the following inequality under $\eta_1 < \eta_2$:

$$\ln\det(\boldsymbol{I}_T + \sigma^{-2}\mathbf{K}(\mathbf{X}_T, \mathbf{X}_T)) \leq C(T, v, \sigma^2) \ln\det\left(\boldsymbol{I}_T + 2\sigma^{-2}\mathbf{K}(\widetilde{\mathbf{X}}_T, \widetilde{\mathbf{X}}_T)\right) + 2C. \tag{156}$$

Finally, if $\eta_1 \geq \eta_2$, we have

$$\ln\det(\boldsymbol{I}_T + \sigma^{-2}\mathbf{K}(\mathbf{X}_T, \mathbf{X}_T)) \tag{157}$$

$$\leq \ln\det(\boldsymbol{I}_T + \sigma^{-2}(\mathbf{K}_1(\mathbf{X}_T, \mathbf{X}_T) + \mathbf{K}_3(\mathbf{X}_T, \mathbf{X}_T))) \tag{158}$$

$$\leq \ln\det(\boldsymbol{I}_T + \sigma^{-2}\mathbf{K}_1(\mathbf{X}_T, \mathbf{X}_T)) + \ln\det(\boldsymbol{I}_T + \sigma^{-2}\mathbf{K}_3(\mathbf{X}_T, \mathbf{X}_T)) \tag{159}$$

$$\leq 2C \tag{160}$$

$$\leq C(T, v, \sigma^2) \ln\det\left(\boldsymbol{I}_T + 2\sigma^{-2}\mathbf{K}(\widetilde{\mathbf{X}}_T, \widetilde{\mathbf{X}}_T)\right) + 2C, \tag{161}$$

where the last inequality follows from $\ln\det\left(\boldsymbol{I}_T + 2\sigma^{-2}\mathbf{K}(\widetilde{\mathbf{X}}_T, \widetilde{\mathbf{X}}_T)\right) \geq 0$ and $C(T, v, \sigma^2) \geq 0$. The desired result is obtained by setting a new absolute constant $C$ as $2C$ in the above inequality. □

---

[9]For any positive semi-definite matrices $A$, $B$ such that $A - B$ is positive semi-definite, we have $\lambda_i^{(A)} \geq \lambda_i^{(B)}$, where $(\lambda_i^{(A)})$ and $(\lambda_i^{(B)})$ is a non-negative eigenvalues of $A$ and $B$ with decreasing order. (This is a consequence of Courant–Fischer's min-max theorem.) Therefore, we have $\det(A) = \prod_i \lambda_i^{(A)} \geq \prod_i \lambda_i^{(B)} = \det(B)$ for such $A$ and $B$.

## B.2 Summary of Mercer Decomposition for Dot-Product Kernel on Sphere

In this subsection, we summarize the basic known results of the Mercer decomposition on $\mathbb{S}^d$. The content of this subsection is related to the analysis of the spherical harmonics. We refer to [1, 16] as the basic textbook. In the kernel method literature, the Mercer decomposition of the dot-product kernel and its eigendecay have been studied. See, e.g., [2, 36, 48]. Furthermore, the existing analysis of the neural tangent kernel also leverages the Mercer decomposition based on the spherical harmonics. We also refer to the appendix of [3, 5] as the related works of this subsection.

We first describe Mercer's theorem. Let $L^2(\mathcal{X}, \mu) := \{f : \mathcal{X} \to \mathbb{R} \mid \int_{\mathcal{X}} f^2(\boldsymbol{x}) \mu(\mathrm{d}\boldsymbol{x}) < \infty\}$ be the square-integrable functions on $\mathcal{X}$ under the measure $\mu$. Furthermore, let us define the kernel integral operator $\mathcal{T}_k : L^2(\mathcal{X}, \mu) \to L^2(\mathcal{X}, \mu)$ of a square-integrable kernel function $k : \mathcal{X} \times \mathcal{X} \to \mathbb{R}$ as $(\mathcal{T}_k f)(\cdot) = \int_{\mathcal{X}} k(\cdot, \boldsymbol{x}) f(\boldsymbol{x}) \mu(\mathrm{d}\boldsymbol{x})$. Then, Mercer's theorem guarantees that the positive kernel $k$ is decomposed based on the eigenvalues and eigenfunctions sequence of $\mathcal{T}_k$ with absolute and uniform convergence on $\mathcal{X} \times \mathcal{X}$. We give the formal statement below.

**Theorem 10** (Mercer's theorem, e.g., Theorem 4.49 in [12]). *Let $\mathcal{X}$ be a compact metric space, $\mu$ be a finite Borel measure whose support is $\mathcal{X}$, and $k : \mathcal{X} \times \mathcal{X} \to \mathbb{R}$ be a continuous and square integrable-positive definite kernel on $(\mathcal{X}, \mu)$. Suppose that $(\phi_i)_{i \in \mathbb{N}}$ and $(\lambda_i)_{i \in \mathbb{N}}$ are eigenfunctions and eigenvalues of the kernel integral operator $\mathcal{T}_k$, respectively. Namely, $(\phi_i)_{i \in \mathbb{N}}$ is an orthonormal bases of the eigenspace $\{\mathcal{T}_k f \mid f \in L^2(\mathcal{X}, \mu)\}$ such that $\mathcal{T}_k \phi_i(\cdot) = \lambda_i \phi_i(\cdot)$ for all $i \in \mathbb{N}$. Then, we have*

$$k(\boldsymbol{x}, \widetilde{\boldsymbol{x}}) = \sum_{i \in \mathbb{N}} \lambda_i \phi_i(\boldsymbol{x}) \phi_i(\widetilde{\boldsymbol{x}}), \tag{162}$$

*where the convergence is absolute and uniform on $\mathcal{X} \times \mathcal{X}$.*

Specifically, our interest is the Mercer decomposition of the kernel on $\mathbb{S}^d$. This is given as spherical harmonics on $\mathbb{S}^d$, which we define below.

**Definition 1** (Spherical harmonics, e.g., Definition 2.7 in [1]). *Fix any $d \geq 1$ and $m \in \mathbb{N}$. Let $\mathbb{Y}_m(\mathbb{R}^{d+1})$ be the all homogeneous polynomials of degree $m$ in $\mathbb{R}^{d+1}$ that are also harmonic[10]. The space $\mathbb{Y}_m^{d+1} = \mathbb{Y}_m(\mathbb{R}^{d+1})\mid_{\mathbb{S}^d}$ is called the spherical harmonic space of order $m$ in $d + 1$ dimensions. Any function in $\mathbb{Y}_m^{d+1}$ is called a spherical harmonic of order $m$ in $d + 1$ dimensions.*

The following lemmas provide the properties of the spherical harmonics, which guarantee that the Mercer decomposition of the continuous dot-product kernel on $\mathbb{S}^d$ is defined based on spherical harmonics.

**Lemma 11** (Dimension and completeness of sphererical harmonics, e.g., Chapter 2.1.3, Corollary 2.15, and Theorem 2.38 in [1]). *Fix any $d \geq 1$. Then, the following statements hold:*

- *For any $m \in \mathbb{N}$, we have $\dim(\mathbb{Y}_m^{d+1}) = N_{d+1,m}$ with $N_{d+1,m} = \frac{(2m+d-1)(m+d-2)!}{m!(d-1)!}$. Furthermore, For any $m, n \in \mathbb{N}$ with $m \neq n$, we have $\mathbb{Y}_m^{d+1} \perp \mathbb{Y}_n^{d+1}$[11].*

- *Let us define $(Y_{m,j})_{j \in [N_{d+1,m}]}$ be an orthonormal bases of $\mathbb{Y}_m^{d+1}$. Then, $\cup_{m \in \mathbb{N}} (Y_{m,j})_{j \in [N_{d+1,m}]}$ becomes an orthonormal bases of $L^2(\mathbb{S}^d, \sigma)$, where $\sigma(\cdot)$ is the induced Lebesgue measure on $\mathbb{S}^d$.*

**Lemma 12** (Funk-Hecke Formula, e.g., Theorem 2.22 in [1] or Theorem 4.24 in [16]). *Fix any $d \geq 1$. Let $f : [-1, 1] \to \mathbb{R}$ be a continuous function. Define $|\mathbb{S}^{d-1}| := \frac{2\pi^{d/2}}{\Gamma(d/2)}$ as the surface area of $\mathbb{S}^{d-1}$. Then, for any $m \in \mathbb{N}$ and $Y_m \in \mathbb{Y}_m^{d+1}$, we have*

$$\int_{\mathbb{S}^d} f(\boldsymbol{z}^\top \boldsymbol{\eta}) Y_m(\boldsymbol{\eta}) \sigma(\mathrm{d}\boldsymbol{\eta}) = \lambda_m Y_m(\boldsymbol{z}), \tag{163}$$

*where $\sigma(\cdot)$ is the induced Lebesgue measure on $\mathbb{S}^d$. Furthermore, $\lambda_m$ is defined as*

$$\lambda_m = |\mathbb{S}^{d-1}| \int_{-1}^{1} P_{m,d+1}(t) f(t) (1 - t^2)^{\frac{d-2}{2}} \mathrm{d}t, \tag{164}$$

---

[10]A polynomial $H(x_1, \ldots, x_{d+1})$ is called homogeneous of degree $m$ if $H(tx_1, \ldots, tx_{d+1}) = t^n H(x_1, \ldots, x_{d+1})$. Furthermore, a polynomial $H(x_1, \ldots, x_{d+1})$ is called harmonic if $\Delta_{d+1} H = 0$, where $\Delta_{d+1}$ is the Laplace operator. See Chapter 4 in [16] or Chapter 2 in [1].

[11]Here, as with the second statement, we consider $L^2(\mathbb{S}^d, \sigma)$. Therefore, the inner product for any $f, g : \mathbb{S}^d \to \mathbb{R}$ is defined on $L^2(\mathbb{S}^d, \sigma)$ as $\int_{\mathbb{S}^d} f(\boldsymbol{x}) g(\boldsymbol{x}) \sigma(\mathrm{d}\boldsymbol{x})$.

where $P_{m,d+1}(t)$ is the Legendre polynomial of degree $m$ in $d + 1$ dimensions, which is defined as

$$P_{m,d+1}(t) = m! \Gamma \left( \frac{d}{2} \right) \sum_{k=0}^{\lfloor m/2 \rfloor} (-1)^k \frac{(1-t^2)^k t^{m-2k}}{4^k k! (m-2k)! \Gamma \left( k + \frac{d}{2} \right)}. \tag{165}$$

Lemma 12 suggests that the spherical harmonics are eigenfunctions of the continuous dot-product kernel $k(\boldsymbol{x}, \widetilde{\boldsymbol{x}}) = \widetilde{k}(\boldsymbol{x}^\top \widetilde{\boldsymbol{x}})$ on $\mathbb{S}^d$. Furthermore, Lemma 11 guarantees the $\cup_{m \in \mathbb{N}} (Y_{m,j})_{j \in [N_{d+1,m}]}$ forms an orthonormal bases of $L^2(\mathbb{S}^d, \sigma)$, which implies that they are the orthonormal bases of the eigenspace of $\mathcal{T}_k$. These facts give the following explicit form of Mercer decomposition for a continuous dot-product kernel.

**Corollary 13.** *Fix any $d \in \mathbb{N}_+$. Suppose $\mathcal{X} = \mathbb{S}^d$. Furthermore, assume the kernel function $k : \mathcal{X} \times \mathcal{X} \to \mathbb{R}$ is the positive definite kernel such that $\forall \boldsymbol{x}, \widetilde{\boldsymbol{x}} \in \mathcal{X}, k(\boldsymbol{x}, \widetilde{\boldsymbol{x}}) = \widetilde{k}(\boldsymbol{x}^\top \widetilde{\boldsymbol{x}})$ with some continuous function $\widetilde{k} : [-1, 1] \to \mathbb{R}$. Then, we have the following Mercer decomposition of $k$:*

$$k(\boldsymbol{x}, \widetilde{\boldsymbol{x}}) = \sum_{m=0}^{\infty} \lambda_m \sum_{j=1}^{N_{d+1,m}} Y_{m,j}(\boldsymbol{x}) Y_{m,j}(\widetilde{\boldsymbol{x}}), \tag{166}$$

*where $(Y_{m,j}(\cdot))_{j \in [N_{d+1,m}]}$ denotes the spherical harmonics, which consist of orthonormal bases of $\mathbb{Y}_m^{d+1}$. Furthermore, $\lambda_m \geq 0$ is defined as*

$$\lambda_m = |\mathbb{S}^{d-1}| \int_{-1}^{1} P_{m,d+1}(t) \widetilde{k}(t) (1-t^2)^{\frac{d-2}{2}} \, dt. \tag{167}$$

Note that $\|\boldsymbol{x} - \widetilde{\boldsymbol{x}}\|_2 = \sqrt{2 - 2\boldsymbol{x}^\top \widetilde{\boldsymbol{x}}}$ for any $\boldsymbol{x}, \widetilde{\boldsymbol{x}} \in \mathbb{S}^d$. Therefore, we can represent $k_{\mathrm{SE}}$ and $k_{\mathrm{Matérn}}$ on $\mathbb{S}^d$ by Eq. (166). Finally, we describe the following addition theorem of the spherical harmonics, which plays a central role in avoiding the uniform boundness assumption in the existing proof of MIG on $\mathbb{S}^d$.

**Lemma 14** (Addition theorem, e.g., Theorem 2.9 in [1] or Theorem 4.11 in [16]). *Fix any $d \geq 1$ and $m \in \mathbb{N}$. Let $(Y_{m,j})_{j \in [N_{d+1,m}]}$ be an orthonormal bases of $\mathbb{Y}_m^{d+1}$. Then, we have*

$$\forall \boldsymbol{x}, \widetilde{\boldsymbol{x}} \in \mathbb{S}^d, \quad \sum_{j=1}^{N_{d+1,m}} Y_{m,j}(\boldsymbol{x}) Y_{m,j}(\widetilde{\boldsymbol{x}}) = \frac{N_{d+1,m}}{|\mathbb{S}^d|} P_{m,d+1}(\boldsymbol{x}^\top \widetilde{\boldsymbol{x}}), \tag{168}$$

*where $P_{m,d+1}(t)$ is the Legendre polynomial of degree $m$ in $d + 1$ dimensions, which is defined in Eq. (165).*

## B.3 Upper Bound of MIG with Mercer Decomposition

By using Corollary 13 and Lemma 14 in the previous subsection, we can derive the following general form of the upper bound of MIG.

**Lemma 15** (Adapted from [57]). *Suppose the kernel function $k$ satisfies the condition in Corollary 13. Furthermore, assume $|k(\boldsymbol{x}, \widetilde{\boldsymbol{x}})| \leq 1$ for all $\boldsymbol{x}, \boldsymbol{x} \in \mathcal{X}$. Then, for any $M \in \mathbb{N}$, MIG on $\mathbb{S}^d$ satisfies*

$$\frac{1}{2} \max_{\boldsymbol{x}_1, \dots, \boldsymbol{x}_T \in \mathbb{S}^d} \ln \det(\boldsymbol{I}_T + \sigma^{-2} \mathbf{K}(\mathbf{X}_T, \mathbf{X}_T)) \leq N_M \ln \left( 1 + \frac{T}{\sigma^2} \right) + \frac{T}{|\mathbb{S}^d| \sigma^2} \sum_{m=M+1}^{\infty} \lambda_m N_{d+1,m}, \tag{169}$$

*where $N_M = \sum_{m=0}^{M} N_{d+1,m}$.*

The proof almost directly follows from [58], while a minor modification is required to deal with the unboundness of the eigenfunctions through the addition theorem. The same proof strategy is already provided in [31, 57] for analyzing the MIG of the neural tangent kernel on the sphere. Although our proof has no intrinsic change from their proof, we give the details below for completeness of our paper.

*Proof.* We first decompose the kernel matrix as $\mathbf{K}(\mathbf{X}_T, \mathbf{X}_T) = \mathbf{K}_{\text{head}} + \mathbf{K}_{\text{tail}}$, where $[\mathbf{K}_{\text{head}}]_{i,l} = \sum_{m=0}^{M} \lambda_m \sum_{j=1}^{N_{d+1,m}} Y_{m,j}(\boldsymbol{x}_i) Y_{m,j}(\boldsymbol{x}_l)$ and $[\mathbf{K}_{\text{tail}}]_{i,l} = \sum_{m=M+1}^{\infty} \lambda_m \sum_{j=1}^{N_{d+1,m}} Y_{m,j}(\boldsymbol{x}_i) Y_{m,j}(\boldsymbol{x}_l)$. Then, as with the proof in [58], the MIG is decomposed as

$$\frac{1}{2} \max_{\boldsymbol{x}_1,\dots,\boldsymbol{x}_T \in \mathbb{S}^d} \ln \det(\boldsymbol{I}_T + \sigma^{-2} \mathbf{K}(\mathbf{X}_T, \mathbf{X}_T)) \tag{170}$$

$$= \frac{1}{2} \ln \det \left( \boldsymbol{I}_T + \frac{1}{\sigma^2} \mathbf{K}_{\text{head}} \right) + \frac{1}{2} \ln \det \left( \boldsymbol{I}_T + \frac{1}{\sigma^2} \left( \boldsymbol{I}_T + \frac{1}{\sigma^2} \mathbf{K}_{\text{head}} \right)^{-1} \mathbf{K}_{\text{tail}} \right). \tag{171}$$

Based on the feature representation of the kernel, the first term is further bounded from above as follows (see [57, 58]):

$$\frac{1}{2} \ln \det \left( \boldsymbol{I}_T + \frac{1}{\sigma^2} \mathbf{K}_{\text{head}} \right) \le N_M \ln \left( 1 + \frac{T}{\sigma^2 N_M} \right) \le N_M \ln \left( 1 + \frac{T}{\sigma^2} \right), \tag{172}$$

where the second inequality follows from $N_M \ge 1$. Regarding the second term, as with [58], we have

$$\frac{1}{2} \ln \det \left( \boldsymbol{I}_T + \frac{1}{\sigma^2} \left( \boldsymbol{I}_T + \frac{1}{\sigma^2} \mathbf{K}_{\text{head}} \right)^{-1} \mathbf{K}_{\text{tail}} \right) \tag{173}$$

$$\le T \ln \left( T^{-1} \mathrm{Tr} \left( \boldsymbol{I}_T + \frac{1}{\sigma^2} \left( \boldsymbol{I}_T + \frac{1}{\sigma^2} \mathbf{K}_{\text{head}} \right)^{-1} \mathbf{K}_{\text{tail}} \right) \right) \tag{174}$$

$$\le T \ln \left( T^{-1} \left( T + \frac{1}{\sigma^2} \mathrm{Tr} \left( \mathbf{K}_{\text{tail}} \right) \right) \right), \tag{175}$$

where the first inequality follows from $\ln \det(A) \le T \ln(\mathrm{Tr}(A)/T)$ for any positive definite matrix $A \in \mathbb{R}^{T \times T}$ (e.g., Lemma 1 in [58]). Then, from addition theorem (Theorem 14), we have

$$\mathrm{Tr} \left( \mathbf{K}_{\text{tail}} \right) = \sum_{t=1}^{T} \sum_{m=M+1}^{\infty} \lambda_m \sum_{j=1}^{N_{d+1,m}} Y_{m,j}(\boldsymbol{x}_t) Y_{m,j}(\boldsymbol{x}_t) \tag{176}$$

$$= \sum_{t=1}^{T} \sum_{m=M+1}^{\infty} \lambda_m \frac{N_{d+1,m}}{|\mathbb{S}^d|} P_{m,d}(\boldsymbol{x}_t^\top \boldsymbol{x}_t) \tag{177}$$

$$= \frac{T}{|\mathbb{S}^d|} \sum_{m=M+1}^{\infty} \lambda_m N_{d+1,m}, \tag{178}$$

where the last line use $P_{m,d}(\boldsymbol{x}_t^\top \boldsymbol{x}_t) = P_{m,d}(1) = 1$. By combining the above equation with Eq. (175), we have

$$\frac{1}{2} \ln \det \left( \boldsymbol{I}_T + \frac{1}{\sigma^2} \left( \boldsymbol{I}_T + \frac{1}{\sigma^2} \mathbf{K}_{\text{head}} \right)^{-1} \mathbf{K}_{\text{tail}} \right) \le T \ln \left( 1 + \frac{1}{\sigma^2 |\mathbb{S}^d|} \sum_{m=M+1}^{\infty} \lambda_m N_{d+1,m} \right) \tag{179}$$

$$\le \frac{T}{\sigma^2 |\mathbb{S}^d|} \sum_{m=M+1}^{\infty} \lambda_m N_{d+1,m}, \tag{180}$$

where the last line use $\forall z \in \mathbb{R}, \ln(1 + z) \le z$. $\qquad \square$

To obtain the explicit upper bound of Eq. (169), we introduce the following lemma.

**Lemma 16** (Upper bound of $N_{d+1,m}$ and $N_M$). *Fix any $d \in \mathbb{N}_+$. Then, for any $m \in \mathbb{N}_+$, we have*

$$N_{d+1,m} \le (d+1)e^{d-1}m^{d-1}. \tag{181}$$

*Futhermore, for any $M \in \mathbb{N}$, we have*

$$N_M \le 1 + (d+1)e^{d-1}M^d. \tag{182}$$

*Proof.* Recall $N_{d+1,m} = \frac{(2m+d-1)(m+d-2)!}{m!(d-1)!}$. Under $d = 1$, we have

$$N_{d+1,m} = \frac{(2m)(m-1)!}{m!} = 2 = (d+1)e^{d-1}m^{d-1} \tag{183}$$

for any $m \in \mathbb{N}_+$. Under $d \geq 2$, since $N_{d+1,m} = \frac{(2m+d-1)(m+d-2)!}{m!(d-1)!} = \frac{2m+d-1}{m}\binom{m+d-2}{d-1}$ and $\binom{m+d-2}{d-1} \leq \left(\frac{(m+d-2)e}{d-1}\right)^{d-1}$, we have

$$N_{d+1,m} \leq \frac{2m+d-1}{m}\left(\frac{(m+d-2)e}{d-1}\right)^{d-1} \tag{184}$$

$$\leq (2+d-1)e^{d-1}\left(\frac{m+d-2}{d-1}\right)^{d-1} \tag{185}$$

$$\leq (d+1)e^{d-1}m^{d-1}. \tag{186}$$

Finally, since $N_{d+1,0} = 1$, we have

$$N_M = 1 + \sum_{m=1}^{M} N_{d+1,m} \leq 1 + (d+1)e^{d-1}\sum_{m=1}^{M} m^{d-1} \leq 1 + (d+1)e^{d-1}M^{d-1}. \tag{187}$$

$\square$

## B.4 Eigendecay of SE and Matérn Kernel

To obtain the explicit upper bound of Eq. (169), we need the upper bound of the eigenvalue in Eq. (167) under SE and Matérn kernel. Regarding SE kernel, several existing works have already studied it [36, 39]. We formally provide the following lemma from [36].

**Lemma 17** (Eigendecay for $k = k_{\mathrm{SE}}$ on $\mathbb{S}^d$, Theorem 2 in [36]). *Fix any $d \in \mathbb{N}_+$, $\theta > 0$, and define $\mathcal{X} = \mathbb{S}^d$. Suppose that $k : \mathcal{X} \times \mathcal{X} \to \mathbb{R}$ is defined as $k(x, \widetilde{x}) = \exp\left(-\frac{\|x-\widetilde{x}\|_2^2}{\theta}\right)$. Then, the eigenvalues $(\lambda_m)_{m\in\mathbb{N}_+}$ defined in (167) satisfy*

$$\lambda_m < |\mathbb{S}^d|\left(\frac{2e}{\theta}\right)^m \frac{(2e)^{\frac{d+1}{2}}\Gamma\left(\frac{d+1}{2}\right)}{\sqrt{\pi}(2m+d-1)^{m+\frac{d}{2}}} \exp\left(-\frac{2}{\theta} + \frac{1}{\theta^2}\right). \tag{188}$$

Regarding Matérn kernel, we provide the upper bound of $\lambda_m$ for $\nu > 1/2$ by extending the proof in [20], which studies $\lambda_m$ for Laplace kernel (Matérn with $\nu = 1/2$). As with the proof in [20], we leverage the following lemma, which relates the spectral density of the kernel to $\lambda_m$.

**Lemma 18** (Eigenvalues and spectral density, Theorem 4.1 in [38]). *Fix any $d \in \mathbb{N}_+$. Suppose that $k : \mathbb{R}^{d+1} \times \mathbb{R}^{d+1} \to \mathbb{R}$ is a positive definite, stationary, and isotropic kernel function on $\mathbb{R}^{d+1}$ such that $\forall x, \widetilde{x}, k(x, \widetilde{x}) = \Phi(x - \widetilde{x})$ for some function $\Phi(\cdot)$. Furthermore, suppose $\Phi(\cdot)$ is represented as*

$$\Phi(x) = \frac{1}{(2\pi)^{d+1}}\int_{\mathbb{R}^{d+1}} \widehat{\Phi}(\|\eta\|_2)e^{i\eta^\top x}\mathrm{d}\eta, \tag{189}$$

*for some function $\widehat{\Phi}$ such that $\forall a \geq 0, \widehat{\Phi}(a) \geq 0$ and $\int_{\mathbb{R}^{d+1}} \widehat{\Phi}(\|\eta\|_2)\mathrm{d}\eta < \infty$. Then, there exists a function $\widetilde{k} : [-1, 1] \to \mathbb{R}$ such that $\forall x, \widetilde{x} \in \mathbb{S}^d, \widetilde{k}(x^\top\widetilde{x}) = k(x, \widetilde{x})$. Furthermore, $\lambda_m$ in Eq. (167) is given by*

$$\lambda_m = \int_0^\infty t\widehat{\Phi}(t)B_{m+\frac{d-1}{2}}^2(t)\mathrm{d}t, \tag{190}$$

*where $B_{m+\frac{d-1}{2}}(\cdot)$ is the usual Bessel function of the first kind and of order $m + \frac{d-1}{2}$.*

In the Matern kernel, the spectral density that satisfies the conditions in the lemma is defined when $\nu > 1/2$. Then, the explicit form of $\widehat{\Phi}(t)$ is given as:

$$\widehat{\Phi}(t) = \frac{C_{d,\nu}}{\ell^{2\nu}}\left(\frac{2\nu}{\ell^2} + t^2\right)^{-\left(\nu+\frac{d+1}{2}\right)}, \tag{191}$$

where

$$C_{d,\nu} = \frac{2^{d+1}\pi^{(d+1)/2}\Gamma\left(\nu + \frac{d+1}{2}\right)(2\nu)^{\nu}}{\Gamma(\nu)}. \tag{192}$$

See, Chapter 4.2 in [41]. By using Lemma 18, we obtain the following lemma.

**Lemma 19** (Eigendecay for $k = k_{\text{Matérn}}$ on $\mathbb{S}^d$)**.** *Fix any $d \in \mathbb{N}_+$, $\ell > 0$, and define $X = \mathbb{S}^d$. Suppose that $k : X \times X \to \mathbb{R}$ is defined as $k(x,\widetilde{x}) = \frac{2^{1-\nu}}{\Gamma(\nu)}\left(\frac{\sqrt{2\nu}\|x-\widetilde{x}\|_2}{\ell}\right)J_{\nu}\left(\frac{\sqrt{2\nu}\|x-\widetilde{x}\|_2}{\ell}\right)$. Then, the eigenvalues $(\lambda_m)_{m\in\mathbb{N}_+}$ defined in (167) satisfies*

$$\lambda_m \leq \frac{\widetilde{C}_{d,\nu}}{\ell^{2\nu}}m^{-2\nu-d}. \tag{193}$$

*if $m > 2\nu$ and $\nu > 1/2$. Here, $\widetilde{C}_{d,\nu}$ is defined as*

$$\widetilde{C}_{d,\nu} = C_{d,\nu}\frac{\Gamma(2\nu + d)}{\Gamma^2\left(\nu + \frac{d+1}{2}\right)}\exp\left(2\nu + d + \frac{1}{6}\right). \tag{194}$$

*Proof.* From Lemma 18, we have

$$\lambda_m = \int_0^\infty t\widehat{\Phi}(t)B^2_{m+\frac{d-1}{2}}(t)\mathrm{d}t \tag{195}$$

$$= \frac{C_{d,\nu}}{\ell^{2\nu}}\int_0^\infty t\left(\frac{2\nu}{\ell^2} + t^2\right)^{-\left(\nu+\frac{d+1}{2}\right)}B^2_{m+\frac{d-1}{2}}(t)\mathrm{d}t \tag{196}$$

$$\leq \frac{C_{d,\nu}}{\ell^{2\nu}}\int_0^\infty t^{-2\nu-d}B^2_{m+\frac{d-1}{2}}(t)\mathrm{d}t. \tag{197}$$

As with the proof of Theorem 7 in [20], we evaluate the integral $\int_0^\infty t^{-2\nu-d}B^2_{m+\frac{d-1}{2}}(t)\mathrm{d}t$ by using the following identity (Chapter 13.4.1 in [22]):

$$\int_0^\infty \frac{B_p(at)B_q(at)}{t^z}\mathrm{d}t = \frac{\left(\frac{1}{2}a\right)^{z-1}\Gamma(z)\Gamma\left(\frac{1}{2}p + \frac{1}{2}q - \frac{1}{2}z + \frac{1}{2}\right)}{2\Gamma\left(\frac{1}{2}z + \frac{1}{2}q - \frac{1}{2}p + \frac{1}{2}\right)\Gamma\left(\frac{1}{2}z + \frac{1}{2}p + \frac{1}{2}q + \frac{1}{2}\right)\Gamma\left(\frac{1}{2}z + \frac{1}{2}p - \frac{1}{2}q + \frac{1}{2}\right)}, \tag{198}$$

where $p + q + 1 > z > 0$. By setting $p = q = m + \frac{d-1}{2}$, $z = 2\nu + d$, and $a = 1$, we have $p + q + 1 > z \Leftrightarrow m > \nu$. Hence, for any $m > \nu$, we have

$$\int_0^\infty t^{-2\nu-d}B^2_{m+\frac{d-1}{2}}(t)\mathrm{d}t = \left(\frac{1}{2}\right)^{2\nu+d-1}\frac{\Gamma(2\nu+d)\Gamma(m-\nu)}{2\Gamma^2\left(\nu+\frac{d+1}{2}\right)\Gamma(m+\nu+d)}. \tag{199}$$

Stirling's formula implies that there exists a constant $C > 0$ such that

$$\Gamma(m-\nu) \leq C(m-\nu)^{m-\nu-\frac{1}{2}}\exp(-m+\nu)\exp\left(\frac{1}{12(m-\nu)}\right), \tag{200}$$

$$\Gamma(m+\nu+d) \geq C(m+\nu+d)^{m+\nu+d-\frac{1}{2}}\exp\left(-(m+\nu+d)\right). \tag{201}$$

Therefore, for any $m \geq 2\nu$ with $\nu > 1/2$, we have

$$\frac{\Gamma(m-\nu)}{\Gamma(m+\nu+d)} \leq \frac{(m-\nu)^{m-\nu-\frac{1}{2}} \exp(-m+\nu) \exp\left(\frac{1}{12(m-\nu)}\right)}{(m+\nu+d)^{m+\nu+d-\frac{1}{2}} \exp(-(m+\nu+d))} \tag{202}$$

$$\leq \frac{(m-\nu)^{m-\nu-\frac{1}{2}}}{(m+\nu+d)^{m+\nu+d-\frac{1}{2}}} \exp\left(2\nu+d+\frac{1}{6}\right) \tag{203}$$

$$\leq \frac{(m-\nu)^{m-\nu-\frac{1}{2}}}{(m-\nu)^{m+\nu+d-\frac{1}{2}}} \exp\left(2\nu+d+\frac{1}{6}\right) \tag{204}$$

$$= (m-\nu)^{-2\nu-d} \exp\left(2\nu+d+\frac{1}{6}\right) \tag{205}$$

$$\leq 2^{2\nu+d} m^{-2\nu-d} \exp\left(2\nu+d+\frac{1}{6}\right), \tag{206}$$

where the second inequality follows from $m-\nu \geq 1/2$ due to $m \geq 2\nu \Leftrightarrow m-\nu \geq \nu$, the third inequality follows from $m+\nu+d \geq m-\nu$, and the last inequality follows from $m-\nu \geq m-m/2 \geq m/2$. By aggregating Eq. (197), (199), and (206), we have

$$\lambda_m \leq \frac{C_{d,\nu}}{\ell^{2\nu}} \left(\frac{1}{2}\right)^{2\nu+d-1} \frac{\Gamma(2\nu+d)}{2\Gamma^2\left(\nu+\frac{d+1}{2}\right)} 2^{2\nu+d} m^{-2\nu-d} \exp\left(2\nu+d+\frac{1}{6}\right) \tag{207}$$

$$= \frac{\widetilde{C}_{d,\nu}}{\ell^{2\nu}} m^{-2\nu-d}. \tag{208}$$

$\square$

## B.5 Proof of Theorem 7

**Squared exponential kernel.** From Lemma 17, we have

$$\lambda_m < |\mathbb{S}^d| \left(\frac{2e}{\theta}\right)^m \frac{(2e)^{\frac{d+1}{2}} \Gamma\left(\frac{d+1}{2}\right)}{\sqrt{\pi}(2m+d-1)^{m+\frac{d}{2}}} \exp\left(-\frac{2}{\theta} + \frac{1}{\theta^2}\right) \tag{209}$$

$$\leq |\mathbb{S}^d| \frac{(2e)^{\frac{d+1}{2}} \Gamma\left(\frac{d+1}{2}\right)}{\sqrt{\pi}} \exp\left(-\frac{2}{\theta} + \frac{1}{\theta^2}\right) \left(\frac{2e}{\theta}\right)^m (2m)^{-m-\frac{d}{2}} \tag{210}$$

$$\leq |\mathbb{S}^d| \frac{(2e)^{\frac{d+1}{2}} \Gamma\left(\frac{d+1}{2}\right)}{\sqrt{\pi}} \exp\left(-\frac{2}{\theta} + \frac{1}{\theta^2}\right) \left(\frac{e}{\theta}\right)^m m^{-m-\frac{d}{2}}. \tag{211}$$

Here, we set $C_{d,\theta}$ as

$$C_{d,\theta} = \frac{(2e)^{\frac{d+1}{2}} \Gamma\left(\frac{d+1}{2}\right)}{\sqrt{\pi}} \exp\left(-\frac{2}{\theta} + \frac{1}{\theta^2}\right). \tag{212}$$

Then,

$$\gamma_T(\mathcal{X}) \leq N_M \ln\left(1 + \frac{T}{\sigma^2}\right) + \frac{T}{|\mathbb{S}^d|\sigma^2} \sum_{m=M+1}^{\infty} \lambda_m N_{d+1,m} \tag{213}$$

$$\leq N_M \ln\left(1 + \frac{T}{\sigma^2}\right) + \frac{C_{d,\theta} T}{\sigma^2} \sum_{m=M+1}^{\infty} \left(\frac{e}{\theta}\right)^m m^{-m-\frac{d}{2}} N_{d+1,m} \tag{214}$$

$$\leq \left[1 + (d+1)e^{d-1}M^d\right] \ln\left(1 + \frac{T}{\sigma^2}\right) + \frac{C_{d,\theta} T}{\sigma^2}(d+1)e^{d-1} \sum_{m=M+1}^{\infty} \left(\frac{e}{\theta m}\right)^m m^{\frac{d}{2}-1}, \tag{215}$$

where the first and last inequalities follow from Lemmas 9, 15 and Lemma 16, respectively. Here, for any $d \in \mathbb{N}_+$ and $m \in \mathbb{N}_+$, we have $m^{\frac{d}{2}-1} \leq c_d^m$ with $c_d = \max\left\{1, \exp\left(\frac{1}{e}\left(\frac{d}{2}-1\right)\right)\right\}$. Indeed, when

$d \leq 2$, we have $m^{d/2-1} \leq 1 = c_d$. When $d \geq 3$, the function $g(m) = m^{\frac{1}{m}\left(\frac{d}{2}-1\right)}$ attains maximum at $m = e$ on $[1, \infty)$, which implies $g(m) \leq \exp\left(\frac{1}{e}\left(\frac{d}{2}-1\right)\right) \Rightarrow m^{d/2-1} \leq \exp\left(\frac{1}{e}\left(\frac{d}{2}-1\right)\right)^m \leq c_d^m$. Hence, we have

$$\gamma_T(\mathcal{X}) \leq \left[1 + (d+1)e^{d-1}M^d\right]\ln\left(1 + \frac{T}{\sigma^2}\right) + \frac{C_{d,\theta}T}{\sigma^2}(d+1)e^{d-1}\sum_{m=M+1}^{\infty}\left(\frac{ec_d}{\theta m}\right)^m. \tag{216}$$

In the remaining proof, we consider the upper bound of $\left(\frac{ec_d}{\theta m}\right)^m$ separately based on $\theta > 0$. If $\theta \leq e^2 c_d$, we have

$$\left(\frac{ec_d}{\theta m}\right)^m = \exp\left(-m\ln\left(\frac{\theta m}{ec_d}\right)\right) \leq \exp(-m) \tag{217}$$

for any $m$ such that $\frac{\theta m}{ec_d} \geq e \Leftrightarrow m \geq \frac{e^2 c_d}{\theta}$. Then, by noting that the condition $T/(e-1) \geq \sigma^2$ implies $\ln\left(1 + \frac{T}{\sigma^2}\right) \geq 1$, we have the following inequalities by setting $M = \left\lfloor \frac{e^2 c_d}{\theta}\ln\left(1 + \frac{T}{\sigma^2}\right)\right\rfloor$:

$$\sum_{m=M+1}^{\infty}\left(\frac{ec_d}{\theta m}\right)^m \leq \sum_{m=M+1}^{\infty}\exp(-m) \tag{218}$$

$$\leq \int_{M}^{\infty}\exp(-m)\mathrm{d}m \tag{219}$$

$$\leq \exp(-M) \tag{220}$$

$$\leq \exp\left(-\frac{e^2 c_d}{\theta}\ln\left(1 + \frac{T}{\sigma^2}\right) + 1\right) \tag{221}$$

$$\leq e\left(1 + \frac{T}{\sigma^2}\right)^{-\frac{e^2 c_d}{\theta}} \tag{222}$$

$$\leq e\left(1 + \frac{T}{\sigma^2}\right)^{-1} \tag{223}$$

$$\leq e\frac{\sigma^2}{T}. \tag{224}$$

Therefore, for $\theta \leq e^2 c_d$, the following inequality holds from Eqs. (216) and (224), and the definition of $M$:

$$\gamma_T(\mathcal{X}) \leq \left[1 + (d+1)e^{d-1}\left(\frac{e^2 c_d}{\theta}\ln\left(1 + \frac{T}{\sigma^2}\right)\right)^d\right]\ln\left(1 + \frac{T}{\sigma^2}\right) + eC_{d,\theta}(d+1)e^{d-1}. \tag{225}$$

Next, if $\theta > e^2 c_d$, we have

$$\left(\frac{ec_d}{\theta m}\right)^m = \exp\left(-m\ln\left(\frac{\theta m}{ec_d}\right)\right) \leq \exp\left(-m\ln\left(\frac{\theta}{ec_d}\right)\right) \tag{226}$$

for any $m \in \mathbb{N}_+$. Then, similarly to the proof under $\theta \leq e^2 c_d$, we have the following inequalities for any $M \in \mathbb{N}$:

$$\sum_{m=M+1}^{\infty}\left(\frac{ec_d}{\theta m}\right)^m \leq \sum_{m=M+1}^{\infty}\exp\left(-m\ln\left(\frac{\theta}{ec_d}\right)\right) \tag{227}$$

$$\leq \int_{M}^{\infty}\exp\left(-m\ln\left(\frac{\theta}{ec_d}\right)\right)\mathrm{d}m \tag{228}$$

$$= \frac{1}{\ln\left(\frac{\theta}{ec_d}\right)}\exp\left(-M\ln\left(\frac{\theta}{ec_d}\right)\right) \tag{229}$$

$$< \exp\left(-M\ln\left(\frac{\theta}{ec_d}\right)\right), \tag{230}$$

where the last inequality follows from $\theta/ec_d > e \Leftrightarrow \theta > e^2 c_d$. By setting $M = \left\lceil \frac{1}{\ln\left(\frac{\theta}{ec_d}\right)} \ln\left(1 + \frac{T}{\sigma^2}\right) \right\rceil$, we have

$$\sum_{m=M+1}^{\infty} \left(\frac{ec_d}{\theta m}\right)^m \leq \exp\left(-\ln\left(1 + \frac{T}{\sigma^2}\right)\right) = \left(1 + \frac{T}{\sigma^2}\right)^{-1} \leq \frac{\sigma^2}{T}. \tag{231}$$

Hence, for $\theta > e^2 c_d$, we have

$$\gamma_T(X) \leq \left[1 + (d+1)e^{d-1}\left(\frac{1}{\ln\left(\frac{\theta}{ec_d}\right)}\ln\left(1 + \frac{T}{\sigma^2}\right) + 1\right)^d\right]\ln\left(1 + \frac{T}{\sigma^2}\right) + C_{d,\theta}(d+1)e^{d-1}. \tag{232}$$

Finally, aligning Eqs. (225) and (232) by focusing on the dependence on $T$, $\sigma^2$, and $\theta$, we obtain the desired result.

**Matérn kernel.** Similarly to the proof for the SE kernel, for any $M \geq 2\nu$, we have

$$\frac{1}{2}\max_{\boldsymbol{x}_1,\ldots,\boldsymbol{x}_T \in \mathbb{S}^d} \ln\det(\boldsymbol{I}_T + \sigma^{-2}\mathbf{K}(\mathbf{X}_T, \mathbf{X}_T)) \tag{233}$$

$$\leq N_M \ln\left(1 + \frac{T}{\sigma^2}\right) + \frac{T}{|\mathbb{S}^d|\sigma^2}\sum_{m=M+1}^{\infty}\lambda_m N_{d+1,m} \tag{234}$$

$$\leq \left[1 + (d+1)e^{d-1}M^d\right]\ln\left(1 + \frac{T}{\sigma^2}\right) + \frac{T(d+1)e^{d-1}}{|\mathbb{S}^d|\sigma^2}\sum_{m=M+1}^{\infty}\lambda_m m^{d-1} \tag{235}$$

$$\leq \left[1 + (d+1)e^{d-1}M^d\right]\ln\left(1 + \frac{T}{\sigma^2}\right) + \frac{T(d+1)\widetilde{C}_{d,\nu}e^{d-1}}{|\mathbb{S}^d|\sigma^2\ell^{2\nu}}\sum_{m=M+1}^{\infty}m^{-2\nu-1} \tag{236}$$

$$= \left[1 + (d+1)e^{d-1}M^d\right]\ln\left(1 + \frac{T}{\sigma^2}\right) + \frac{T\overline{C}_{d,\nu}}{\sigma^2\ell^{2\nu}}\sum_{m=M+1}^{\infty}m^{-2\nu-1}, \tag{237}$$

where the second inequality follows from Lemma 16, and the third inequality follows from $M \geq 2\nu$ and Lemma 19. In the last equation, we set $\overline{C}_{d,\nu} = \frac{(d+1)\widetilde{C}_{d,\nu}e^{d-1}}{|\mathbb{S}^d|}$. Furthermore,

$$\sum_{m=M+1}^{\infty}m^{-2\nu-1} \leq \int_M^{\infty}m^{-2\nu-1}\mathrm{d}m = \frac{M^{-2\nu}}{2\nu}. \tag{238}$$

By balancing $M^d\ln(1 + T/\sigma^2)$ and $\frac{TM^{-2\nu}}{\sigma^2\ell^{2\nu}}$ under the condition $M \geq 2\nu$, we set $M = \left\lceil\max\left\{2\nu, \left[\frac{T}{\sigma^2\ell^{2\nu}}\ln^{-1}\left(1 + \frac{T}{\sigma^2}\right)\right]^{1/(2\nu+d)}\right\}\right\rceil$. Then,

$$\frac{1}{2}\max_{\boldsymbol{x}_1,\ldots,\boldsymbol{x}_T \in \mathbb{S}^d} \ln\det(\boldsymbol{I}_T + \sigma^{-2}\mathbf{K}(\mathbf{X}_T, \mathbf{X}_T)) \tag{239}$$

$$\leq \left[1 + (d+1)e^{d-1}M^d\right]\ln\left(1 + \frac{T}{\sigma^2}\right) + \frac{T\overline{C}_{d,\nu}}{\sigma^2\ell^{2\nu}}\frac{M^{-2\nu}}{2\nu} \tag{240}$$

$$\leq \left[1 + (d+1)e^{d-1}M^d\right]\ln\left(1 + \frac{T}{\sigma^2}\right) + \frac{\overline{C}_{d,\nu}}{2\nu}M^d\ln\left(1 + \frac{T}{\sigma^2}\right) \tag{241}$$

$$= \ln\left(1 + \frac{T}{\sigma^2}\right) + \left[\frac{\overline{C}_{d,\nu}}{2\nu} + (d+1)e^{d-1}\right]M^d\ln\left(1 + \frac{T}{\sigma^2}\right) \tag{242}$$

$$= \ln\left(1 + \frac{T}{\sigma^2}\right) + C'_{d,\nu}\left((2\nu)^d + \left(\frac{T}{\sigma^2\ell^{2\nu}}\right)^{\frac{d}{2\nu+d}}\ln^{-\frac{d}{2\nu+d}}\left(1 + \frac{T}{\sigma^2}\right)\right)\ln\left(1 + \frac{T}{\sigma^2}\right) \tag{243}$$

$$= \left(C'_{d,\nu}(2\nu)^d + 1\right)\ln\left(1 + \frac{T}{\sigma^2}\right) + C'_{d,\nu}\left(\frac{T}{\sigma^2\ell^{2\nu}}\right)^{\frac{d}{2\nu+d}}\ln^{\frac{2\nu}{2\nu+d}}\left(1 + \frac{T}{\sigma^2}\right), \tag{244}$$

where we set $C'_{d,\nu} = \frac{\overline{C}_{d,\nu}}{2\nu} + (d+1)e^{d-1}$. In the above equations, the second inequality follows from

$$M^d \ln(1 + T/\sigma^2) \geq \frac{TM^{-2\nu}}{\sigma^2 \ell^{2\nu}} \Leftrightarrow \frac{\sigma^2 \ell^{2\nu}}{T} \ln(1 + T/\sigma^2) \geq M^{-2\nu-d} \quad (245)$$

$$\Leftrightarrow \left[ \frac{\sigma^2 \ell^{2\nu}}{T} \ln(1 + T/\sigma^2) \right]^{-\frac{1}{2\nu+d}} \leq M \quad (246)$$

$$\Leftrightarrow \left[ \frac{T}{\sigma^2 \ell^{2\nu}} \ln^{-1}(1 + T/\sigma^2) \right]^{\frac{1}{2\nu+d}} \leq M. \quad (247)$$

Finally, combining Eq. (244) with Lemma 9, we obtain the desired result [12]. □

## C  Auxiliary Lemmas

**Lemma 20** (Sub-optimality gap and the neighborhood around the maximizer). *Suppose $f$ is continuous. Then, under conditions 1 and 3 in Lemma 2, $\boldsymbol{x} \in \mathcal{B}_2(\rho_{\mathrm{quad}}; \boldsymbol{x}^*)$ holds for any $\boldsymbol{x} \in X$ such that $f(\boldsymbol{x}^*) - f(\boldsymbol{x}) \leq \varepsilon$ with $\varepsilon = \min\{c_{\mathrm{gap}}, c_{\mathrm{quad}}\rho_{\mathrm{quad}}^2\}$.*

*Proof.* When $\mathcal{B}_2(\rho_{\mathrm{quad}}; \boldsymbol{x}^*) = X$, the statement is trivial. Hereafter, we assume $\mathcal{B}_2(\rho_{\mathrm{quad}}; \boldsymbol{x}^*) \neq X$. Here, note that $f(\boldsymbol{x}^*) - f(\widetilde{\boldsymbol{x}}) \geq c_{\mathrm{quad}}\rho_{\mathrm{quad}}^2$ holds for any $\widetilde{\boldsymbol{x}} \in \mathcal{B}_2^b(\rho_{\mathrm{quad}}; \boldsymbol{x}^*)$ from condition 3 in Lemma 2, where $\mathcal{B}_2^b(\rho_{\mathrm{quad}}; \boldsymbol{x}^*) = \{\boldsymbol{x} \in X \mid \|\boldsymbol{x} - \boldsymbol{x}^*\|_2 = \rho_{\mathrm{quad}}\}$. Furthermore, from the continuity of $f$ and the compactness of $(X \setminus \mathcal{B}_2(\rho_{\mathrm{quad}}; \boldsymbol{x}^*)) \cup \mathcal{B}_2^b(\rho_{\mathrm{quad}}; \boldsymbol{x}^*)$, there exists $\widetilde{\boldsymbol{x}}_* \in \mathrm{argmax}_{\boldsymbol{x} \in (X \setminus \mathcal{B}_2(\rho_{\mathrm{quad}}; \boldsymbol{x}^*)) \cup \mathcal{B}_2^b(\rho_{\mathrm{quad}}; \boldsymbol{x}^*)} f(\boldsymbol{x})$. Then, we consider the following two cases separately.

- When $c_{\mathrm{gap}} \geq c_{\mathrm{quad}}\rho_{\mathrm{quad}}^2$, $\varepsilon = c_{\mathrm{quad}}\rho_{\mathrm{quad}}^2$ holds. If there exists $\boldsymbol{x} \in X \setminus \mathcal{B}_2(\rho_{\mathrm{quad}}; \boldsymbol{x}^*)$ such that $f(\boldsymbol{x}^*) - f(\boldsymbol{x}) \leq \varepsilon = c_{\mathrm{quad}}\rho_{\mathrm{quad}}^2$, we can choose $\widetilde{\boldsymbol{x}}_*$ such that $\widetilde{\boldsymbol{x}}_* \in X \setminus \mathcal{B}_2(\rho_{\mathrm{quad}}; \boldsymbol{x}^*)$ since $f(\boldsymbol{x}^*) - f(\widetilde{\boldsymbol{x}}) \geq c_{\mathrm{quad}}\rho_{\mathrm{quad}}^2$ holds for any $\widetilde{\boldsymbol{x}} \in \mathcal{B}_2^b(\rho_{\mathrm{quad}}; \boldsymbol{x}^*)$. Furthermore, such $\widetilde{\boldsymbol{x}}_*$ is the local maximizer on $X$, which satisfies $f(\boldsymbol{x}^*) - f(\widetilde{\boldsymbol{x}}_*) \leq f(\boldsymbol{x}^*) - f(\boldsymbol{x}) \leq \varepsilon_1 \leq c_{\mathrm{gap}}$. This contradicts condition 1 in Lemma 2.

- When $c_{\mathrm{gap}} < c_{\mathrm{quad}}\rho_{\mathrm{quad}}^2$, $\varepsilon = c_{\mathrm{gap}}$ holds. If there exists $\boldsymbol{x} \in X \setminus \mathcal{B}_2(\rho_{\mathrm{quad}}; \boldsymbol{x}^*)$ such that $f(\boldsymbol{x}^*) - f(\boldsymbol{x}) \leq \varepsilon = c_{\mathrm{gap}}$, we can choose $\widetilde{\boldsymbol{x}}_*$ such that $\widetilde{\boldsymbol{x}}_* \in X \setminus \mathcal{B}_2(\rho_{\mathrm{quad}}; \boldsymbol{x}^*)$ since $f(\boldsymbol{x}^*) - f(\widetilde{\boldsymbol{x}}) \geq c_{\mathrm{quad}}\rho_{\mathrm{quad}}^2 > c_{\mathrm{gap}}$ holds for any $\widetilde{\boldsymbol{x}} \in \mathcal{B}_2^b(\rho_{\mathrm{quad}}; \boldsymbol{x}^*)$. Furthermore, such $\widetilde{\boldsymbol{x}}_*$ is the local maximizer on $X$, which satisfies $f(\boldsymbol{x}^*) - f(\widetilde{\boldsymbol{x}}_*) \leq f(\boldsymbol{x}^*) - f(\boldsymbol{x}) \leq \varepsilon = c_{\mathrm{gap}}$. This contradicts condition 1 in Lemma 2.

From the above two arguments, we have $\forall \boldsymbol{x} \in X \setminus \mathcal{B}_2(\rho_{\mathrm{quad}}; \boldsymbol{x}^*), f(\boldsymbol{x}^*) - f(\boldsymbol{x}) > \varepsilon$. This implies that it is necessary to satisfy $\boldsymbol{x} \in \mathcal{B}_2(\rho_{\mathrm{quad}}; \boldsymbol{x}^*)$ under $f(\boldsymbol{x}^*) - f(\boldsymbol{x}) \leq \varepsilon$. □

**Lemma 21** (Upper bound of regret of GP-UCB for any index subset). *Fix any index set $\mathcal{T} \subset [T]$. Then, when running GP-UCB, we have the following inequality under $\mathcal{A}$:*

$$\sum_{t \in \mathcal{T}} f(\boldsymbol{x}^*) - f(\boldsymbol{x}_t) \leq 2\sqrt{C\beta_T |\mathcal{T}| I(\mathbf{X}_{\mathcal{T}})} + \frac{\pi^2}{6} \leq 2\sqrt{C\beta_T |\mathcal{T}| \gamma_{|\mathcal{T}|}(X)} + \frac{\pi^2}{6}, \quad (248)$$

*where $C = 2/\ln(1 + \sigma^{-2})$ and $\mathbf{X}_{\mathcal{T}} = (\boldsymbol{x}_t)_{t \in \mathcal{T}}$.*

*Proof.* By following the proof strategy of GP-UCB, we have

$$\sum_{t \in \mathcal{T}} f(\boldsymbol{x}^*) - f(\boldsymbol{x}_t) \leq \sum_{t \in \mathcal{T}} \frac{1}{t^2} + \sum_{t \in \mathcal{T}} f([\boldsymbol{x}^*]_t) - f(\boldsymbol{x}_t) \leq 2\beta_T^{1/2} \sum_{t \in \mathcal{T}} \sigma(\boldsymbol{x}_t; \mathbf{X}_{t-1}) + \frac{\pi^2}{6} \quad (249)$$

---

[12]Note that we need to adjust the noise variance parameter by a factor $1/\sqrt{2}$ from Lemma 9.

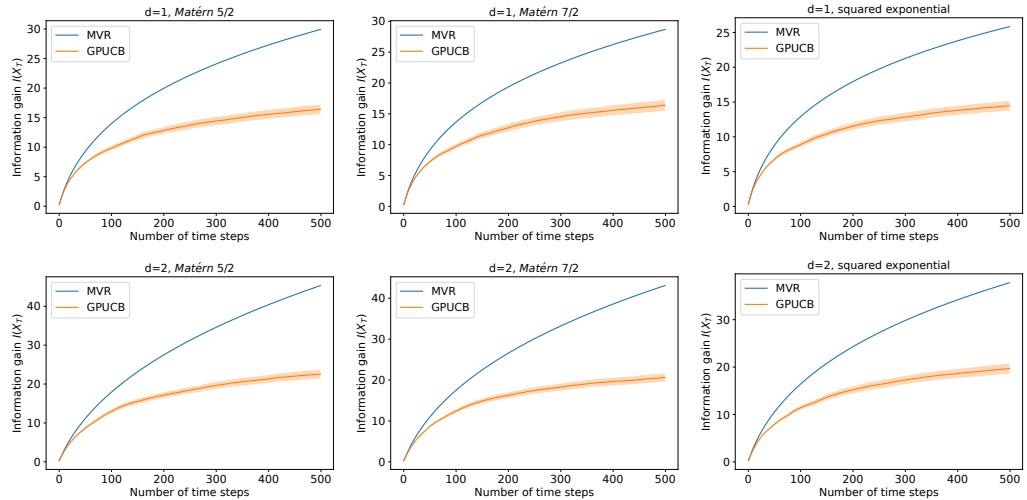

Figure 2: Average results of information gain under 20 different sample paths. The top row shows the results for the input space $\mathcal{X} = [0, 1]$ with lengthscale parameter $\ell = 0.1$. The bottom row corresponds to the input space $\mathcal{X} = [0, 1]^2$ with lengthscale $\ell = 0.25$, and from left to right, we use the SE kernel, the Matérn 5/2 kernel, and the Matérn 7/2 kernel. The shaded regions indicate one standard error.

due to event $\mathcal{A}$. Here, we define a new input sequence $(\widetilde{\boldsymbol{x}}_t)_{t \leq |\mathcal{T}|}$ as $\widetilde{\boldsymbol{x}}_t = \boldsymbol{x}_{j_t}$, where $j_t$ is the $t$-th element in $\mathcal{T}$. Furthermore, we define $\widetilde{\mathbf{X}}_t = (\widetilde{\boldsymbol{x}}_1, \ldots, \widetilde{\boldsymbol{x}}_t)$. Then, from $\widetilde{\mathbf{X}}_t \subset \mathbf{X}_{j_t}$ and the monotonicity of the posterior variance against the input data, we have

$$\sum_{t \in \mathcal{T}} \sigma(\boldsymbol{x}_t; \mathbf{X}_{t-1}) = \sum_{t=1}^{|\mathcal{T}|} \sigma(\widetilde{\boldsymbol{x}}_t; \mathbf{X}_{j_t - 1}) \tag{250}$$

$$\leq \sum_{t=1}^{|\mathcal{T}|} \sigma(\widetilde{\boldsymbol{x}}_t; \widetilde{\mathbf{X}}_{t-1}) \tag{251}$$

$$\leq \sqrt{C|\mathcal{T}| I(\widetilde{\mathbf{X}}_{|\mathcal{T}|})} \tag{252}$$

$$= \sqrt{C|\mathcal{T}| I(\mathbf{X}_\mathcal{T})} \tag{253}$$

$$\leq \sqrt{C|\mathcal{T}| \gamma_{|\mathcal{T}|}(\mathcal{X})}, \tag{254}$$

where the second inequality follows from Theorems 5.3 and 5.4 in [51]. □

# D   Numerical Simulation for Information Gain

In addition to the simple example provided in Figure 1, we empirically confirm the gap between the worst-case (MVR) and GP-UCB's information gain under the Bayesian assumption. In Figures 2 and 3, we report the average and the quantile of realized information gain with GP-UCB, over 20 different sample paths, generated by changing the random seed, respectively. We conduct experiments under the same settings as in Figure 1 of the main text. We also report the information gain corresponding to the sequence of maximum variance reduction (MVR), following the same setup as Figure 1. In all cases, consistent with Figure 1 in the main text, we observe a noticeable gap in information gain between GP-UCB and MVR.

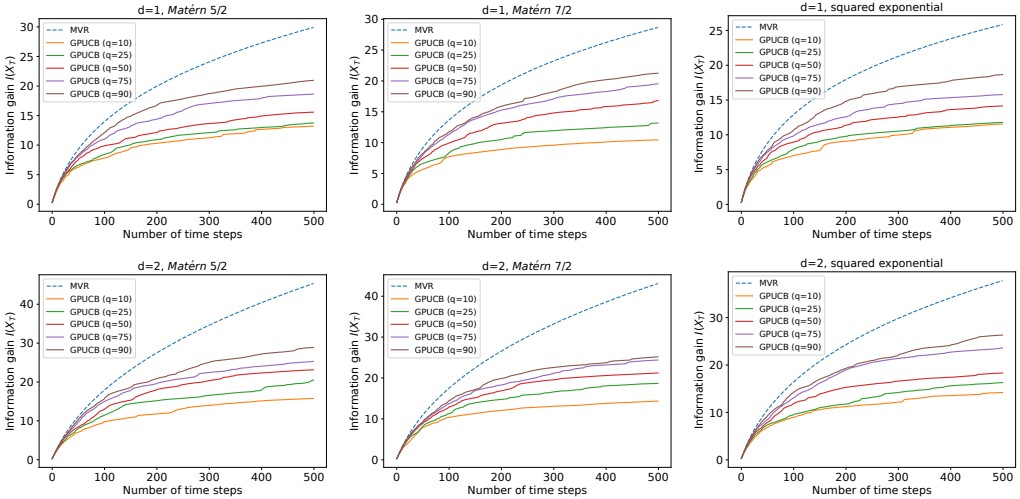

Figure 3: Quantiles of information gain over 20 different sample paths. We report the 10%, 25%, 50%, 75%, and 90% quantiles of the information gain of GP-UCB.

