# OpenReview forum: "Improved Regret Bounds for Gaussian Process Upper Confidence Bound in Bayesian Optimization"
_NeurIPS.cc/2025/Conference — NeurIPS 2025 oral_

### Official Review · Reviewer_ETNr · 2025-06-26

**Clarity:** 3
**Significance:** 3
**Originality:** 3
**Rating:** 5
**Confidence:** 2

**Summary:**

This paper studies the theoretical performance of the GP-UCB algorithm in Bayesian Optimization, which presents a refined analysis showing that the existing GP-UCB achieves
$\tilde{O}(\sqrt{T})$ regret for Matérn kernels and $O(\sqrt{T \ln^{4} T})$ regret for the squared exponential kernel under certain conditions. The core contribution lies in analyzing the actual input sequence generated by GP-UCB rather than relying solely on worst-case information gain bounds, and thus improves the analytical performance of GP-UCB.

**Questions:**

Could the authors provide more precise characterizations or examples where the realized information gain $I(X_T)$ deviates substantially from $\gamma_T$​? Are there quantifiable metrics for how close GP-UCB is to the ideal case in practice?

**Ethical Concerns:**

["NO or VERY MINOR ethics concerns only"]

**Final Justification:**

The authors have carefully addressed my questions. They emphasize that their main contribution is the derivation of Lemma 4, which they claim is the first non-worst-case information gain-based regret analysis for GP-UCB. While I'm not a 100% sure of this, this appears to be true and since it has been over 15 years since the original analysis in Srinivas et al. (2010), the contribution could be quite significant.

**Limitations:**

Yes

**Quality:**

3

**Strengths And Weaknesses:**

**Strengths**

1. This paper establishes new high-probability regret bounds for the standard GP-UCB algorithm, which improve over the classic bounds in Srinivas et al. [1] and match or approach state-of-the-art [2].

2. One advantage is that the algorithm analyzed in the paper does not require prior knowledge of regularity constants around the maximizer compared to state-of-the-art [2], making the result more practically relevant.

3.  The presentation is very clear and easy to follow. The core proof  idea of capturing the concentration behavior of the input sequence by GP-UCB is well illustrated.


**Weaknesses**

1. While the paper provides a refined regret analysis, the algorithm itself seems to remain the standard GP-UCB without modification. As such, the contribution is best viewed as an analytical extension rather than a methodological advance. It illustrates that an existing algorithm can achieve near-optimal performance under more careful analysis, but does not propose new algorithmic ideas or frameworks.

2. Aside from the regret decomposition built on the core insight of leveraging the concentration behavior of the input sequence under GP-UCB, the overall proof strategy appears to rely heavily on established techniques from prior work. While well-executed, this may limit the broader technical novelty of the paper.

3. The improved bound for the Matérn kernel holds only under specific smoothness conditions (e.g., $2v + d \leq v^2$, shown in Theorem3), which may not hold in many practical settings, especially in high dimensions.

4. While the paper is theoretical, a small empirical experiment could help illustrate how the actual information gain or input concentration behaves in practice, supporting the core intuition beyond the single example in Figure 1.

[1] Srinivas, Niranjan, et al. "Gaussian process optimization in the bandit setting: No regret and experimental design." ICML (2010).
[2] Jonathan Scarlett. "Tight regret bounds for Bayesian optimization in one dimension." ICML (2018).

---

> ### Author Rebuttal · Authors · 2025-07-30
>
> Thank you for taking the time to review our paper. However, we believe that we can resolve the reviewer's concerns that the reviewer raised at points 2-4 in "Weaknesses". We hope that the reviewer will reconsider the current score if our comments resolve the reviewer's initial concerns.
>
> **Regarding weakness 2: Aside from the regret decomposition built on the core insight of leveraging the concentration behavior of the input sequence under GP-UCB, the overall proof strategy appears to rely heavily on established techniques from prior work. While well-executed, this may limit the broader technical novelty of the paper.**
>
> Our novel technical contribution is not the regret decomposition but the derivation of Lemma 4, which provides a tighter regret based on a tighter quantification of the information gain terms. To our knowledge, there are no existing results or techniques to obtain such results. To obtain our result, we combine the sample path properties provided in Lemma 2 with worst-case regret upper bounds, leading to the first non-worst-case information gain-based regret analysis in the field. As detailed in Appendix A.1, to our knowledge, no existing work employs the same technical approach, and we believe the novelty in the derivation of Lemma 4 is clear and indisputable.
>
> Furthermore, as Reviewer Mw5W kindly pointed out, it has been over 15 years since the maximum information gain-based analysis for GP-UCB was introduced in [Srinivas et al., 2010], with no major updates to the theoretical analysis for GP-UCB since. Therefore, we believe that the new analytical approach proposed in this paper represents a significant milestone and could serve as a foundation for future refined analysis of BO algorithms.
>
> **Regarding weakness 3: The improved bound for the Matérn kernel holds only under specific smoothness conditions (e.g., $2\nu + d \leq \nu^2 $, shown in Theorem 3), which may not hold in many practical settings, especially in high dimensions.**
>
> These smoothness conditions are employed to derive upper bounds that match the \$\tilde{O}(\sqrt{T})\$ expected regret lower bound proven by [Scarlett, 2018], but they are not essential assumptions for the analysis itself. In order to claim $R\_T = \tilde{O}(\sqrt{T})$, the information gain-based term must become polylogarithmic order. For this, we require that the input space over which the information gain is defined decrease sufficiently fast. Here, the decreasing speed of the input domain of the information gain is derived from the worst-case upper bounds of GP-UCB. Therefore, greater smoothness guarantees faster convergence of this input space (more precisely, the decreasing speed of the input domain is governed by $\eta\_i$ in Lemma 4, which decays more quickly to zero as the smoothness increases). Based on these observations, to ensure a polylogarithmic information gain in Lemma 4, the input domain of the information gain must shrink quickly such that the effect of the dominant polynomial term of information gain vanishes, which leads to the condition $2\nu + d \leq \nu^2$. These points are also described in Lines 214-215 of the main paper.
>
> On the other hand, importantly, the analytical framework we propose can still be applied even when the smoothness condition in Theorem 3 is violated. In such cases, although a $\tilde{O}(\sqrt{T})$ regret is no longer guaranteed, our result still strictly improves upon the \$O(T^{\frac{\nu + d}{2\nu + d}})\$ bound originally provided by [Srinivas et al., 2010]. Concretely, when the smoothness condition is not satisfied,
> the term \$\tilde{O}(T^{\frac{d}{2\nu + d} \cdot \frac{2\nu + d - \nu^2}{2\nu + d}}) $ in Eq. (63) appears in the regret bound as a replacement for the standard maximum information gain bound \$\gamma\_T(\mathcal{X}) = \tilde{O}(T^{\frac{d}{2\nu + d}})$ used in existing GP-UCB analyses.
> Consequently, even if the smoothness condition does not hold,
> we can derive $\tilde{O}(\sqrt{T^{1 + \frac{d}{2\nu + d} \cdot \frac{2\nu + d - \nu^2}{2\nu + d}}}) $ regret, which is strictly tighter than $\tilde{O}(T^{\frac{\nu + d}{2\nu + d}}) := \tilde{O}(\sqrt{T^{1 + \frac{d}{2\nu + d}}})$ regret provided by [Srinivas et al., 2010].
>
> We focused on the \$\tilde{O}(\sqrt{T})\$ case in the main presentation for simplicity. Still, as the reviewer points out, this may have caused some confusion about whether the smoothness assumption is strictly necessary. We will therefore clarify this point in a revision by adding remarks to Theorem 3.
>
> **Regarding weakness 4: While the paper is theoretical, a small empirical experiment could help illustrate how the actual information gain or input concentration behaves in practice, supporting the core intuition beyond the single example in Figure 1.**
>
> We provide the additional experimental results, which empirically verify our claim that the information gain of GP-UCB is quite smaller than that of maximum variance reduction beyond the single example in Figure 1.
>
> **Table 1 below**. Average results of information gain under different sample paths.
>  We report the information gain of GP-UCB and MVR, averaging over the 20 different runs under different sample paths, using the same settings as in Figure 1 of the main text. In addition, we also report the ratio between MVR and GP-UCB's information gains.
> The results are reported under the input space $\mathcal{X} = [0,1]$ with lengthscale parameter $\ell = 0.1$ and $\mathcal{X} = [0,1]^2$ with $\ell = 0.25$. We use the SE kernel, Mat\'ern 5/2 kernel, and Mat\'ern 7/2 kernel. In all cases, consistent with Figure 1 in the main text, we observe noticeable gaps in information gain between GP-UCB and MVR, and these gaps diverge as $T$ increases.
> | | T=100 | T=200 | T=300 | T=400 | T = 500 |
> | --- | --- | --- | --- | --- | --- |
> | Matern 5/2 (d=1) MVR | 13.989 | 19.970 | 24.088 | 27.292 | 29.917 |
> | Matern 5/2 (d=1) GP-UCB | 9.839 | 12.829 | 14.428 | 15.563 | 16.381 |
> | Matern 5/2 (d=1) MVR/GP-UCB | 1.422 | 1.557 | 1.669 | 1.754 | 1.826 |
> | Matern 7/2 (d=1) MVR | 13.677 | 19.371 | 23.247 | 26.240 | 28.677 |
> | Matern 7/2 (d=1) GP-UCB | 9.735 | 12.742 | 14.526 | 15.580 | 16.393 |
> | Matern 7/2 (d=1) MVR/GP-UCB | 1.405 | 1.520 | 1.600 | 1.684 | 1.749 |
> | Squared exponential (d=1) MVR | 12.915 | 17.948 | 21.283 | 23.810 | 25.840 |
> | Squared exponential (d=1) GP-UCB | 8.910 | 11.542 | 12.847 | 13.820 | 14.453 |
> | Squared exponential (d=1) MVR/GP-UCB | 1.449 | 1.555 | 1.657 | 1.723 | 1.788 |
> | Matern 5/2 (d=2) MVR | 17.940 | 27.503 | 34.605 | 40.406 | 45.327 |
> | Matern 5/2 (d=2) GP-UCB | 13.015 | 17.152 | 19.630 | 21.322 | 22.572 |
> | Matern 5/2 (d=2) MVR/GP-UCB | 1.378 | 1.603 | 1.763 | 1.895 | 2.008 |
> | Matern 7/2 (d=2) MVR | 17.520 | 26.588 | 33.226 | 38.588 | 43.101 |
> | Matern 7/2 (d=2) GP-UCB | 12.431 | 16.282 | 18.275 | 19.626 | 20.641 |
> | Matern 7/2 (d=2) MVR/GP-UCB | 1.409 | 1.633 | 1.818 | 1.966 | 2.088 |
> | Squared exponential (d=2) MVR | 16.431 | 24.299 | 29.836 | 34.189 | 37.783 |
> | Squared exponential (d=2) GP-UCB | 11.437 | 15.204 | 17.274 | 18.640 | 19.679 |
> | Squared exponential (d=2) MVR/GP-UCB | 1.437 | 1.598 | 1.727 | 1.834 | 1.920 |
>
>
> **Table 2 below**.  90\% percentile of information gain under different sample paths.
> We report 90\% quantiles of the information gain of GP-UCB under 20 different sample paths generated with different seeds, using the same settings as in Figure 1 of the main text. Note that the information gain for MVR is completely determined only by the kernel and input space choice and is the same as those in Table 1 above, since the MVR algorithm is not dependent on the randomness of the underlying function and observation noises in the experiment.
> From these results, we can empirically confirm consistent gaps in information gain between GP-UCB and MVR under the sample paths generated with high probability.
> | | T=100 | T=200 | T=300 | T=400 | T = 500 |
> | --- | --- | --- | --- | --- | --- |
> | Matern 5/2 (d=1) MVR | 13.989 | 19.970 | 24.088 | 27.292 | 29.917 |
> | Matern 5/2 (d=1) GP-UCB | 12.110 | 16.764 | 18.720 | 19.964 | 20.965 |
> | Matern 5/2 (d=1) MVR/GP-UCB | 1.155 | 1.191 | 1.287 | 1.367 | 1.427 |
> | Matern 7/2 (d=1) MVR | 13.677 | 19.371 | 23.247 | 26.240 | 28.677 |
> | Matern 7/2 (d=1) GP-UCB | 11.722 | 15.895 | 18.245 | 20.203 | 21.264 |
> | Matern 7/2 (d=1) MVR/GP-UCB | 1.167 | 1.219 | 1.274 | 1.299 | 1.349 |
> | Squared exponential (d=1) MVR | 12.915 | 17.948 | 21.283 | 23.810 | 25.840 |
> | Squared exponential (d=1) GP-UCB | 10.655 | 14.805 | 16.902 | 17.777 | 18.652 |
> | Squared exponential (d=1) MVR/GP-UCB | 1.212 | 1.212 | 1.259 | 1.339 | 1.385 |
> | Matern 5/2 (d=2) MVR | 17.940 | 27.503 | 34.605 | 40.406 | 45.327 |
> | Matern 5/2 (d=2) GP-UCB | 15.764 | 20.887 | 25.024 | 27.206 | 28.872 |
> | Matern 5/2 (d=2) MVR/GP-UCB | 1.138 | 1.317 | 1.383 | 1.485 | 1.570 |
> | Matern 7/2 (d=2) MVR | 17.520 | 26.588 | 33.226 | 38.588 | 43.101 |
> | Matern 7/2 (d=2) GP-UCB | 14.443 | 19.860 | 22.577 | 23.740 | 25.186 |
> | Matern 7/2 (d=2) MVR/GP-UCB | 1.213 | 1.339 | 1.472 | 1.625 | 1.711 |
> | Squared exponential (d=2) MVR | 16.431 | 24.299 | 29.836 | 34.189 | 37.783 |
> | Squared exponential (d=2) GP-UCB | 14.235 | 19.312 | 22.091 | 24.240 | 26.316 |
> | Squared exponential (d=2) MVR/GP-UCB | 1.154 | 1.258 | 1.351 | 1.410 | 1.436 |

---

> > ### Comment · Reviewer_ETNr · 2025-08-04
> > **Response to author's comments**
> >
> > Thank you very much for your detailed response to my review. I believe I now have a better understanding of the contribution and have decided to increase my score.

---

> > > ### Author Response · Authors · 2025-08-04
> > >
> > > We are pleased that our responses have helped improve your understanding of our paper. Thank you again for taking the time to review our paper.

---

### Official Review · Reviewer_Mw5W · 2025-06-29

**Clarity:** 3
**Significance:** 4
**Originality:** 4
**Rating:** 6
**Confidence:** 4

**Summary:**

The paper presents an improved regret bound for the famous GP-UCB algorithm under Bayesian setting (i.e. Gaussian-Process Bandits). While standard GP-UCB uses the worst-case maximum information gain (MIG), corresponding to the scenario where the algorithm always queries the maximum variance, the paper observes that this worst-case bound will be very loose in the later stage of optimisation, where the algorithm queries point around the optimum. Authors split the total regret of the algorithm into two parts and show that the regret of this "later" stage asymptotically dominates the regret of the "early" stage. Then, due to the fact that the radius of the region in which the algorithm queries points in the "later" stage is smaller than that of full domain, this results in substantial reduction in MIG. Importantly, the resulting bound for SE kernel does not depend on dimensionality (previous bound had exponential dependance) and for the Matern kernel, the dependance of the bound on the dimensionality is significantly improved.

**Questions:**

- The authors present a very detailed analysis regarding the MIG in the reduced search space. I wonder is it really necessary to go through it and instead think about reducing the search space as increasing the length scale. Clearly if the radius of input space shrinks from $1$ to $\eta < 1$, then we could define a GP on the original space with radius $1$ with a longer length scale $\hat{l} = \frac{l}{\eta} $ and then map the points from the reduced space to the original space as $\hat{x} = \frac{x}{\eta}$. We can easily check that for a stationary kernel $k_{\hat{l}}(\hat{x},\hat{x}^\prime) = k(\frac{\hat{x}}{\hat{l}},\frac{\hat{x}^\prime}{\hat{l}}) = k(\frac{x}{l},\frac{x^\prime}{l}) = k_{l}(x,x^\prime)$ and thus $\max_{x:||x||\le \eta} \sigma_{l}(x)  = \max_{\hat{x}:||\hat{x}||\le 1} \sigma_{\hat{l}}(\hat{x})$. We could then just plug the MIG bounds with explicit dependence on lengthscale [1,2] and obtain the results that $\gamma_T(\{x: ||x||\le \eta\}) = \eta^d \gamma_T(\{x: ||x||\le 1\})$, which I believe is much stronger than $\frac{1}{\log^d(\frac{1}{\eta})}$ that authors derive. Is the analysis above correct for the problem setting authors consider? If so, would it make your final bound even stronger?

- Is it correct to think that the proposed result effectively says BO can "escape" the curse of dimensionality? This seems to contradict the common knowledge. Could authors explain whether this is what their result implies and if so, why it seems to contradict the empirical experience.

- In theorems authors state they treat dimensionality $d$ as constant. However, since the derived results seems monumental it terms of reduced dimensionality dependence, I would like to see the full order scaling with $d$ of the bound. I assume, for large enough $T$ this scaling will be dominated by the stated bound, but it would still give more context if authors were able to provide an explicit bound.

[1] Berkenkamp, Felix, Angela P. Schoellig, and Andreas Krause. "No-regret Bayesian optimization with unknown hyperparameters." Journal of Machine Learning Research 20.50 (2019): 1-24.

[2] Ziomek, Juliusz, Masaki Adachi, and Michael A. Osborne. "Bayesian Optimisation with Unknown Hyperparameters: Regret Bounds Logarithmically Closer to Optimal." The Thirty-eighth Annual Conference on Neural Information Processing Systems (2024).

**Ethical Concerns:**

["NO or VERY MINOR ethics concerns only"]

**Final Justification:**

My main issue was the fact that authors did not provide any reference for Lemmas 1 and 2. With those included, I do not have any serious criticism regarding the paper that was left unanswered, hence I decided to increase my score.

**Limitations:**

yes

**Quality:**

4

**Strengths And Weaknesses:**

Strengths:
+ The bound seems monumental, as it significantly improves the dependance on dimension of the problem. Previously it was believed that exponential scaling of the bound with dimensionality for more complex kernels is unavoidable due to the curse of dimensionality.
+ Although GP-UCB was known for years, it seems like no-one was able to significantly improve on the initial regret bound derived by (Srinivas et al, 2010). The fact that for this relatively simple algorithm one can derive much better regret guarantees than previously thought, is definitely very important to the community.
+ As far as I understand, the paper also presents the first non-worst case analysis of the MIG, which seems like a very important step for the community that was essentially relying on the worst-case analysis of (Srinivas et al, 2010) for the last 15 years.
+ I believe due to the novelty of the approach, the work has the potential to inspire a number of follow-up works, rethinking the regret bounds in BO in general

Weaknesses:
- For Lemmas 1 and 2, authors just state that these properties are existing and known, do not provide any proof for them and just cite other paper. The problem, however, is that none of the cited papers actually proved these properties (to the best of my knowledge), they merely used them as assumptions. As such it would appear, Lemmas 1 and 2 are Assumptions that have to be made for the presented analysis to hold, rather than being results that follow from previous assumptions. I thus believe that these Lemmas should be changed into Assumptions.
- In Corollary 8, page 23, just above equation 123, authors write $\gamma_T(\lambda^2)$. I believe in the adopted notation, $\gamma_T(\cdot)$ took a set of points as an argument, but $\lambda$ is a number, so not sure what is meant by that.
- (minor) I believe the formula for Matern kernel on page 3 is wrong, I believe the distance between points should be raised to the power of $\nu$

---

> ### Author Rebuttal · Authors · 2025-07-30
>
> Thank you for your overall positive assessments.
>
> **Q. For Lemmas 1 and 2, authors do not provide any proof for them and just cite other paper.  None of the cited papers actually proved these properties.**
>
> We agree that the referenced prior works do not provide fully self-contained proofs of these points, which may impose a burden on readers attempting to verify the soundness of Lemmas 1 and 2. To address this, we will add clarifications in the revision to specify exactly which parts of the prior works support the claims in Lemmas 1 and 2.
>
> **Regarding Lemma 1:**
>   Lemma 1 holds for stationary kernels of the form \$k(x, x') = \tilde{k}(|x - x'|)\$, provided that \$\tilde{k}\$ is four times differentiable, as shown in Theorem 5 of [Ghosal & Roy, 2006].
>   For the squared exponential (SE) kernel, \$\tilde{k}(t) = \exp(-t/(2\ell^2))\$ is clearly infinitely differentiable, and for the Matérn kernel, it is known that if \$\nu > k\$ for some integer \$k\$, then the kernel is \$2k\$-times differentiable (e.g., see Chapter 2.7 of [Stein, 1999]). Therefore, Lemma 1 holds for SE and Matérn kernels with $\nu > 2$. These descriptions are also stated in the last parts of Appendix A.2 of [Srinivas et al., 2010].
>
> **Regarding Lemma 2:**
>   The conditions stated in Lemma 2 are discussed in prior works such as [De Freitas et al.,2012, Scarlett, 2018], which confirm their validity for SE and Matérn kernels with \$\nu > 2\$. In addition, we will supplement references to fill in some missing details in these prior works:
>
>   - Property 1 is implied by the fact that the GP sample path has a unique maximizer almost surely under SE and Matérn kernels. This ensures the existence of a \$\delta\_{\mathrm{GP}}\$-dependent constant $c\_{\mathrm{gap}}$ satisfying Property 1.
> The fact that GP sample paths under SE and Matérn kernels almost surely have unique maxima is confirmed from existing results, such as Chapter 2.7 of [Garnett, 2023]
> - Property 2 follows from, e.g., the compactness of the domain $\mathcal{X}$ and the almost sure continuity of GP sample paths. These guarantee that the function values do not diverge on $\mathcal{X}$, implying the existence of an upper bound $c\_{\mathrm{sup}}$ that depends on \$\delta\_{\mathrm{GP}}\$. Since GP sample paths are almost surely continuous under SE and Matérn kernels, Property 2 holds automatically in these cases.
> - Properties 3 and 4 are discussed in detail following Theorem 5 of (the arXiv version of) [De Freitas et al., 2012], along with additional references. Like Properties 1 and 2, these properties hold almost surely using constants that depend on any realized sample path, which ensures the existence of constants (dependent on $\delta\_{\mathrm{GP}}$) satisfying Properties 3 and 4 with high probability.
>
> **Referencese**
> - De Freitas, N., Smola, A. J., & Zoghi, M. (2012). Exponential regret bounds for Gaussian process bandits with deterministic observations.
> - De Freitas, N., Smola, A., & Zoghi, M. (2012). Regret bounds for deterministic Gaussian process bandits. arXiv preprint.
> - Ghosal, S., & Roy, A. (2006). Posterior consistency of Gaussian process prior for nonparametric binary regression.
> - Scarlett, J. (2018). Tight regret bounds for Bayesian optimization in one dimension.
> - Stein, M. L. (1999). Interpolation of spatial data: some theory for kriging. Springer Science & Business Media.
> - Garnett, R. (2023). Bayesian optimization. Cambridge University Press.
>
> **Q. Authors write $\gamma_T(\lambda^2)...$**
>
> **Q. The formula for Matérn kernel on page 3 is wrong.**
>
> We correct these typos in the revision. Thank you for pointing these out.
>
> **Q. Could we just plug the MIG bounds with explicit dependence on lengthscale [1,2]  and obtain the stronger results ?**
>
> We would like to correct a misunderstanding. The existing upper bound of information gain for the squared exponential kernel, $\gamma\_T = O(\theta^{-d} \ln^{d+1} T)$, is derived under the assumption that the lengthscale parameter $\theta \coloneqq 2\ell^2$ satisfies $\theta \to 0$. Consequently, this rate is not applicable to our setting, where we must consider the case $\theta \to \infty$ for the analysis under the shrinking input domain. Specifically, in the proof of Proposition 2 in [Berkenkamp et al., 2019] (Appendix B.1), the validity of Eq. (21) requires that $\theta$ be upper bounded by some constant $\theta\_0$. Therefore, even for the squared exponential kernel, if the input space radius shrinks (i.e., the lengthscale grows), the result $\gamma\_T = O(\theta^{-d} \ln^{d+1} T)$ from existing work cannot be applied.
>
> - Berkenkamp, F., Schoellig, A. P., & Krause, A. (2019). No-regret Bayesian optimization with unknown hyperparameters.
>
> **Q. Is it correct to think that the proposed result effectively says BO can "escape" the curse of dimensionality?**
>
> At present, our analysis does not circumvent the curse of dimensionality. While we did not explicitly state this due to assuming $d$ is fixed, the implicit constants in our regret bounds do depend on $d$, and this dependence, as in existing GP-UCB results from [Srinivas et al., 2010], may grow exponentially with $d$. Hence, our theorems do not provide guarantees in high-dimensional regimes where $d, T \to \infty$. More precisely, as shown in Eq. (103) of the Appendix, if $d$ is not fixed, the joint dependence on $d$ and $T$ becomes $O(\ln^{2d+4} T + \sqrt{T \ln^4 T})$. Thus, in high-dimensional settings with $d \to \infty$ (as $T \to \infty$), the first $O(\ln^{2d+4} T)$ term becomes dominant, indicating that our bound is exposed to the curse of dimensionality. That said, our result still improves upon existing bounds by removing the exponential dependence on $d$ from the leading-order term of the regret bound. This makes our result stronger in fixed-dimensional settings. However, unlike existing works for high-dimensional settings that analyze simultaneous limits of $T \to \infty$ and $d \to \infty$, we do not claim to have resolved the curse of dimensionality.
>
> **Q. I would like to see the full order scaling with $d$ of the bound.**
>
> For the squared exponential kernel, the joint dependence on $T$ and $d$ can be deduced from Eqs. (103) and (115) in the appendix of our paper, resulting in a regret bound of \$R\_T = O(\ln^{2d+4} T + \sqrt{T \ln^4 T})\$. Furthermore, constants such as \$C\_d^{(3)}, C\_d^{(4)}, C\_d^{(5)}\$—which appear in the information gain bound—are also present in the regret bound as implicit constants and may exhibit exponential dependence on $d$, similar to the original analysis by [Srinivas et al., 2010]. Explicitly stating the dependence of these constants on $d$ would clutter the theorem unnecessarily. In fact, most previous works (including [Srinivas et al.,2010]) have chosen to hide the dependence on $d$. Following this precedent, we have opted not to detail the dependence of these constants on $d$ in our statements.

---

> ### Comment · Reviewer_Mw5W · 2025-08-03
>
> Thank you very much for addressing my comments and answering questions. I remain very positive about the paper and with the issues around Lemmas 1 and 2 resolved, I am willing to increase my score to 6 (Strong Accept).
>
> I would expect the authors to make following revisions for the camera ready version:
> - cite the proper references for Lemma 1 and 2 in a detailed way (e.g. Proposition/Theorem ... in ...)
> - highlight the scaling of the bound with $d$ in the main body
> - include short discussion (even 1-2 sentence) on the fact that it does not mean GP-UCB escapes curse of dimensionality

---

> > ### Author Response · Authors · 2025-08-03
> >
> > We thank the reviewer for your reply. We really appreciate your thoughtful feedback. We will carefully revise the points that the reviewer pointed out.

---

### Official Review · Reviewer_UNfA · 2025-07-02

**Clarity:** 3
**Significance:** 4
**Originality:** 3
**Rating:** 5
**Confidence:** 4

**Summary:**

The paper studies the standard Bayesian optimization problem under the assumptions that the objective function is sampled from a Gaussian Process prior and the noise is Gaussian. It focuses on the standard GP-UCB algorithm and provides an improved regret bound under certain assumptions that hold for the Matérn family of kernels. Specifically, the paper removes the dependence on the maximum information gain $\gamma_T$—which can grow polynomially with $T$—and replaces it with logarithmic terms, leading to tighter theoretical guarantees.

**Questions:**

1. Can the authors explain why the assumptions on the smoothness parameter of the kernel are necessary? Specifically, where does the proof break down if these assumptions are not satisfied?

2. Can the authors elaborate on why the proof technique used in Section 3.2 does not extend to GP-TS, in contrast to GP-UCB?

3. Can the authors comment on the noise-free setting? How would the results change if the noise variance were zero?

**Ethical Concerns:**

["NO or VERY MINOR ethics concerns only"]

**Final Justification:**

After reading the rebuttal and other reviews, I maintain my recommendation for acceptance of the paper.

**Limitations:**

yes

**Quality:**

4

**Strengths And Weaknesses:**

The paper fills a significant gap in the literature by deriving regret bounds for GP-UCB that match the known lower bound. I consider this a substantial contribution to the field.

A limitation is the assumption on the smoothness parameter of the kernel in Theorem 3, which the authors themselves acknowledge. However, the reason for this restriction is not clearly discussed, and it is difficult to understand the necessity of this assumption from either the main text or the appendix.

The main parts of proof are well explained in Section 3.2.

A very minor comment: the quotation marks around favorable on line 191 should be corrected.

---

> ### Author Rebuttal · Authors · 2025-07-30
>
> Thank you for your overall positive assessments.
>
> **Q. Can the authors explain why the assumptions on the smoothness parameter of the kernel are necessary? Specifically, where does the proof break down if these assumptions are not satisfied?**
>
> These smoothness conditions are employed to derive upper bounds that match the \$\tilde{O}(\sqrt{T})\$ expected regret lower bound proven by [Scarlett, 2018], but they are not essential assumptions for the analysis itself. In order to claim $R\_T = \tilde{O}(\sqrt{T})$, the information gain-based term must become polylogarithmic order. For this, we require that the input space over which the information gain is defined decrease sufficiently fast. Here, the decreasing speed of the input domain of the information gain is derived from the worst-case upper bounds of GP-UCB. Therefore, greater smoothness guarantees faster convergence of this input space (more precisely, the decreasing speed of the input domain is governed by $\eta\_i$ in Lemma 4, which decays more quickly to zero as the smoothness increases). Based on these observations, to ensure a polylogarithmic information gain in Lemma 4, the input domain of the information gain must shrink quickly such that the effect of the dominant polynomial term of information gain vanishes, which leads to the condition $2\nu + d \leq \nu^2$. These points are also described in Lines 214-215 of the main paper.
>
> On the other hand, importantly, the analytical framework we propose can still be applied even when the smoothness condition in Theorem 3 is violated. In such cases, although a $\tilde{O}(\sqrt{T})$ regret is no longer guaranteed, our result still strictly improves upon the \$O(T^{\frac{\nu + d}{2\nu + d}})\$ bound originally provided by [Srinivas et al., 2010]. Concretely, when the smoothness condition is not satisfied,
> the term \$\tilde{O}(T^{\frac{d}{2\nu + d} \cdot \frac{2\nu + d - \nu^2}{2\nu + d}}) $ in Eq. (63) appears in the regret bound as a replacement for the standard maximum information gain bound \$\gamma\_T(\mathcal{X}) = \tilde{O}(T^{\frac{d}{2\nu + d}})$ used in existing GP-UCB analyses.
> Consequently, even if the smoothness condition does not hold,
> we can derive $\tilde{O}(\sqrt{T^{1 + \frac{d}{2\nu + d} \cdot \frac{2\nu + d - \nu^2}{2\nu + d}}}) $ regret, which is strictly tighter than $\tilde{O}(T^{\frac{\nu + d}{2\nu + d}}) := \tilde{O}(\sqrt{T^{1 + \frac{d}{2\nu + d}}})$ regret provided by [Srinivas et al., 2010].
>
> We focused on the \$\tilde{O}(\sqrt{T})\$ case in the main presentation for simplicity. Still, as the reviewer points out, this may have caused some confusion about whether the smoothness assumption is strictly necessary. We will therefore clarify this point in a revision by adding remarks to Theorem 3.
>
>
> **Q. Can the authors elaborate on why the proof technique used in Section 3.2 does not extend to GP-TS, in contrast to GP-UCB?**
>
> The primary reason we can not extend our analysis to GP-TS is due to the unknown dependence of the constants in Lemma 2 on $\delta\_{GP}$. In Bayesian analyses of GP-TS (e.g., [Russo & Roy, 2014]), the expected regret is typically considered, which requires understanding how the regret scales with respect to the confidence level $\delta\_{GP} + \delta$. However, because the dependence of the constants in Lemma 2 on $\delta\_{GP}$ is unknown, such an analysis becomes technically challenging. To the best of our knowledge, no existing work in the literature has discussed this dependence, and therefore, within the limits of currently known theoretical tools in the Bayesian optimization field, our proof techniques cannot be straightforwardly extended to GP-TS. This issue is discussed in detail in Lines 250-264 of the main text.
>
> - Russo, D., & Van Roy, B. (2014). Learning to optimize via posterior sampling. Mathematics of Operations Research, 39(4), 1221-1243.
>
> **Q. Can the authors comment on the noise-free setting? How would the results change if the noise variance were zero?**
>
> Regarding the noise-free setting, at least one can reduce the analysis to that of the standard noisy GP-UCB and recover the same regret rate we establish. This can be done by considering a fixed, small positive noise variance in the GP prior and using the monotonicity of the posterior variance. This reduction has been analyzed in the frequentist setting (e.g., [Lyu et al., 2019]) and is also applicable to the analysis in the Bayesian setting. On the other hand, recent works in the frequentist GP-bandit literature have shown that one can achieve stronger regret by letting the noise variance \$\sigma^2 > 0\$ decay with the time horizon $T$ (e.g., [Flynn & Reeb, 2024; Iwazaki & Takeno, 2025; Iwazaki, 2025]). Combining such an approach with our analysis would require a nontrivial investigation into how the information gain behaves under the decaying noise variance parameter of GP. We believe that integrating our theory with existing noise-free analyses such as [Flynn & Reeb, 2024; Iwazaki & Takeno, 2025; Iwazaki, 2025] is a promising direction for future work.
>
> - Lyu, Y., Yuan, Y., & Tsang, I. W. (2019). Efficient batch black-box optimization with deterministic regret bounds. arXiv preprint arXiv:1905.10041.
> - Flynn, H., & Reeb, D. (2024). Tighter confidence bounds for sequential kernel regression. arXiv preprint arXiv:2403.12732.
> - Iwazaki, S., & Takeno, S (2025). Improved Regret Analysis in Gaussian Process Bandits: Optimality for Noiseless Reward, RKHS norm, and Non-Stationary Variance. In Forty-second International Conference on Machine Learning.
> - Iwazaki, S. (2025). Gaussian process upper confidence bound achieves nearly-optimal regret in noise-free Gaussian process bandits. arXiv preprint arXiv:2502.19006.

---

> > ### Comment · Reviewer_UNfA · 2025-08-04
> >
> > Thank you for your response. After reading the rebuttal and other reviews, I maintain my recommendation for acceptance of the paper.

---

### Official Review · Reviewer_aVgd · 2025-07-05

**Clarity:** 4
**Significance:** 4
**Originality:** 4
**Rating:** 5
**Confidence:** 4

**Summary:**

The paper improves the regret bound for the GP-UCP algorithm.

The algorithm applies to the following framework. There is a domain ${\cal X}$ (restricted to $[0,1]^d$ in the paper) and a zero-mean Gaussian random field on it (sorry, I prefer a field to a process in this context). For every $x\in{\cal X}$ there is a fandom value $f(x)$ so that all joint distributions of $(f(x_1), f(x_2),\ldots,f(x_n))$ are Gaussian, the means $Ef(x)$ are 0, and the covariances $\mathrm{cov}(x_1,x_2)$ are given by a known kernel function ${\cal K}(x_1,x_2)$. The learner can request a value at $x\in {\cal X}$ and get $f(x)+\varepsilon$ corrupted by i.i.d.\ Gaussian noise $\varepsilon=\varepsilon(x)$. Given a set of pairs $(x_i,f(x_i)+\varepsilon)$, the conditional mean $\mu(x)$ of $f(x)$ at an arbitrary $x$ is then given by the well-known kernel ridge regression formula $Y'(\lambda I +K)^{-1}k(x)$ .

Here the goal of the learner is to choose $x_t$ sequentially aiming to hit near $x^* \in\arg\max f $ as soon is possible. The learner's performance is measured by the cumulative regret $\sum_t (f(x^*) - f(x_t))$.

A natural approach here is to choose $x_t$ maximising $\mu$ as worked out on the basis of the previously observed values at $x_1,x_2,\ldots,x_{t-1}$. However, this suffers from the usual optimism under uncertainty faults: the value $\mu(x)$ may be high due to insufficient exploration. So instead of $\mu(x)$ Srinivas et al (2010) suggest optimising an upper confidence bound $\mu+\beta$.

This paper tunes the algorithm to improve the regret bounds of Srinivas et al (2010) for the rbf and Matern kernels. The resulting bounds get within the logarithmic factor to the known lower bounds.

The authors try to explain the intuition behind the improvement through information gain on a sequence.

**Questions:**

None

**Ethical Concerns:**

["NO or VERY MINOR ethics concerns only"]

**Final Justification:**

Upon reading the discussion, I am inclined to keep my assessment of the paper.

**Limitations:**

Yes

**Paper Formatting Concerns:**

Typos etc.

Page 4, line 105: there exists the constants -> there exist the constants

Page 5, line 152: Suppose Assumptions 1 and 2. -> Suppose Assumptions 1 and 2 hold.

**Quality:**

4

**Strengths And Weaknesses:**

The paper improves on state-of-the-art results, which are quite fundamental to machine learning. It does not quite match the lower bounds, but is really close. As a downside, I would mention that the analysis only applies to specific kernels.

---

> ### Author Rebuttal · Authors · 2025-07-30
>
> Thank you for your overall positive assessment. We will carefully incorporate the corrections for typos that the reviewer pointed out.

---

### Decision · Program_Chairs · 2025-09-17

**Decision:**

Accept (oral)

**Comment:**

This paper improves the regret bound for GP-UCB. This paper fills a significant gap in the literature between known upper and lower bounds. The way the improvement was done avoids the worst-case information gain, which is quite interesting on its own.

It seems that the paper is following the NeurIPS'25 formatting correctly. The line spacing is too large. Note:
> Use 10 point type, with a vertical spacing (leading) of 11 points

The vertical spacing seems off. Please fix this for the final version.